# ACCELERATED POLICY GRADIENT: ON THE NESTEROV MOMENTUM FOR REINFORCEMENT LEARNING

## ABSTRACT

Policy gradient methods have recently been shown to enjoy global convergence at a $\Theta(1/t)$ rate in the non-regularized tabular softmax setting. Accordingly, one important research question is whether this convergence rate can be further improved, with only first-order updates. In this paper, we answer the above question from the perspective of momentum by adapting the celebrated Nesterov's accelerated gradient (NAG) method to reinforcement learning (RL), termed *Accelerated Policy Gradient* (APG). To demonstrate the potential of APG in achieving faster global convergence, we formally show that with the true gradient, APG with softmax policy parametrization converges to an optimal policy at a $\tilde{O}(1/t^2)$ rate. To the best of our knowledge, this is the first characterization of the global convergence rate of NAG in the context of RL. Notably, our analysis relies on one interesting finding: Regardless of the initialization, APG could end up reaching a locally nearly-concave regime, where APG could benefit significantly from the momentum, within finite iterations. By means of numerical validation, we confirm that APG exhibits $\tilde{O}(1/t^2)$ rate as well as show that APG could significantly improve the convergence behavior over the standard policy gradient.

## 1 INTRODUCTION

Policy gradient (PG) is a fundamental technique utilized in the field of reinforcement learning (RL) for policy optimization. It operates by directly optimizing the RL objectives to determine the optimal policy, employing first-order derivatives similar to the gradient descent algorithm in the conventional optimization problems. Notably, PG has demonstrated empirical success (Mnih et al., 2016; Wang et al., 2016; Silver et al., 2014; Lillicrap et al., 2016; Schulman et al., 2017; Espeholt et al., 2018) and is supported by strong theoretical guarantees (Agarwal et al., 2021; Fazel et al., 2018; Liu et al., 2020; Bhandari & Russo, 2019; Mei et al., 2020; Wang et al., 2021; Mei et al., 2021a; 2022; Xiao, 2022). In a recent study by (Mei et al., 2020), they characterized the convergence rate of $\Theta(1/t)$ in the non-regularized tabular softmax setting. This convergence behavior aligns with that of the gradient descent algorithm for optimizing convex functions, despite that the RL objectives lack concave characteristics. Consequently, one critical open question arises as to whether this $\Theta(1/t)$ convergence rate can be further improved solely with first-order updates. In the realm of optimization, Nesterov's Accelerated Gradient (NAG) method, introduced by (Nesterov, 1983), is a first-order method originally designed for convex functions in order to improve the convergence rate to $O(1/t^2)$. Over the past decades since its introduction, to the best of our knowledge, NAG has never been formally analyzed or evaluated in the context of RL for its global convergence, mainly due to the non-concavity of the RL objective. Therefore, it is natural to ask the following research question: *Could Nesterov acceleration further improve the global convergence rate beyond the $\Theta(1/t)$ rate achieved by PG in RL?*

To answer this question, this paper introduces Accelerated Policy Gradient (APG), which utilizes Nesterov acceleration to address the policy optimization problem of RL. Despite the existing knowledge about the NAG methods from previous research (Beck & Teboulle, 2009a;b; Ghadimi & Lan, 2016; Krichene et al., 2015; Li & Lin, 2015; Su et al., 2014; Muehlebach & Jordan, 2019; Carmon et al., 2018), there remain several fundamental challenges in establishing the global convergence in the context of RL: (i) *NAG convergence results under nonconvex problems*: Although there is a plethora of theoretical works studying the convergence of NAG under general nonconvex problems,

these results only establish convergence to a stationary point. Under these conditions, we cannot determine global convergence in RL. Furthermore, it is not possible to assess whether the convergence rate improves beyond $\Theta(1/t)$ based on these results. (ii) *Inherent characteristics of the momentum term*: From an analytical perspective, the momentum term demonstrates intricate interactions with the previous updates. As a result, accurately quantifying the specific impact of momentum during the execution of APG poses a considerable challenge. Moreover, despite the valuable insights provided by the non-uniform Polyak-Łojasiewicz (PL) condition in the field of RL proposed by (Mei et al., 2020), the complex influences of the momentum term present a significant obstacle in determining the convergence rate of APG. (iii) *The nature of the unbounded optimal parameter under softmax parameterization*: A crucial factor in characterizing the sub-optimality gap in the theory of optimization is the norm of the distance between the initial parameter and the optimal parameter (Beck & Teboulle, 2009a;b; Jaggi, 2013; Ghadimi & Lan, 2016). However, in the case of softmax parameterization, the parameter of the optimal action tends to approach infinity. As a result, the norm involved in the sub-optimality gap becomes infinite, thereby hindering the characterization of the desired convergence rate.

**Our Contributions.** Despite the above challenges, we present an affirmative answer to the research question described above and provide the first characterization of the global convergence rate of NAG in the context of RL. Specifically, we present useful insights and novel techniques to tackle the above technical challenges: Regarding (i), we show that the RL objective enjoys local near-concavity in the proximity of the optimal policy, despite its non-concave global landscape. To better illustrate this, we start by presenting a motivating two-action bandit example, which demonstrates the local concavity directly via the corresponding sigmoid-type characteristic. Subsequently, we show that this intuitive argument could be extended to the general MDP case. Regarding (ii), we show that the locally-concave region is *absorbing* in the sense that even with the effect of the momentum term, the policy parameter could stay in the nearly locally-concave region indefinitely once it enters this region. This result is obtained by carefully quantifying the cumulative effect of each momentum term. Regarding (iii), we introduce a surrogate optimal parameter, which has bounded norm and induces a nearly-optimal policy, and thereby characterize the convergence rate of APG. We summarize the contributions of this paper as follows:

- We propose APG, which leverages the Nesterov's momentum scheme to accelerate the convergence performance of PG for RL. To the best of our knowledge, this is the first formal attempt at understanding the convergence of Nesterov momentum in the context of RL.

- To demonstrate the potential of APG in achieving fast global convergence, we formally establish that APG enjoys a $\tilde{O}(1/t^2)$ convergence rate under softmax policy parameterization[1]. To achieve this, we present several novel insights into RL and APG, including the local near-concavity property as well as the absorbing behavior of APG. Moreover, we show that these properties can also be applied to establish $\tilde{O}(1/t)$ convergence rate of PG, which is of independent interest. Furthermore, we further show that the derived rate for APG is tight (up to a logarithmic factor) by providing a $\Omega(1/t^2)$ lower bound of the sub-optimality gap.

- Through numerical validation on both bandit and MDP problems, we confirm that APG exhibits $\tilde{O}(1/t^2)$ rate and hence substantially improves the convergence behavior over the standard PG.

## 2 RELATED WORK

**Policy Gradient.** Policy gradient (Sutton et al., 1999) is a popular RL technique that directly optimizes the objective function by computing and using the gradient of the expected return with respect to the policy parameters. It has several popular variants, such as the REINFORCE algorithm (Williams, 1992), actor-critic methods (Konda & Tsitsiklis, 1999), trust region policy optimization (TRPO) (Schulman et al., 2015), and proximal policy optimization (PPO) (Schulman et al., 2017). Recently, policy gradient methods have been shown to enjoy global convergence. The global convergence of standard policy gradient methods under various settings has been proven by (Agarwal et al., 2021). Furthermore, (Mei et al., 2020) characterizes a $O(1/t)$ convergence rate of policy gradient based on a Polyak-Lojasiewicz condition under the non-regularized tabular softmax parameterization. Moreover, (Fazel et al., 2018; Liu et al., 2020; Wang et al., 2021; Xiao, 2022) conduct

---

[1]Note that this result does not contradict the $\Omega(1/t)$ lower bound of the sub-optimality gap of PG in (Mei et al., 2020). Please refer to Section 6.2 for a detailed discussion.

theoretical analyses of several variants of policy gradient methods under various policy parameterizations and establish the global convergence guarantees for these methods. In our work, we rigorously establish the accelerated $\tilde{O}(1/t^2)$ convergence rate for the proposed APG method under softmax parameterization.

**Accelerated Gradient.** Accelerated gradient methods (Nesterov, 1983; 2005; Beck & Teboulle, 2009b; Carmon et al., 2017; Jin et al., 2018) play a pivotal role in the optimization literature due to their ability to achieve faster convergence rates when compared to the conventional gradient descent algorithm. Notably, in the convex regimes, the accelerated gradient methods enjoy a convergence rate as fast as $O(1/t^2)$, surpassing the limited convergence rate $O(1/t)$ offered by the gradient descent algorithm. The superior convergence behavior could also be characterized from the perspective of ordinary differential equations (Su et al., 2014; Krichene et al., 2015; Muehlebach & Jordan, 2019). Additionally, in order to enhance the performance of accelerated gradient methods, several variants have been proposed. For instance, (Beck & Teboulle, 2009a) proposes a variant of the proximal accelerated gradient method which incorporates monotonicity to further improve its efficiency. (Ghadimi & Lan, 2016) presents a unified analytical framework for a family of accelerated gradient methods that can be applied to solve convex, non-convex, and stochastic optimization problems. Moreover, (Li & Lin, 2015) proposes an accelerated gradient approach with restart to achieve monotonic improvement with sufficient descent, providing convergence guarantees to stationary points for non-convex problems. Other restart mechanisms have also been applied and analyzed in multiple recent accelerated gradient methods (O'donoghue & Candes, 2015; Li & Lin, 2022). The above list of works is by no means exhaustive and is only meant to provide a brief overview of the accelerated gradient methods. Our paper introduces APG, a novel approach that combines accelerated gradient methods and policy gradient methods for RL. This integration enables a substantial acceleration of the convergence rate compared to the standard policy gradient method.

## 3 PRELIMINARIES

**Markov Decision Processes.** For a finite set $\mathcal{X}$, we use $\Delta(\mathcal{X})$ to denote a probability simplex over $\mathcal{X}$. We consider that a finite Markov decision process (MDP) $\mathcal{M} = (\mathcal{S}, \mathcal{A}, \mathcal{P}, r, \gamma, \rho)$ is determined by: (i) a finite state space $\mathcal{S}$, (ii) a finite action space $\mathcal{A}$, (iii) a transition kernel $\mathcal{P} : \mathcal{S} \times \mathcal{A} \to \Delta(\mathcal{S})$, determining the transition probability $\mathcal{P}(s'|s, a)$ from each state-action pair $(s, a)$ to the next state $s'$, (iv) a reward function $r : \mathcal{S} \times \mathcal{A} \to \mathbb{R}$, (v) a discount factor $\gamma \in [0, 1)$, and (vi) an initial state distribution $\rho \in \Delta(\mathcal{S})$. Given a policy $\pi : \mathcal{S} \to \Delta(\mathcal{A})$, the value of state $s$ under $\pi$ is defined as

$$V^\pi(s) := \mathbb{E}\left[ \sum_{t=0}^\infty \gamma^t r(s_t, a_t) \middle| \pi, s_0 = s \right]. \tag{1}$$

We use $\boldsymbol{V}^\pi$ to denote the vector of $V^\pi(s)$ of all the states $s \in \mathcal{S}$. The goal of the learner (or agent) is to search for a policy that maximizes the following objective function as $V^\pi(\rho) := \mathbb{E}_{s \sim \rho}[V^\pi(s)]$. The Q-value (or action-value) and the advantage function of $\pi$ at $(s, a) \in \mathcal{S} \times \mathcal{A}$ are defined as

$$Q^\pi(s, a) := r(s, a) + \gamma \sum_{s'} \mathcal{P}(s'|s, a) V^\pi(s') \ , \tag{2}$$

$$A^\pi(s, a) := Q^\pi(s, a) - V^\pi(s), \tag{3}$$

where the advantage function reflects the relative benefit of taking the action $a$ at state $s$ under policy $\pi$. The (discounted) state visitation distribution of $\pi$ is defined as

$$d_{s_0}^\pi(s) := (1 - \gamma) \sum_{t=0}^\infty \gamma^t Pr(s_t = s|s_0, \pi, \mathcal{P}), \tag{4}$$

which reflects how frequently the learner would visit the state $s$ under policy $\pi$. And we let $d_\rho^\pi(s) := \mathbb{E}_{s_0 \sim \rho}\left[ d_{s_0}^\pi(s) \right]$ be the expected state visitation distribution under the initial state distribution $\rho$. Given $\rho$, there exists an optimal policy $\pi^*$ such that

$$V^{\pi^*}(\rho) = \max_{\pi : \mathcal{S} \to \Delta(\mathcal{A})} V^\pi(\rho). \tag{5}$$

For ease of exposition, we denote $V^*(\rho) := V^{\pi^*}(\rho)$.

---

**Algorithm 1** Policy Gradient (PG) in (Mei et al., 2020)

---

**Input**: Learning rate $\eta = \frac{1}{L}$, where $L$ is the Lipschitz constant of the objective function $V^{\pi_\theta}(\mu)$.
**Initialize**: $\theta^{(1)}(s, a)$ for all $(s, a)$.
**for** $t = 1$ to $T$ **do**

$$\theta^{(t+1)} \leftarrow \theta^{(t)} + \eta \nabla_\theta V^{\pi_\theta}(\mu)\Big|_{\theta=\theta^{(t)}} \tag{6}$$

**end for**

---

Although obtaining the true initial state distribution $\rho$ in practical problems is challenging, it is fortunate that this challenge can be eased by considering other *surrogate* initial state distribution $\mu$ that are strictly positive for every state $s \in \mathcal{S}$. Notably, it can be demonstrated in the following theoretical proof that even in the absence of knowledge about $\rho$, convergence guarantees for $V^*(\rho)$ can still be obtained under the condition of strictly positive $\mu$. Hence, we make the following assumption, which has also been adopted by (Agarwal et al., 2021) and (Mei et al., 2020).

**Assumption 1** (**Strict positivity of surrogate initial state distribution**). *The surrogate initial state distribution satisfies* $\min_s \mu(s) > 0$.

Since $\mathcal{S} \times \mathcal{A}$ is finite, without loss of generality, we assume that the one-step reward is bounded in the $[0, 1]$ interval:

**Assumption 2** (**Bounded reward**). $r(s, a) \in [0, 1], \forall s \in \mathcal{S}, a \in \mathcal{A}$.

For simplicity, we assume that the optimal action is unique. This assumption can be relaxed by considering the sum of probabilities of all optimal actions in the theoretical results.

**Assumption 3** (**Unique optimal action**). *There is a unique optimal action* $a^*(s)$ *for each state* $s \in \mathcal{S}$.

**Softmax Parameterization.** For unconstrained $\theta \in \mathbb{R}^{|\mathcal{S}||\mathcal{A}|}$, the softmax parameterization of $\theta$ is defined as

$$\pi_\theta(\cdot|s) \coloneqq \frac{\exp(\theta_{s,\cdot})}{\sum_{a' \in \mathcal{A}} \exp(\theta_{s,a'})}.$$

We use the shorthand for denoting the optimal policy $\pi^* \coloneqq \pi_{\theta^*}$, where $\theta^*$ is the optimal policy parameter.

**Policy Gradient.** Policy gradient (Sutton et al., 1999) is a policy search technique that involves defining a set of policies parametrized by a finite-dimensional vector $\theta$ and searching for an optimal policy $\pi^*$ by exploring the space of parameters. This approach reduces the search for an optimal policy to a search in the parameters space. In policy gradient methods, the parameters are updated by the gradient of $V^{\pi_\theta}(\mu)$ with respect to $\theta$ under a surrogate initial state distribution $\mu \in \Delta(\mathcal{S})$. Algorithm 1 presents the pseudo code of PG provided by (Mei et al., 2020).

**Nesterov's Accelerated Gradient (NAG).** Nesterov's Accelerated Gradient (NAG) (Nesterov, 1983) is an optimization algorithm that utilizes a variant of momentum known as Nesterov's momentum to expedite the convergence rate. Specifically, it computes an intermediate "lookahead" estimate of the gradient by evaluating the objective function at a point slightly ahead of the current estimate. We provide the pseudo code of NAG method as Algorithm 4 in Appendix A.

**Notations.** Throughout the paper, we use $\|x\|$ to denote the $L_2$ norm of a real vector $x$.

## 4 METHODOLOGY

In this section, we present our proposed algorithm, Accelerated Policy Gradient (APG), which integrates Nesterov acceleration with gradient-based reinforcement learning algorithms. In Section 4.1, we introduce our central algorithm, APG. Subsequently, in Section 4.2, we provide a motivating example in the bandit setting to illustrate the convergence behavior of APG.

## 4.1 Accelerated Policy Gradient

We propose Accelerated Policy Gradient (APG) and present the pseudo code of our algorithm in Algorithm 2. Our algorithm design draws inspiration from the renowned and elegant Nesterov's accelerated gradient updates as introduced in (Su et al., 2014). For the sake of comparison, we include the pseudo code of the approach in (Su et al., 2014) as Algorithm 4 in Appendix A. We adapt these updates to the reinforcement learning objective, specifically $V^{\pi_\theta}(\mu)$. It is important to note that we will specify the learning rate $\eta^{(t)}$ in Lemma 2, as presented in Section 5.

In Algorithm 2, the gradient update is performed in (7). Following this, (9) calculates the momentum for our parameters, which represents a fundamental technique employed in accelerated gradient methods. It is worth noting that in (7), the gradient is computed with respect to $\omega^{(t-1)}$, which is the parameter that the momentum brings us to, rather than $\theta^{(t)}$ itself. This distinction sets apart (7) from the standard policy gradient updates (Algorithm 1).

**Remark 1.** Algorithm 2 leverages the monotone version of NAG, which was originally proposed in the seminal work (Beck & Teboulle, 2009a) with a restart mechanism as in (9) and has been widely adopted by variants of NAG, e.g., (Li & Lin, 2015; O'donoghue & Candes, 2015; Li & Lin, 2022).

**Remark 2.** Note that there is also an algorithm named Nesterov Accelerated Policy Gradient Without Restart Mechanisms (Algorithm 6), which directly follows the original NAG algorithm (Algorithm 4). However, in the absence of monotonicity, we must address the intertwined effects between the gradient and the momentum to achieve convergence results. Specifically, we empirically demonstrate that the non-monotonicity could occur during the updates of Algorithm 6. For more detailed information, please refer to Appendix G.

---

**Algorithm 2** Accelerated Policy Gradient (APG)

---

**Input**: Learning rate $\eta^{(t)} > 0$.
**Initialize**: $\theta^{(0)} \in \mathbb{R}^{|\mathcal{S}||\mathcal{A}|}, \tau^{(0)} = 0, \omega^{(0)} = \theta^{(0)}$.
**for** $t = 1$ to $T$ **do**

$$\theta^{(t)} \leftarrow \omega^{(t-1)} + \eta^{(t)} \nabla_\theta V^{\pi_\theta}(\mu)\Big|_{\theta=\omega^{(t-1)}} \tag{7}$$

$$\varphi^{(t)} \leftarrow \theta^{(t)} + \frac{t-1}{t+2}(\theta^{(t)} - \theta^{(t-1)}) \tag{8}$$

$$\omega^{(t)} \leftarrow \begin{cases} \varphi^{(t)}, & \text{if } V^{\pi_\varphi^{(t)}}(\mu) \geq V^{\pi_\theta^{(t)}}(\mu), \\ \theta^{(t)}, & \text{otherwise.} \end{cases} \tag{9}$$

**end for**

---

## 4.2 A Motivating Example of APG

Prior to the exposition of convergence analysis, we aim to provide further insights into why APG has the potential to attain a convergence rate of $\tilde{O}(1/t^2)$, especially under the intricate non-concave objectives in reinforcement learning.

Consider a simple two-action bandit with actions $a^*, a_2$ and reward function $r(a^*) = 1, r(a_2) = 0$. Accordingly, the objective we aim to optimize is $\mathbb{E}_{a \sim \pi_\theta}[r(a)] = \pi_\theta(a^*)$. By deriving the Hessian matrix with respect to our policy parameters $\theta_{a^*}$ and $\theta_{a_2}$, we could characterize the curvature of the objective function around the current policy parameters, which provides useful insights into its local concavity. Upon analyzing the Hessian matrix, we observe that it exhibits concavity when $\pi_\theta(a^*) \geq 0.5$. The detailed derivation is provided in Appendix E. The aforementioned observation implies that the objective function demonstrates *local concavity* when $\pi_\theta(a^*) \geq 0.5$. Since $\pi^*(a^*) = 1$, it follows that the objective function exhibits local concavity for the optimal policy $\pi^*$. As a result, if one initializes the policy with a high probability assigned to the optimal action $a^*$, then the policy would directly fall in the locally concave part of the objective function. This allows us to apply the theoretical findings from the existing convergence rate of NAG in (Nesterov, 1983), which has demonstrated convergence rates of $O(1/t^2)$ for convex problems. Based on this insight, we establish the global convergence rate of APG in the general MDP setting in Section 5.

## 5 CONVERGENCE ANALYSIS

In this section, we take an important first step towards understanding the convergence behavior of APG and discuss the theoretical results of APG in the general MDP setting under softmax parameterization. Due to the space limit, we defer the proofs of the following theorems to Appendix C and D.

### 5.1 ASYMPTOTIC CONVERGENCE OF APG

In this subsection, we will formally present the asymptotic convergence result of APG. This necessitates addressing several key challenges outlined in the introduction. We highlight the challenges tackled in our analysis as follows: (C1) *The existing results of first-order stationary points under NAG are not directly applicable*: Note that the asymptotic convergence of standard PG is built on the standard convergence result of gradient descent for non-convex problems (i.e., convergence to a first-order stationary point), as shown in (Agarwal et al., 2021). While it appears natural to follow the same approach for APG, two fundamental challenges are that the existing results of NAG for non-convex problems typically show best-iterate convergence (e.g., (Ghadimi & Lan, 2016)) rather than last-iterate convergence, and moreover these results hold under the assumption of a bounded domain (e.g., see Theorem 2 of (Ghadimi & Lan, 2016)), which does not hold under the softmax parameterization in RL as the domain of the policy parameters and the optimal $\theta$ could be unbounded. These are yet additional salient differences between APG and PG. (C2) *Characterization of the cumulative effect of each momentum term*: Based on (C1), even if the limiting value functions exist, another crucial obstacle is to precisely quantify the memory effect of the momentum term on the policy's overall evolution. To address this challenge, we thoroughly examine the accumulation of the gradient and momentum terms, as well as the APG updates, to offer an accurate characterization of the momentum's memory effect on the policy.

Despite the above, we are still able to tackle the above challenges and establish the asymptotic global convergence of APG as follows. Recall that optimal objective is defined by (5).

**Assumption 4.** *Under a surrogate initial state distribution $\mu$, for any two deterministic policies $\pi_1$ and $\pi_2$ with distinct value vectors (i.e., $\boldsymbol{V}^{\pi_1} \neq \boldsymbol{V}^{\pi_2}$), we have $V^{\pi_1}(\mu) \neq V^{\pi_2}(\mu)$.*

**Remark 3.** Assumption 4 is a rather mild condition and could be achieved by properly selecting a surrogate distribution $\mu$ in practice. For example, let $\pi$ and $\pi'$ be two deterministic policies with $\boldsymbol{V}^{\pi} \neq \boldsymbol{V}^{\pi'}$ and let $\Lambda := \{\mu' \in \Delta(\mathcal{S}) : V^{\pi}(\mu') = V^{\pi'}(\mu')\}$, which is a hyperplane in the $|\mathcal{S}|$-dimensional real space. If we draw a $\bar{\mu}$ uniformly at random from the simplex $\Delta(\mathcal{S})$ (e.g., from a symmetric Dirichlet distribution), then we know the event $\bar{\mu} \in \Lambda$ shall occur with probability zero.

In the subsequent analysis, we assume that Assumption 1, 2, 3, 4 are satisfied.

**Theorem 1 (Global convergence under softmax parameterization).** *Consider a tabular softmax parameterized policy $\pi_\theta$. Under APG with $\eta^{(t)} = \frac{t}{t+1} \cdot \frac{(1-\gamma)^3}{16}$, we have $V^{\pi_\theta^{(t)}}(s) \to V^*(s)$ as $t \to \infty$, for all $s \in \mathcal{S}$.*

The complete proof is provided in Appendix C. Specifically, we address the challenge (C1) in Appendix C.1 and (C2) in Appendix B.1

**Remark 4.** Note that Theorem 1 suggests the use of a time-varying learning rate $\eta^{(t)}$. This choice is related to one inherent issue of NAG: the choices of learning rate are typically different for the convex and the non-convex problems (e.g., (Ghadimi & Lan, 2016)). Recall from Section 4.2 that the RL objective could be locally concave around the optimal policy despite its non-concavity of the global landscape. To enable the use of the same learning rate scheme throughout the whole training process, we find that incorporating the ratio $t/(t+1)$ could achieve the best of both world.

### 5.2 CONVERGENCE RATE OF APG

In this subsection, we leverage the asymptotic convergence of APG and proceed to characterize the convergence rate of APG under softmax parameterization. For ease of notation, for each state we denote the actions as $a^*(s), a_2(s), \ldots, a_{|\mathcal{A}|}(s)$ such that $Q^*(s, a^*(s)) > Q^*(s, a_2(s)) \geq \cdots \geq Q^*(s, a_{|\mathcal{A}|}(s))$. We will begin by introducing the two most fundamental concepts throughout this paper: the $C$-nearly concave property and the feasible update domain, as follows:

**Definition 1** ($C$-**Near Concavity**). *A function $f$ is said to be $C$-nearly concave on a convex set $\mathcal{X}_\theta$, where $\mathcal{X}_\theta$ is $\theta$-dependent, if for any $\theta' \in \mathcal{X}_\theta$, we have*

$$f(\theta') \leq f(\theta) + C \cdot \left\langle \nabla f(\theta), \theta' - \theta \right\rangle, \tag{10}$$

*for some $C > 1$.*

**Remark 5.** It is worth noting that we introduce a relaxation in the first-order approximation of $\theta$ by a factor of $C > 1$. This adjustment is made to loosen the constraints of the standard concave condition for $\theta'$ that yield improvement.

**Definition 2.** *We define a set of update vectors as feasible update domain $\mathcal{U}$ such that*

$$\mathcal{U} := \{\boldsymbol{d} \in \mathbb{R}^{|\mathcal{S}| \times |\mathcal{A}|} : d(s, a^*(s)) > d(s, a), \; \forall s \in \mathcal{S}, a \neq a^*(s)\}, \tag{11}$$

*where $d(s, a) \in \mathbb{R}$ is the direction of the action $a$ at state $s$.*

Notably, we use $\mathcal{U}$ to help us characterize the locally nearly-concave regime as shown below and subsequently show that the updates of APG would all fall into $\mathcal{U}$ after some finite time.

**Lemma 1** (**Local $C$-Nearly Concavity; Informal**). *The objective function $\theta \mapsto V^{\pi_\theta}(\mu)$ is $C$-nearly concave on the set $\mathcal{X}_\theta = \{\theta' = \theta + \boldsymbol{d} : \theta \in \mathbb{R}^{|\mathcal{S}||\mathcal{A}|}, \boldsymbol{d} \in \mathcal{U}\}$ for any $\theta$ satisfying the following conditions: (i) $V^{\pi_\theta}(s) > Q^*(s, a_2(s))$ for all $s \in \mathcal{S}$; (ii) For some $M > 0$, $\theta_{s,a^*(s)} - \theta_{s,a} > M$, for all $s \in \mathcal{S}$, and $a \neq a^*(s)$.*

The formal information regarding the constant $M$ mentioned in Lemma 1 is provided in Appendix D.

**Remark 6.** Note that the notion of *weak-quasi-convexity*, as defined in (Guminov et al., 2017; Hardt et al., 2018; Bu & Mesbahi, 2020), shares similarities to our own definition in Definition 1. While their definition encompasses the direction from any $\theta$ to the optimal parameter $\theta^*$, ours in Lemma 1 is more relaxed. This is due to the fact that, by the definition of the feasible update domain, the parameter $\theta^*$ for the optimal policy $\pi^*$ in our problem is always contained within $\mathcal{X}_\theta$ for any $\theta$.

After deriving Lemma 1, our goal is to investigate whether APG can reach the local $C$-nearly concavity regime within a finite number of time steps. To address this, we establish Lemma 2, which guarantees the existence of a finite time $T$ such that our policy will indeed achieve the condition stated in Lemma 1 through the APG updates and remain within this region without exiting.

**Lemma 2.** *Consider a tabular softmax parameterized policy $\pi_\theta$. Under APG $\eta^{(t)} = \frac{t}{t+1} \cdot \frac{(1-\gamma)^3}{16}$, given any $M > 0$, there exists a finite time $T$ such that for all $t \geq T$, $s \in \mathcal{S}$, and $a \neq a^*(s)$, we have, (i) $\theta_{s,a^*(s)}^{(t)} - \theta_{s,a}^{(t)} > M$, (ii) $V^{\pi_\theta^{(t)}}(s) > Q^*(s, a_2(s))$, (iii) $\left.\frac{\partial V^{\pi_\theta}(\mu)}{\partial \theta_{s,a^*(s)}}\right|_{\theta = \omega^{(t)}} \geq 0 \geq \left.\frac{\partial V^{\pi_\theta}(\mu)}{\partial \theta_{s,a}}\right|_{\theta = \omega^{(t)}}$, (iv) $\omega_{s,a^*(s)}^{(t)} - \theta_{s,a^*(s)}^{(t)} \geq \omega_{s,a}^{(t)} - \theta_{s,a}^{(t)}$.*

**Remark 7.** Conditions (i) and (ii) are formulated to establish the local $C$-nearly concave conditions within a finite number of time steps. On the other hand, conditions (iii) and (iv) describe two essential properties for verifying that, after the finite time, all the update directions of APG fall within the feasible update domain $\mathcal{U}$. Consequently, the updates executed by APG are aligned with the nearly-concave structure. For a detailed proof regarding the examination of the feasible directions, please refer to Appendix D.

With the results of Lemma 1 and 2, we are able to establish the main result for APG under softmax parameterization, which is an $\tilde{O}(1/t^2)$ convergence rate.

**Theorem 2** (**Convergence Rate of APG; Informal**). *Consider a tabular softmax parameterized policy $\pi_\theta$. Under APG with $\eta^{(t)} = \frac{t}{t+1} \cdot \frac{(1-\gamma)^3}{16}$, there exists a finite time $T$ such that for all $t \geq T$, we have*

$$V^*(\rho) - V^{\pi_\theta^{(t)}}(\rho) \leq \frac{1}{1-\gamma} \left\| \frac{d_\rho^{\pi^*}}{\mu} \right\|_\infty \left( \frac{32(2+T)\left(\left\|\theta^{(T)}\right\| + 2\ln(t-T)\right)^2 + K_T}{(1-\gamma)^3(t+T+1)(t+T)} \right. \tag{12}$$

$$\left. + \frac{4|\mathcal{S}|}{(1-\gamma)^2} \left( \frac{|\mathcal{A}|-1}{(t-T)^2 + |\mathcal{A}|-1} \right) \right), \tag{13}$$

*where $K_T$ is a finite constant depending on the finite time $T$.*

**Remark 8.** It is important to highlight that the intriguing local $C$-near concavity property is not specific to APG. Specifically, we can also establish that PG enters the local $C$-nearly concave regime within finite time steps. Moreover, its updates align with the directions in $\mathcal{U}$, allowing us to view it as an optimization problem under the $C$-nearly concave objective. Accordingly, we can demonstrate that PG under softmax parameterization also enjoys $C$-near concavity and thereby achieves a convergence rate of $\tilde{O}(1/t)$. For more details, please refer to Appendix F.

**Remark 9.** It is important to note that the logarithmic factor in the sub-optimality gap is a consequence of the unbounded nature of the optimal parameter in softmax parameterization.

## 6 Discussions

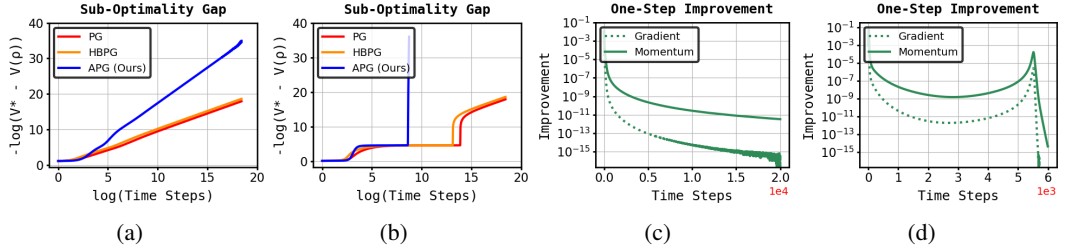

(a)    (b)    (c)    (d)

Figure 1: A comparison between the performance of APG, PG and HBPG under a 3-armed bandit with the uniform initialization and hard policy initialization: (a)-(b) show the sub-optimality gaps under uniform and hard initializations, respectively; (c)-(d) show the one-step improvements of APG from the momentum (i.e., ${\pi_\omega^{(t)}}^\top r - {\pi_\theta^{(t)}}^\top r$) and the gradient (i.e., ${\pi_\theta^{(t+1)}}^\top r - {\pi_\omega^{(t)}}^\top r$) under the uniform and the hard initialization, respectively.

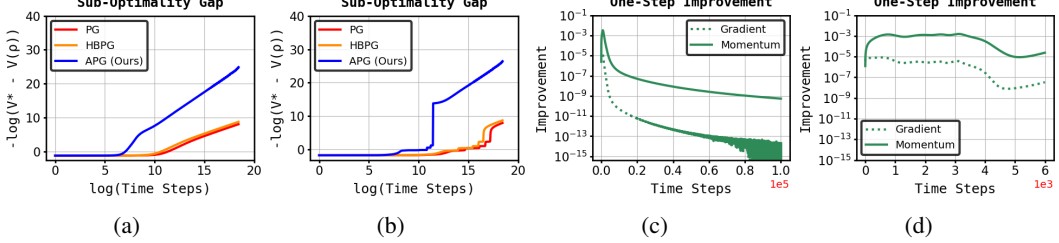

(a)    (b)    (c)    (d)

Figure 2: A comparison between the performance of APG, PG, and HBPG under an MDP with 5 states, 5 actions, with the uniform and hard policy initialization: (a)-(b) show the sub-optimality gaps under the uniform and the hard initialization, respectively; (c)-(d) show the one-step improvements of APG from the momentum (i.e., $V^{\pi_\omega^{(t)}}(\rho) - V^{\pi_\theta^{(t)}}(\rho)$) and the gradient (i.e., $V^{\pi_\theta^{(t+1)}}(\rho) - V^{\pi_\omega^{(t)}}(\rho)$), under the uniform and the hard initialization, respectively.

### 6.1 Numerical Validation of the Convergence Rates

In this subsection, we empirically validate the convergence rate of APG by conducting experiments on a 3-armed bandit as well as an MDP with 5 states and 5 actions. We validate the convergence behavior of APG and two popular methods, namely standard PG and PG with the heavy-ball momentum (HBPG) (Polyak, 1964). The detailed configuration is provided in Appendix E and the pseudo code of HBPG is provided as Algorithm 5 in Appendix A.

**(Bandit)** We first conduct a 3-armed bandit experiment with both a uniform initialization ($\theta^{(0)} = [0, 0, 0]$) and a hard initialization ($\theta^{(0)} = [1, 3, 5]$ and hence the optimal action has the smallest initial probability). First, upon plotting the sub-optimality gaps of PG, HBPG, and APG under uniform initialization on a log-log graph in Figure 1(a). We observe that both PG and HBPG exhibit

a slope of approximately 1, while APG demonstrates a slope of 2. These slopes match the respective convergence rates of $O(1/t)$ for PG and HBPG and the $\tilde{O}(1/t^2)$ convergence rate for APG, as shown in Theorem 2. Under the hard initialization, Figure 1(b) shows that APG could escape from sub-optimality much faster than PG and heavy-ball method and thereby enjoys fast convergence. Moreover, Figure 1(c)-1(d) further show that the momentum term in APG does contribute substantially in terms of policy improvement, under both initializations.

**(MDP)** We proceed to validate the convergence rate on an MDP with 5 states and 5 actions: (i) *Uniform initialization*: The training curves of value functions and sub-optimality gap for PG, HBPG, and APG are depicted in Figure 2. As shown by the log-log graph in Figure 2(a), the sub-optimality gap curves of APG exhibit a remarkable alignment with the $\tilde{O}(1/t^2)$ curve. Notably, Figure 2(a) also highlights that APG can converge significantly faster than the other two benchmark methods. Figure 2(c) further confirms that the momentum term in APG still contributes substantially in terms of policy improvement in the MDP case. (ii) *Hard initialization*: We also evaluate PG, HBPG, and APG under a hard policy initialization. Figure 2(b) and Figure 2(d) show that APG could still escape from sub-optimality much faster than PG and HBPG in the MDP case. This further showcases APG's superiority over PG and HBPG.

## 6.2 Lower Bounds of Policy Gradient

Regarding the fundamental capability of PG, (Mei et al., 2020) has presented a lower bound of sub-optimality gap for PG. For ease of exposition, we restate the theorem in (Mei et al., 2020) as follows.

**Theorem 3.** *(Lower bound of sub-optimality gap for PG in Theorem 10 of (Mei et al., 2020)). Take any MDP. For large enough $t \geq 1$, using Algorithm 1 with $\eta \in (0, 1]$,*

$$V^*(\mu) - V^{\pi_\theta^{(t)}}(\mu) \geq \frac{(1-\gamma)^5 \cdot (\Delta^*)^2}{12 \cdot t}, \tag{14}$$

*where $\Delta^* := \min_{s \in \mathcal{S}, a \neq a^*(s)} \{Q^*(s, a^*(s)) - Q^*(s, a)\} > 0$ is the optimal value gap of the MDP.*

Recall that PG has been shown to have a $O(1/t)$ convergence rate (Mei et al., 2020). Therefore, Theorem 3 indicates that the $O(1/t)$ convergence rate achievable by PG cannot be further improved.

On the other hand, despite the lower bound shown in Theorem 3 by (Mei et al., 2020), our results of APG do not contradict theirs. Specifically, while both APG and PG are first-order methods that rely solely on first-order derivatives for updates, it is crucial to highlight that APG encompasses a broader class of policy updates with the help of the momentum term in Nesterov acceleration. This allows APG to utilize the gradient with respect to parameters that PG cannot attain. As a result, APG exhibits improved convergence behavior compared to PG. Our findings extend beyond the scope of PG, demonstrating the advantages of APG in terms of convergence rate and overall performance.

## 7 Concluding Remarks

The Nesterov's Accelerated Gradient method, proposed in the optimization literature almost four decades ago, provides a powerful first-order scheme for fast convergence under a broad class of optimization problems. Over the past decades since its introduction, NAG has never been formally analyzed or evaluated in the context of RL for its global convergence, mainly due to the non-concavity of the RL objective. In this paper, we propose APG and take an important first step towards understanding NAG in RL. We rigorously show that APG can converge to a globally optimal policy at a $\tilde{O}(1/t^2)$ rate in the general MDP setting under softmax policies. This demonstrates the potential of APG in attaining fast convergence in RL.

On the other hand, our work also leaves open several interesting research questions: (i) Given that our convergence rate is tight up to a logarithmic factor, it remains open whether this limitation could be addressed by closing this logarithmic gap. (ii) As this paper mainly focuses on the exact gradient setting, another promising research direction is to extend our results of APG to the stochastic gradient setting, where the advantage function as well as the gradient are estimated from sampled transitions.

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

APPENDIX

CONTENTS

## A SUPPORTING ALGORITHM

For ease of exposition, we restate the accelerated gradient algorithm stated in (Ghadimi & Lan, 2016) as follows. Note that we've made several revisions so that one could easily compare Algorithm 2 and Algorithm 3: (i) We have exchanged the positions of the superscript and subscript. (ii) We've replaced the original gradient symbol with the gradient of our objective (i.e. $\nabla_\theta V^{\pi_\theta}(\mu)$). (iii) We've replaced the time variable $k$ with $t$. (iv) We've changed the algorithm into ascent algorithm (i.e. the sign in (16) and (17) is plus instead of minus.). (v) We've introduced a momentum variable $m^{(1)}$ in (15) in order to consider the non-zero initial momentum while entering the concave regime, which will be clarified in the subsequent lemma.

---

**Algorithm 3** The Accelerated Policy Gradient (APG) Algorithm Revised From Ghadimi & Lan (2016)

---

Input: $\theta^{(0)} \in \mathbb{R}^n$, $\{\alpha^{(t)}\}$ s.t. $\alpha^{(1)} = 1$ and $\alpha^{(t)} \in (0,1)$ for any $t \geq 2$, $\{\beta^{(t)} > 0\}$, and $\{\lambda^{(t)} > 0\}$, and $m^{(1)} \in \mathbb{R}^n$.

0. Set the initial points $\theta_{ag}^{(0)} = \theta^{(0)}$ and $t = 1$.
1. Set

$$\theta_{md}^{(t)} = \begin{cases} (1 - \alpha^{(t)})\theta_{ag}^{(t-1)} + \alpha^{(t)}\theta^{(t-1)} + m^{(1)}, & \text{if } t = 1, \\ (1 - \alpha^{(t)})\theta_{ag}^{(t-1)} + \alpha^{(t)}\theta^{(t-1)}, & \text{otherwise.} \end{cases} \tag{15}$$

2. Compute $\nabla\Psi(\theta_{md}^{(t)})$ and set

$$\theta^{(t)} = \theta^{(t-1)} + \lambda^{(t)}\nabla_\theta V^{\pi_\theta}(\mu)\Big|_{\theta=\theta_{md}^{(t)}}, \tag{16}$$

$$\theta_{ag}^{(t)} = \theta_{md}^{(t)} + \beta^{(t)}\nabla_\theta V^{\pi_\theta}(\mu)\Big|_{\theta=\theta_{md}^{(t)}}. \tag{17}$$

3. Set $t \leftarrow t + 1$ and go to step 1.

---

**Lemma 3** (Equivalence between Algorithm 2 and Algorithm 3)**.** *Using Algorithm 2 and setting $\alpha^{(t)}\lambda^{(t)} = \beta^{(t)}$, $\alpha^{(t)} = \frac{2}{t+1}, \forall t \geq 1$ and $m^{(1)} = \mathbf{0}$ leads to Algorithm 3 where $\eta^{(t)} = \beta^{(t)}$.*

**Remark 10.** Lemma 3 shows that our Algorithm 2 is equivalent to Algorithm 3 so that one could leverage the theoretical result stated in (Ghadimi & Lan, 2016) and adopt the general accelerated algorithm simultaneously.

*Proof of Lemma 3.* Since $\alpha^{(t)}\lambda^{(t)} = \beta^{(t)}$, by subtracting (17) from $\alpha^{(t)}$ times (16), we have:

$$\alpha^{(t)}\theta^{(t)} - \theta_{ag}^{(t)} = \alpha^{(t)}\theta^{(t-1)} - \theta_{md}^{(t)}. \tag{18}$$

Then, substituting $\theta_{md}^{(t)}$ in (18) by (15), we have:

$$\theta^{(t)} = \frac{\theta_{ag}^{(t)} - (1 - \alpha^{(t)})\theta_{ag}^{(t-1)}}{\alpha^{(t)}}. \tag{19}$$

Plugging (19) back into (15), we get:

$$\theta_{md}^{(t)} = (1 - \alpha^{(t)})\theta_{ag}^{(t-1)} + \alpha^{(t)}\theta^{(t-1)} \tag{20}$$

$$= (1 - \alpha^{(t)})\theta_{ag}^{(t-1)} + \alpha^{(t)}\frac{\theta_{ag}^{(t-1)} - (1 - \alpha^{(t-1)})\theta_{ag}^{(t-2)}}{\alpha^{(t-1)}} \tag{21}$$

$$= \theta_{ag}^{(t-1)} + \frac{\alpha^{(t)}(1 - \alpha^{(t-1)})}{\alpha^{(t-1)}}(\theta_{ag}^{(t-1)} - \theta_{ag}^{(t-2)}). \tag{22}$$

So by (15) and (22), we could simplify Algorithm 3 into a two variables update:

$$\theta_{md}^{(t)} = \theta_{ag}^{(t-1)} + \frac{\alpha^{(t)}(1 - \alpha^{(t-1)})}{\alpha^{(t-1)}}(\theta_{ag}^{(t-1)} - \theta_{ag}^{(t-2)}) \tag{23}$$

$$\theta_{ag}^{(t)} = \theta_{md}^{(t)} + \beta^{(t)} \nabla_\theta V^{\pi_\theta}(\mu)\Big|_{\theta=\theta_{md}^{(t)}} \tag{24}$$

Finally, by plugging $\alpha^{(t)} = \frac{2}{t+1}$ and $\beta^{(t)} = \eta^{(t)}$, we reach our desired result:

$$\theta_{md}^{(t)} = \theta_{ag}^{(t-1)} + \frac{t-2}{t+1}(\theta_{ag}^{(t-1)} - \theta_{ag}^{(t-2)}) \tag{25}$$

$$\theta_{ag}^{(t)} = \theta_{md}^{(t)} + \eta^{(t)} \nabla_\theta V^{\pi_\theta}(\mu)\Big|_{\theta=\theta_{md}^{(t)}} \tag{26}$$

Note that we've rearranged the ordering of (25) and (26) to reach our Algorithm 2. In summary, $\theta_{md}, \theta_{ag}$ in (25) and (26) corresponds to $\omega, \theta$ in Algorithm 2 respectively. And also we've turned the first step (25) into initializing $\omega$ in Algorithm 2 and follow the residual update.

$\square$

---

**Algorithm 4** Nesterov's Accelerated Gradient (NAG) algorithm in (Su et al., 2014)

---

**Input**: Learning rate $s = \frac{1}{L}$, where $L$ is the Lipschitz constant of the objective function $f$.
**Initialize**: $x_0$ and $y_0 = x_0$.
**for** $t = 1$ to $T$ **do**

$$x_k = y_{k-1} - s\nabla f(y_{k-1}) \tag{27}$$

$$y_k = x_k + \frac{k-1}{k+2}(x_k - x_{k-1}) \tag{28}$$

**end for**

---

**Algorithm 5** Heavy Ball Accelerated Gradient algorithm in (Ghadimi et al., 2015)

---

**Input**: Learning rate $s = \frac{1}{L}$, where $L$ is the Lipschitz constant of the objective function $f$.
**Initialize**: $x_0$.
**for** $t = 1$ to $T$ **do**

$$x_k = x_{k-1} - s\nabla f(x_{k-1}) + \beta(x_k - x_{k-1}) \tag{29}$$

**end for**

---

## B    Supporting Lemmas

### B.1    Useful Properties

For ease of notation, we use $\nabla_{s,a}^{(t)}$ as the shorthand for $\frac{\partial V^{\pi_\theta}(\mu)}{\partial \theta_{s,a}}\big|_{\theta=\omega^{(t)}}$. Moreover, in the sequel, for ease of exposition, for any pair of positive integers $(j,t)$, we define

$$
G(j,t) := \begin{cases}
1 & \text{, if } t = j, \\[2mm]
1 + \mathbb{I}\left\{V^{\pi_\varphi^{(j+1)}}(\mu) \geq V^{\pi_\theta^{(j+1)}}(\mu)\right\}\frac{j}{j+3} & \text{, if } t = j+1, \\[2mm]
1 + \mathbb{I}\left\{V^{\pi_\varphi^{(j+1)}}(\mu) \geq V^{\pi_\theta^{(j+1)}}(\mu)\right\}\frac{j}{j+3} & \\
\quad + \mathbb{I}\left\{V^{\pi_\varphi^{(j+2)}}(\mu) \geq V^{\pi_\theta^{(j+2)}}(\mu)\right\}\frac{(j+1)j}{(j+4)(j+3)} & \text{, if } t = j+2, \\[2mm]
1 + \mathbb{I}\left\{V^{\pi_\varphi^{(j+1)}}(\mu) \geq V^{\pi_\theta^{(j+1)}}(\mu)\right\}\frac{j}{j+3} & \\
\quad + \mathbb{I}\left\{V^{\pi_\varphi^{(j+2)}}(\mu) \geq V^{\pi_\theta^{(j+2)}}(\mu)\right\}\frac{(j+1)j}{(j+4)(j+3)} & \\
\quad + \mathbb{I}\left\{V^{\pi_\varphi^{(j+3)}}(\mu) \geq V^{\pi_\theta^{(j+3)}}(\mu)\right\}\frac{(j+2)(j+1)j}{(j+5)(j+4)(j+3)} & \text{, if } t = j+3, \\[2mm]
1 + \mathbb{I}\left\{V^{\pi_\varphi^{(j+1)}}(\mu) \geq V^{\pi_\theta^{(j+1)}}(\mu)\right\}\frac{j}{j+3} & \\
\quad + \mathbb{I}\left\{V^{\pi_\varphi^{(j+2)}}(\mu) \geq V^{\pi_\theta^{(j+2)}}(\mu)\right\}\frac{(j+1)j}{(j+4)(j+3)} & \\
\quad + \mathbb{I}\left\{V^{\pi_\varphi^{(j+3)}}(\mu) \geq V^{\pi_\theta^{(j+3)}}(\mu)\right\}\frac{(j+2)(j+1)j}{(j+5)(j+4)(j+3)} & \\
\quad + \sum_{k=4}^{t-j} \mathbb{I}\left\{V^{\pi_\varphi^{(j+k)}}(\mu) \geq V^{\pi_\theta^{(j+k)}}(\mu)\right\}\frac{(j+2)(j+1)j}{(j+k+2)(j+k+1)(j+k)} & \text{, if } t \geq j+4 \\[2mm]
0 & \text{, otherwise.}
\end{cases}
\tag{30}
$$

**Lemma 4.** *Under APG, we could express the policy parameter as follows:*

*a) For $t \in \{1,2,3,4\}$, we have*

$$
\theta_{s,a}^{(1)} = \eta^{(1)}\nabla_{s,a}^{(0)} + \theta_{s,a}^{(0)}, \tag{31}
$$

$$
\theta_{s,a}^{(2)} = \eta^{(2)}\nabla_{s,a}^{(1)} + \eta^{(1)}\nabla_{s,a}^{(0)} + \theta_{s,a}^{(0)} \tag{32}
$$

$$
\theta_{s,a}^{(3)} = \eta^{(3)}\nabla_{s,a}^{(2)} + \eta^{(2)}\left(1 + \mathbb{I}\left\{V^{\pi_\varphi^{(2)}}(\mu) \geq V^{\pi_\theta^{(2)}}(\mu)\right\} \cdot \frac{1}{4}\right)\nabla_{s,a}^{(1)} + \eta^{(1)}\nabla_{s,a}^{(0)} + \theta_{s,a}^{(0)} \tag{33}
$$

$$
\theta_{s,a}^{(4)} = \eta^{(4)}\nabla_{s,a}^{(3)} \tag{34}
$$

$$
+ \eta^{(3)}\left(1 + \mathbb{I}\left\{V^{\pi_\varphi^{(3)}}(\mu) \geq V^{\pi_\theta^{(3)}}(\mu)\right\} \cdot \frac{2}{5}\right)\nabla_{s,a}^{(2)} \tag{35}
$$

$$
+ \eta^{(2)}\left(1 + \mathbb{I}\left\{V^{\pi_\varphi^{(2)}}(\mu) \geq V^{\pi_\theta^{(2)}}(\mu)\right\} \cdot \frac{1}{4} + \mathbb{I}\left\{V^{\pi_\varphi^{(3)}}(\mu) \geq V^{\pi_\theta^{(3)}}(\mu)\right\} \cdot \frac{2 \cdot 1}{5 \cdot 4}\right)\nabla_{s,a}^{(1)} \tag{36}
$$

$$
+ \eta^{(1)}\nabla_{s,a}^{(0)} \tag{37}
$$

$$
+ \theta_{s,a}^{(0)} \tag{38}
$$

*b) For $t \geq 4$, we have*

$$
\theta_{s,a}^{(t+1)} = \eta^{(t+1)}\nabla_{s,a}^{(t)} \tag{39}
$$

$$
+ \eta^{(t)}\left(1 + \mathbb{I}\left\{V^{\pi_\varphi^{(t)}}(\mu) \geq V^{\pi_\theta^{(t)}}(\mu)\right\} \cdot \frac{t-1}{t+2}\right)\nabla_{s,a}^{(t-1)} \tag{40}
$$

$$
+ \eta^{(t-1)}\Big(1 + \mathbb{I}\left\{V^{\pi_\varphi^{(t-1)}}(\mu) \geq V^{\pi_\theta^{(t-1)}}(\mu)\right\} \cdot \frac{t-2}{t+1} \tag{41}
$$

$$
\quad + \mathbb{I}\left\{V^{\pi_\varphi^{(t)}}(\mu) \geq V^{\pi_\theta^{(t)}}(\mu)\right\} \cdot \frac{(t-1)(t-2)}{(t+2)(t+1)}\Big)\nabla_{s,a}^{(t-2)} \tag{42}
$$

$$
+ \sum_{j=1}^{t-3} \eta^{(j+1)}\left(1 + \mathbb{I}\left\{V^{\pi_\varphi^{(j+1)}}(\mu) \geq V^{\pi_\theta^{(j+1)}}(\mu)\right\} \cdot \frac{j}{j+3}\right) \tag{43}
$$

$$+ \mathbb{I}\left\{V^{\pi_\varphi^{(j+2)}}(\mu) \geq V^{\pi_\theta^{(j+2)}}(\mu)\right\} \cdot \frac{(j+1)j}{(j+4)(j+3)} \tag{44}$$

$$+ \mathbb{I}\left\{V^{\pi_\varphi^{(j+3)}}(\mu) \geq V^{\pi_\theta^{(j+3)}}(\mu)\right\} \cdot \frac{(j+2)(j+1)j}{(j+5)(j+4)(j+3)} \tag{45}$$

$$+ \sum_{k=4}^{t-j} \mathbb{I}\left\{V^{\pi_\varphi^{(j+k)}}(\mu) \geq V^{\pi_\theta^{(j+k)}}(\mu)\right\} \cdot \frac{(j+2)(j+1)j}{(j+k+2)(j+k+1)(j+k)}\right) \nabla_{s,a}^{(j)} \tag{46}$$

$$+ \eta^{(1)} \nabla_{s,a}^{(0)} \tag{47}$$

$$+ \theta_{s,a}^{(0)} \tag{48}$$

$$= \sum_{j=1}^{t} G(j,t) \cdot \eta^{(j+1)} \nabla_{s,a}^{(j)} + \eta^{(1)} \nabla_{s,a}^{(0)} + \theta_{s,a}^{(0)}. \tag{49}$$

*Proof of Lemma 4.* Regarding a), one could verify (31)-(38) by directly using the APG update in Algorithm 2. Regarding b), we prove this by induction. Specifically, suppose (39)-(49) hold for all iterations up to $t$. By the APG update, we know

$$\theta_{s,a}^{(t+1)} = \theta_{s,a}^{(t)} + \eta^{(t+1)} \nabla_{s,a}^{(t)} + \mathbb{I}\left\{V^{\pi_\varphi^{(t)}}(\mu) \geq V^{\pi_\theta^{(t)}}(\mu)\right\} \frac{t-1}{t+2}(\theta_{s,a}^{(t)} - \theta_{s,a}^{(t-1)}). \tag{50}$$

By plugging into (50) the expressions of $\theta_{s,a}^{(t)}$ and $\theta_{s,a}^{(t-1)}$ as suggested by (39)-(49), we could verify that (39)-(49) into hold for iteration $t+1$. $\qquad\square$

**Lemma 5 (Performance Difference Lemma in (Kakade & Langford, 2002)).** *For each state $s_0$, the difference in the value of $s_0$ between two policies $\pi$ and $\pi'$ can be characterized as:*

$$V^\pi(s_0) - V^{\pi'}(s_0) = \frac{1}{1-\gamma} \mathbb{E}_{s \sim d_{s_0}^\pi} \mathbb{E}_{a \sim \pi(\cdot|s)} \left[A^{\pi'}(s,a)\right]. \tag{51}$$

**Lemma 6 (Lemma 1. in (Mei et al., 2020)).** *Softmax policy gradient with respect to $\theta$ is*

$$\frac{\partial V^{\pi_\theta}(\mu)}{\partial \theta_{s,a}} = \frac{1}{1-\gamma} \cdot d_\mu^{\pi_\theta}(s) \cdot \pi_\theta(a|s) \cdot A^{\pi_\theta}(s,a). \tag{52}$$

**Lemma 7 (Equation in (Agarwal et al., 2021)).** *Softmax policy gradient with respect to $\pi_\theta$ is*

$$\frac{\partial V^{\pi_\theta}(\mu)}{\partial \pi_\theta(a|s)} = \frac{1}{1-\gamma} \cdot d_\mu^{\pi_\theta}(s) \cdot A^{\pi_\theta}(s,a). \tag{53}$$

**Lemma 8 (Lemma 2. in (Mei et al., 2020)).** $\forall r \in [0,1]^{|\mathcal{A}|}, \theta \to \pi_\theta^\top r$ is $5/2$-*smooth.*

**Lemma 9 (Lemma 7. in (Mei et al., 2020)).** $\theta \to V^{\pi_\theta}(\rho)$ is $\frac{8}{(1-\gamma)^3}$-*smooth.*

**Remark 11.** *Note that (Mei et al., 2020) not only establishes the smoothness of $V^{\pi_\theta}(\rho)$ but also the smoothness of $V^{\pi_\theta}(s)$ for all $s \in \mathcal{S}$.*

**Lemma 10.** $d_\mu^\pi(s) \geq (1-\gamma) \cdot \mu(s)$, *for any $\pi, s \in \mathcal{S}$ where $\mu(s)$ is some starting state distribution of the MDP.*

*Proof of Lemma 10.*

$$d_\mu^\pi(s) = \mathbb{E}_{s_0 \sim \mu}\left[d_\mu^\pi(s)\right] \tag{54}$$

$$= \mathbb{E}_{s_0 \sim \mu}\left[(1-\gamma) \cdot \sum_{t=0}^{\infty} \gamma^t \cdot \mathbb{P}(s_t = s \mid s_0, \pi)\right] \tag{55}$$

$$\geq \mathbb{E}_{s_0 \sim \mu}\left[(1-\gamma) \cdot \mathbb{P}(s_0 = s \mid s_0, \pi)\right] \tag{56}$$

$$= (1-\gamma) \cdot \mu(s). \tag{57}$$

The first equation holds by Lemma 5.

$\qquad\square$

**Lemma 11.** $V^*(\rho) - V^{\pi_\theta^{(t)}}(\rho) \leq \frac{1}{1-\gamma} \cdot \left\| \frac{d_\rho^{\pi^*}}{\mu} \right\|_\infty \cdot \left( V^*(\mu) - V^{\pi_\theta^{(t)}}(\mu) \right)$

*Proof of Lemma 11.*

$$V^*(\rho) - V^{\pi_\theta^{(t)}}(\rho) = \frac{1}{1-\gamma} \cdot \sum_{s \in \mathcal{S}} d_\rho^{\pi^*}(s) \sum_{a \in \mathcal{A}} \pi^*(a|s) \cdot A^{\pi_\theta^{(t)}}(s,a) \tag{58}$$

$$= \frac{1}{1-\gamma} \cdot \sum_{s \in \mathcal{S}} d_\mu^{\pi^*}(s) \cdot \frac{d_\rho^{\pi^*}(s)}{d_\mu^{\pi^*}(s)} \sum_{a' \in \mathcal{A}} \pi^*(a|s) \cdot A^{\pi_\theta^{(t)}}(s,a) \tag{59}$$

$$\leq \frac{1}{1-\gamma} \cdot \left\| \frac{d_\rho^{\pi^*}}{d_\mu^{\pi^*}} \right\|_\infty \cdot \sum_{s \in \mathcal{S}} d_\mu^{\pi^*}(s) \sum_{a' \in \mathcal{A}} \pi^*(a|s) \cdot A^{\pi_\theta^{(t)}}(s,a) \tag{60}$$

$$\leq \frac{1}{(1-\gamma)^2} \cdot \left\| \frac{d_\rho^{\pi^*}}{\mu} \right\|_\infty \cdot \sum_{s \in \mathcal{S}} d_\mu^{\pi^*}(s) \sum_{a' \in \mathcal{A}} \pi^*(a|s) \cdot A^{\pi_\theta^{(t)}}(s,a) \tag{61}$$

$$= \frac{1}{1-\gamma} \cdot \left\| \frac{d_\rho^{\pi^*}}{\mu} \right\|_\infty \cdot \left( V^*(\mu) - V^{\pi_\theta^{(t)}}(\mu) \right). \tag{62}$$

The first and the last equation holds by Lemma 5.

$\square$

**Lemma 12.** *Let $a^*(s)$ be the optimal action at the state $s$, and $a_i(s)$ where $2 \leq i \leq |\mathcal{A}|$ such that $Q^*(s, a^*(s)) \geq Q^*(s, a_2(s)) \geq \cdots \geq Q^*(s, a_{|\mathcal{A}|}(s))$. Then, for any policy $\pi$ and $a_i(s)$ for $2 \leq i \leq |\mathcal{A}|$, we have $Q^*(s, a_2(s)) \geq Q^\pi(s, a_i(s))$.*

*Proof of Lemma 12.* By the definition of $Q^*$, we have $Q^*(s,a) \coloneqq \sup_\pi Q^\pi(s,a)$ for all $(s,a) \in \mathcal{S} \times \mathcal{A}$. By Theorem 1.7 of (Agarwal et al., 2019), there exists a policy $\pi^*$ such that $Q^{\pi^*}(s,a) = Q^*(s,a)$ for all $(s.a) \in \mathcal{S} \times \mathcal{A}$. For any policy $\pi$ and $a_i(s)$ for $2 \leq i \leq |\mathcal{A}|$, we have

$$Q^\pi(s, a_i(s)) \leq \sup_{\pi'} Q^{\pi'}(s, a_i(s)) \tag{63}$$

$$\leq Q^*(s, a_i(s)) \tag{64}$$

$$\leq Q^*(s, a_2(s)). \tag{65}$$

where (63) holds by the definition of the supremum, (64) holds by the definition of $Q^*$ and (65) is because $a_2(s)$ is the best sub-optimal action at the state $s$. Since the argument holds for any given state $s \in \mathcal{S}$, we obtain our desired result. $\square$

**Lemma 13.** *Under APG, for any iteration $k$ and any state-action pair $(s,a)$, we have*

$$\sum_{a \in \mathcal{A}} \theta_{s,a}^{(k)} = \sum_{a \in \mathcal{A}} \theta_{s,a}^{(0)}. \tag{66}$$

*Proof of Lemma 13.* We prove this by induction and show the following two claims:

**Claim a).** $\sum_{a \in \mathcal{A}} \theta_{s,a}^{(1)} = \sum_{a \in \mathcal{A}} \theta_{s,a}^{(0)}$ and $\sum_{a \in \mathcal{A}} \theta_{s,a}^{(2)} = \sum_{a \in \mathcal{A}} \theta_{s,a}^{(0)}$.

Note that under APG, we have

$$\theta_{s,a}^{(1)} = \omega_{s,a}^{(0)} + \eta^{(1)} \frac{\partial V^{\pi_\theta}(\mu)}{\partial \theta_{s,a}} \Big|_{\theta = \omega^{(0)}} = \theta_{s,a}^{(0)} + \eta^{(1)} \cdot \frac{1}{1-\gamma} d^{\pi_\theta^{(0)}}(s) \pi_\theta^{(0)}(a|s) A^{\pi_\theta^{(0)}}(s,a), \tag{67}$$

where the second equality holds by the initial condition of APG (i.e., $\omega^{(0)} = \theta^{(0)}$) as well as the softmax policy gradient in Lemma 6. By taking the sum of (67) over all the actions, we have

$\sum_{a \in \mathcal{A}} \theta_{s,a}^{(1)} = \sum_{a \in \mathcal{A}} \theta_{s,a}^{(0)}$ due to the fact that $\sum_{a \in \mathcal{A}} \pi_\theta(a|s) A^{\pi_\theta}(s,a) = 0$, for any $\theta$. Similarly, we have

$$\theta_{s,a}^{(2)} = \omega_{s,a}^{(1)} + \eta^{(2)} \cdot \left. \frac{\partial V^{\pi_\theta}(\mu)}{\partial \theta_{s,a}} \right|_{\theta = \omega^{(1)}} \tag{68}$$

$$= \theta_{s,a}^{(1)} + \mathbb{I}\left\{ V^{\pi_\varphi^{(1)}}(\mu) \geq V^{\pi_\theta^{(1)}}(\mu) \right\} \cdot \frac{0}{3} \cdot (\theta_{s,a}^{(1)} - \theta_{s,a}^{(0)}) \tag{69}$$

$$+ \eta^{(2)} \cdot \frac{1}{1-\gamma} d^{\pi_\omega^{(1)}}(s) \pi_\omega^{(1)}(a|s) A^{\pi_\omega^{(1)}}(s,a). \tag{70}$$

By taking the sum of (69)-(70) over all the actions, we have $\sum_{a \in \mathcal{A}} \theta_{s,a}^{(2)} = \sum_{a \in \mathcal{A}} \theta_{s,a}^{(0)}$ by $\sum_{a \in \mathcal{A}} \theta_{s,a}^{(1)} = \sum_{a \in \mathcal{A}} \theta_{s,a}^{(0)}$ and the fact that $\sum_{a \in \mathcal{A}} \pi_\theta(a|s) A^{\pi_\theta}(s,a) = 0$, for all $\theta$.

**Claim b).** If $\sum_{a \in \mathcal{A}} \theta_{s,a}^{(k)} = \sum_{a \in \mathcal{A}} \theta_{s,a}^{(0)}$ for all $k \in \{1, \cdots, M\}$, then $\sum_{a \in \mathcal{A}} \theta_{s,a}^{(M+1)} = \sum_{a \in \mathcal{A}} \theta_{s,a}^{(0)}$. We use an argument similar to (69)-(70). That is,

$$\theta_{s,a}^{(M+1)} = \omega_{s,a}^{(M)} + \eta^{(M+1)} \cdot \left. \frac{\partial V^{\pi_\theta}(\mu)}{\partial \theta_{s,a}} \right|_{\theta = \omega^{(M)}} \tag{71}$$

$$= \theta_{s,a}^{(M)} + \mathbb{I}\left\{ V^{\pi_\varphi^{(M)}}(\mu) \geq V^{\pi_\theta^{(M)}}(\mu) \right\} \cdot \frac{M-1}{M+2} \cdot (\theta_{s,a}^{(M)} - \theta_{s,a}^{(M-1)}) \tag{72}$$

$$+ \eta^{(M+1)} \cdot \frac{1}{1-\gamma} d^{\pi_\omega^{(M)}}(s) \pi_\omega^{(M)}(a|s) A^{\pi_\omega^{(M)}}(s,a). \tag{73}$$

By taking the sum of (72)-(73) over all the actions, we could verify that $\sum_{a \in \mathcal{A}} \theta_{s,a}^{(M+1)} = \sum_{a \in \mathcal{A}} \theta_{s,a}^{(0)}$. $\square$

## B.2 $O(1/t^2)$ CONVERGENCE RATE UNDER NEARLY CONCAVE OBJECTIVES

For ease of exposition, we restate several theoretical results stated in (Ghadimi & Lan, 2016) as follows. Note that we have made a minor modification to ensure that Theorem 4 can be easily applied from the convex regime to the *nearly concave regime* without any loss of generality. This modification also allows for the use of a unified symbol across both regimes, providing a streamlined and consistent approach.

**Theorem 4 (Theorem 1(b) in (Ghadimi & Lan, 2016) with a slight modification).** *Let* $\left\{\theta_{md}^{(t)}, \theta_{ag}^{(t)}\right\}_{t \geq 1}$ *be computed by Algorithm 3 and* $\Gamma_t$ *be defined by:*

$$\Gamma^{(t)} := \begin{cases} 1, & t = 1 \\ (1 - \alpha^{(t)})\Gamma^{(t-1)}, & t \geq 2 \end{cases} \tag{74}$$

*Given a set* $\mathcal{X}$ *such that* $V^{\pi_\theta}(\mu)$ *is C-nearly concave in* $\mathcal{X}$. *Suppose* $\left\{\theta_{md}^{(t)}, \theta_{ag}^{(t)}\right\}_{t \geq 1}$ *always remain in the set* $\mathcal{X}$, *for all t. If* $\alpha^{(t)}, \beta^{(t)}, \lambda^{(t)}$ *are chosen such that*

$$0 < \alpha^{(t)}\lambda^{(t)} \leq \beta^{(t)} < \frac{2 - C}{L}, \tag{75}$$

$$\frac{\alpha^{(1)}}{\lambda^{(1)}\Gamma^{(1)}} \geq \frac{\alpha^{(2)}}{\lambda^{(2)}\Gamma^{(2)}} \geq \cdots, \tag{76}$$

*where L is the Lipschitz constant of the objective and* $C \leq \frac{3}{2}$ *is the constant defined in Definition 1. Then for any* $t \geq 1$ *and any* $\theta^{**}$, *we have*

$$V^{\pi_{\theta^{**}}}(\mu) - V^{\pi_{\theta_{ag}^{(t)}}}(\mu) \leq \Gamma^{(t)} \left[ \frac{\left\| \theta^{**} - \theta^{(0)} \right\|^2}{\lambda^{(1)}} + 2 \left| \left\langle \nabla_\theta V^{\pi_\theta}(\mu) \Big|_{\theta = \theta_{md}^{(1)}}, m^{(1)} \right\rangle \right| \right], \tag{77}$$

where $m^{(1)}$ is the initial momentum defined in Algorithm 3.

*Proof of Theorem 4.* First, by Lemma 9 and (17), we have

$$-V^{\pi_{\theta_{ag}^{(t)}}}(\mu) \leq -V^{\pi_{\theta_{md}^{(t)}}}(\mu) - \left\langle \nabla_\theta V^{\pi_\theta}(\mu) \Big|_{\theta = \theta_{md}^{(t)}}, \theta_{ag}^{(t)} - \theta_{md}^{(t)} \right\rangle + \frac{L}{2} \left\| \theta_{ag}^{(t)} - \theta_{md}^{(t)} \right\|^2 \tag{78}$$

$$= -V^{\pi_{\theta_{md}^{(t)}}}(\mu) - \beta^{(t)} \left\| \nabla_\theta V^{\pi_\theta}(\mu) \Big|_{\theta = \theta_{md}^{(t)}} \right\|^2 + \frac{L\beta^{(t)2}}{2} \left\| \nabla_\theta V^{\pi_\theta}(\mu) \Big|_{\theta = \theta_{md}^{(t)}} \right\|^2. \tag{79}$$

Also by the near-concavity of objective in the set $\mathcal{X}$, and (15), we have

$$- V^{\pi_{\theta_{md}^{(t)}}}(\mu) + \left[ (1 - \alpha^{(t)}) V^{\pi_{\theta_{ag}^{(t-1)}}}(\mu) + \alpha^{(t)} V^{\pi_{\theta^{**}}}(\mu) \right] \tag{80}$$

$$= \alpha^{(t)} \left[ V^{\pi_{\theta^{**}}}(\mu) - V^{\pi_{\theta_{md}^{(t)}}}(\mu) \right] + (1 - \alpha^{(t)}) \left[ V^{\pi_{\theta_{ag}^{(t-1)}}}(\mu) - V^{\pi_{\theta_{md}^{(t)}}}(\mu) \right] \tag{81}$$

$$\leq C \cdot \alpha^{(t)} \left\langle \nabla_\theta V^{\pi_\theta}(\mu) \Big|_{\theta = \theta_{md}^{(t)}}, \theta^{**} - \theta_{md}^{(t)} \right\rangle \tag{82}$$

$$+ C \cdot (1 - \alpha^{(t)}) \left\langle \nabla_\theta V^{\pi_\theta}(\mu) \Big|_{\theta = \theta_{md}^{(t)}}, \theta_{ag}^{(t-1)} - \theta_{md}^{(t)} \right\rangle \tag{83}$$

$$= C \cdot \left\langle \nabla_\theta V^{\pi_\theta}(\mu) \Big|_{\theta = \theta_{md}^{(t)}}, \alpha^{(t)}(\theta^{**} - \theta_{md}^{(t)}) + (1 - \alpha^{(t)})(\theta_{ag}^{(t-1)} - \theta_{md}^{(t)}) \right\rangle \tag{84}$$

$$= C \cdot \alpha^{(t)} \left\langle \nabla_\theta V^{\pi_\theta}(\mu) \Big|_{\theta = \theta_{md}^{(t)}}, \theta^{**} - \theta^{(t-1)} \right\rangle - C \cdot \left\langle \nabla_\theta V^{\pi_\theta}(\mu) \Big|_{\theta = \theta_{md}^{(t)}}, \mathbb{I}\{t = 1\} m^{(1)} \right\rangle. \tag{85}$$

And by (16), we have

$$\left\| \theta^{**} - \theta^{(t-1)} \right\|^2 - 2\lambda^{(t)} \left\langle \nabla_\theta V^{\pi_\theta}(\mu) \Big|_{\theta = \theta_{md}^{(t)}}, \theta^{**} - \theta^{(t-1)} \right\rangle + \lambda^{(t)2} \left\| \nabla_\theta V^{\pi_\theta}(\mu) \Big|_{\theta = \theta_{md}^{(t)}} \right\|^2 \tag{86}$$

$$= \left\| \theta^{**} - \theta^{(t-1)} - \lambda^{(t)} \nabla_\theta V^{\pi_\theta}(\mu) \Big|_{\theta=\theta_{md}^{(t)}} \right\|^2 = \left\| \theta^{**} - \theta^{(t)} \right\|^2, \tag{87}$$

which directly leads to

$$\alpha^{(t)} \left\langle \nabla_\theta V^{\pi_\theta}(\mu) \Big|_{\theta=\theta_{md}^{(t)}}, \theta^{**} - \theta^{(t-1)} \right\rangle = \frac{\alpha^{(t)}}{2\lambda^{(t)}} \left[ \left\| \theta^{**} - \theta^{(t-1)} \right\|^2 - \left\| \theta^{**} - \theta^{(t)} \right\|^2 \right] \tag{88}$$

$$+ \frac{\alpha^{(t)}\lambda^{(t)}}{2} \left\| \nabla_\theta V^{\pi_\theta}(\mu) \Big|_{\theta=\theta_{md}^{(t)}} \right\|^2. \tag{89}$$

Combining (78-89), we have

$$-V^{\pi_{\theta_{ag}}^{(t)}}(\mu) \leq - \left[ (1-\alpha^{(t)}) V^{\pi_{\theta_{ag}}^{(t-1)}}(\mu) + \alpha^{(t)} V^{\pi_{\theta^{**}}}(\mu) \right] \tag{90}$$

$$+ C \cdot \frac{\alpha^{(t)}}{2\lambda^{(t)}} \left[ \left\| \theta^{**} - \theta^{(t-1)} \right\|^2 - \left\| \theta^{**} - \theta^{(t)} \right\|^2 \right] \tag{91}$$

$$+ C \cdot \frac{\alpha^{(t)}\lambda^{(t)}}{2} \left\| \nabla_\theta V^{\pi_\theta}(\mu) \Big|_{\theta=\theta_{md}^{(t)}} \right\|^2 \tag{92}$$

$$- C \cdot \left\langle \nabla_\theta V^{\pi_\theta}(\mu) \Big|_{\theta=\theta_{md}^{(t)}}, \mathbb{I}\{t=1\} m^{(1)} \right\rangle \tag{93}$$

$$- \beta^{(t)} \left\| \nabla_\theta V^{\pi_\theta}(\mu) \Big|_{\theta=\theta_{md}^{(t)}} \right\|^2 \tag{94}$$

$$+ \frac{L\beta^{(t)^2}}{2} \left\| \nabla_\theta V^{\pi_\theta}(\mu) \Big|_{\theta=\theta_{md}^{(t)}} \right\|^2 \tag{95}$$

$$\leq - \left[ (1-\alpha^{(t)}) V^{\pi_{\theta_{ag}}^{(t-1)}}(\mu) + \alpha^{(t)} V^{\pi_{\theta^{**}}}(\mu) \right] \tag{96}$$

$$+ C \cdot \frac{\alpha^{(t)}}{2\lambda^{(t)}} \left[ \left\| \theta^{**} - \theta^{(t-1)} \right\|^2 - \left\| \theta^{**} - \theta^{(t)} \right\|^2 \right] \tag{97}$$

$$- \frac{\beta^{(t)}}{2} \left( 2 - C - L\beta^{(t)} \right) \left\| \nabla_\theta V^{\pi_\theta}(\mu) \Big|_{\theta=\theta_{md}^{(t)}} \right\|^2 \tag{98}$$

$$- C \cdot \left\langle \nabla_\theta V^{\pi_\theta}(\mu) \Big|_{\theta=\theta_{md}^{(t)}}, \mathbb{I}\{t=1\} m^{(1)} \right\rangle. \tag{99}$$

Adding $V^{\pi_{\theta^{**}}}(\mu)$ from both side of the above inequality and using (74), we have

$$\frac{V^{\pi_{\theta^{**}}}(\mu) - V^{\pi_{\theta_{ag}}^{(t)}}(\mu)}{\Gamma^{(t)}} \leq C \cdot \sum_{k=1}^{t} \frac{\alpha^{(k)}}{2\lambda^{(k)}\Gamma^{(k)}} \left[ \left\| \theta^{**} - \theta^{(k-1)} \right\|^2 - \left\| \theta^{**} - \theta^{(k)} \right\|^2 \right] \tag{100}$$

$$- \sum_{k=1}^{t} \frac{\beta^{(k)}}{2\Gamma^{(k)}} \underbrace{\left( 2 - C - L\beta^{(k)} \right)}_{>0} \left\| \nabla_\theta V^{\pi_\theta}(\mu) \Big|_{\theta=\theta_{md}^{(k)}} \right\|^2 \tag{101}$$

$$- C \cdot \left\langle \nabla_\theta V^{\pi_\theta}(\mu) \Big|_{\theta=\theta_{md}^{(1)}}, m^{(1)} \right\rangle \tag{102}$$

$$\leq \frac{C \cdot \left\| \theta^{**} - \theta^{(0)} \right\|^2}{2\lambda^{(1)}} - C \cdot \left\langle \nabla_\theta V^{\pi_\theta}(\mu) \Big|_{\theta=\theta_{md}^{(1)}}, m^{(1)} \right\rangle \tag{103}$$

$$\leq \frac{\left\| \theta^{**} - \theta^{(0)} \right\|^2}{\lambda^{(1)}} + 2 \underbrace{\left| \left\langle \nabla_\theta V^{\pi_\theta}(\mu) \Big|_{\theta=\theta_{md}^{(1)}}, m^{(1)} \right\rangle \right|}_{<\infty}, \tag{104}$$

where (103) holds by the fact that

$$\sum_{k=1}^{t} \frac{\alpha^{(k)}}{\lambda^{(k)}\Gamma^{(k)}} \left[ \left\| \theta^{**} - \theta^{(k-1)} \right\|^2 - \left\| \theta^{**} - \theta^{(k)} \right\|^2 \right] \le \frac{\alpha^{(1)} \left\| \theta^{**} - \theta^{(0)} \right\|^2}{\lambda^{(1)}\Gamma^{(1)}} = \frac{\left\| \theta^{**} - \theta^{(0)} \right\|^2}{\lambda^{(1)}},$$

(105)

and (104) holds since (75) leads to the fact that $2 > C$.

Finally, we obtain the desired result by rearranging (103). □

**Corollary 1** (**Corollary 1 in (Ghadimi & Lan, 2016) with a slight modification**). *Suppose that* $\left\{ \alpha^{(t)} \right\}$, $\left\{ \lambda^{(t)} \right\}$ *and* $\left\{ \beta^{(t)} \right\}$ *in Algorithm 3 are set to*

$$\alpha^{(t)} = \frac{2}{(t+1)+c}, \quad \lambda^{(t)} = \frac{(t+1)+c}{2} \cdot \beta^{(t)}, \quad \beta^{(t)} = \frac{t+c}{(t+1)+c} \cdot \frac{1}{2L}, \quad where\ c > 0.$$

(106)

*Given a set* $\mathcal{X}$ *such that* $V^{\pi_\theta}(\mu)$ *is* $\frac{3}{2}$*-nearly concave in* $\mathcal{X}$. *Suppose* $\left\{ \theta_{md}^{(t)}, \theta_{ag}^{(t)} \right\}_{t \ge 1}$ *always remain in the set* $\mathcal{X}$, *for all t. Then, for any* $t \ge 1$ *and any* $\theta^{**}$, *we have*

$$V^{\pi_{\theta^{**}}}(\mu) - V^{\pi_\theta^{(t)}}(\mu) \le \frac{4L(2+c)\left\| \theta^{(0)} - \theta^{**} \right\|^2 + 2(2+c)(1+c)\left| \left\langle \nabla_\theta V^{\pi_\theta}(\mu) \right|_{\theta=\theta_{md}^{(1)}}, m^{(1)} \right\rangle \right|}{(t+c+1)(t+c)}$$

(107)

$$= O(\frac{1}{t^2}).$$

(108)

**Remark 12.** Note that we have made the following minor modifications: (i) We have introduced a constant $c$ since our objective is not concave initially and hence the theoretical result had to be revised to account for the shifted initial learning rate. (ii) We have adjusted lambda from $\frac{t}{2}$ to $\frac{t+1}{2}$ and $\beta$ from $\frac{1}{2L}$ to $\frac{t+c}{(t+1)+c} \cdot \frac{1}{2L}$ to ensure the applicability of both Lemma 3 and Theorem 4 results. (iii) We've introduced a momentum variable $m^{(1)}$ in (15) in order to consider the non-zero initial momentum while entering the concave regime.

**Remark 13.** Note that Theorem 4 and Corollary 1 are built on the *local near concavity* of the objective function. In Appendix D, we formally show that such local near concavity indeed holds under APG in the MDP setting.

*Proof of Corollary 1.* We leverage Theorem 4 to reach our desired result. And it remains to show that the chosen of $\left\{ \alpha^{(t)}, \lambda^{(t)}, \beta^{(t)} \right\}$ in (106) satisfy (75) and (76). Note that $\alpha^{(t)} \cdot \lambda^{(t)} = \beta^{(t)} < \frac{1}{2L}$, (75) easily holds. And by the definition of $\Gamma^{(t)}$, we have:

$$\Gamma^{(t)} = \frac{(2+c)(1+c)}{((t+1)+c)(t+c)}.$$

(109)

Hence we have:

$$\frac{\alpha^{(t)}}{\lambda^{(t)}\Gamma^{(t)}} = \frac{\frac{2}{(t+1)+c}}{\frac{(t+1)+c}{2} \cdot \frac{t+c}{(t+1)+c} \cdot \frac{1}{2L} \cdot \frac{(2+c)(1+c)}{((t+1)+c)(t+c)}} = \frac{8L}{(2+c)(1+c)},$$

(110)

which makes the condition (76) holds. And hence we reach the desired result by plugging $\Gamma^{(t)}$ and $\lambda^{(1)}$ into (77). □

## C ASYMPTOTIC CONVERGENCE

### C.1 SUPPORTING LEMMAS FOR ASYMPTOTIC CONVERGENCE OF APG

**Lemma 14.** *Under APG, we have that both the limits of $V^{\pi_\omega^{(t)}}(\mu)$ and $V^{\pi_\theta^{(t)}}(\mu)$ shall exist and $\lim_{t\to\infty} V^{\pi_\omega^{(t)}}(\mu) = \lim_{t\to\infty} V^{\pi_\theta^{(t)}}(\mu)$. Moreover, for all state $s \in \mathcal{S}$,*

$$\lim_{t\to\infty} \left\| \nabla_\theta V^{\pi_\theta}(s)|_{\theta=\omega_t} \right\| = 0, \tag{111}$$

$$\lim_{t\to\infty} \left\| \nabla_\theta V^{\pi_\theta}(s)|_{\theta=\theta_t} \right\| = 0. \tag{112}$$

*Proof of Lemma 14.* Under APG, we have that for any $t \in \mathbb{N}$,

$$V^{\pi_\omega^{(t+1)}}(\mu) \geq V^{\pi_\theta^{(t+1)}}(\mu) \geq V^{\pi_\omega^{(t)}}(\mu) \geq V^{\pi_\theta^{(t)}}(\mu). \tag{113}$$

By (113) and the fact that $V^{\pi_\theta}(\mu) \leq \frac{1}{1-\gamma}$ for all $\theta$, we know that both the limits of $V^{\pi_\omega^{(t)}}(\mu)$ and $V^{\pi_\theta^{(t)}}(\mu)$ shall exist and $\lim_{t\to\infty} V^{\pi_\omega^{(t)}}(\mu) = \lim_{t\to\infty} V^{\pi_\theta^{(t)}}(\mu)$. As a result, we have

$$\lim_{t\to\infty} \left\| \nabla_\theta V^{\pi_\theta}(\mu)|_{\theta=\omega_t} \right\| = 0. \tag{114}$$

By (114), the update rule (7) of APG, and the Lipschitz continuity of $\nabla_\theta V^{\pi_\theta}(\mu)$ (Agarwal et al., 2021, Lemma D.4), we also have

$$\lim_{t\to\infty} \left\| \nabla_\theta V^{\pi_\theta}(\mu)|_{\theta=\theta_t} \right\| = 0. \tag{115}$$

By the expression of softmax policy gradient, (114) and (115) imply that for all $(s,a)$,

$$\pi_\omega^{(t)} A^{\pi_\omega^{(t)}}(s,a) \to 0, \quad \text{as } t \to \infty, \tag{116}$$

$$\pi_\theta^{(t)} A^{\pi_\theta^{(t)}}(s,a) \to 0, \quad \text{as } t \to \infty. \tag{117}$$

Therefore, we have that for every state $s \in \mathcal{S}$,

$$\lim_{t\to\infty} \left\| \nabla_\theta V^{\pi_\theta}(s)|_{\theta=\omega_t} \right\| = 0. \tag{118}$$

$$\lim_{t\to\infty} \left\| \nabla_\theta V^{\pi_\theta}(s)|_{\theta=\theta_t} \right\| = 0. \tag{119}$$

$\square$

**Lemma 15.** *Under APG, the limits $\lim_{t\to\infty} V^{\pi_\theta^{(t)}}(s)$, $\lim_{t\to\infty} Q^{\pi_\theta^{(t)}}(s,a)$, and $\lim_{t\to\infty} A^{\pi_\theta^{(t)}}(s,a)$ all exist, for all state $s \in \mathcal{S}$.*

*Proof of Lemma 15.* Recall that we use $\boldsymbol{V}^\pi$ to denote the $|\mathcal{S}|$-dimensional vector of the $V^\pi(s)$ of all the states $s \in \mathcal{S}$. Note that Lemma 14 tells us that under APG, the sequence of policies would converge to either one or multiple stationary points with the same $V^\pi(\mu)$. In other words, if we consider the trajectory of $\boldsymbol{V}^{\pi^{(t)}}$, there could possibly be multiple accumulation points, which all have the same $V^\pi(\mu)$. With that said, our goal is to show that the limit of the value vector $\boldsymbol{V}^{\pi^{(t)}}$ also exists. To achieve this, we first divide all the policies into multiple categories by defining equivalence classes as follows: For each real vector $\boldsymbol{V} \in \mathbb{R}^{|\mathcal{S}|}$,

$$\mathcal{C}(\boldsymbol{V}) := \left\{ \pi_\theta : \nabla_\theta V^{\pi_\theta}(\mu) = 0 \text{ and } \boldsymbol{V}^{\pi_\theta} = \boldsymbol{V} \right\}. \tag{120}$$

***Claim (a).*** *There are at most $|\mathcal{S}||\mathcal{A}|$ non-empty equivalence classes.*
To prove Claim (a), let us first fix some non-empty $\mathcal{C}(\boldsymbol{V})$. For any stochastic policy in $\mathcal{C}(\boldsymbol{V})$, one can verify that there must be a corresponding deterministic policy in $\mathcal{C}(\boldsymbol{V})$ by using the stationarity condition in (120) and the performance difference lemma. Hence, every non-empty $\mathcal{C}(\boldsymbol{V})$ must consist of at least one deterministic policy. Moreover, since there are only $|\mathcal{S}||\mathcal{A}|$ deterministic policies, there shall at most be $|\mathcal{S}||\mathcal{A}|$ non-empty equivalence classes.
Therefore, by Claim (a) and Assumption 4, we know that any two non-empty equivalence classes $\mathcal{C}(\boldsymbol{V})$ shall correspond to different $\langle \mu, \boldsymbol{V} \rangle$. Hence, by Lemma 14, we know the limit of the value vector $\boldsymbol{V}^{\pi^{(t)}}$ also exists, i.e., $\lim_{t\to} V^{\pi^{(t)}}(s)$ exists, for every state $s \in \mathcal{S}$. Finally, by the Bellman equation, we know $\lim_{t\to} Q^{\pi^{(t)}}(s)$ exists and therefore $\lim_{t\to} Q^{\pi^{(t)}}(s)$ exists, for all $s \in \mathcal{S}$. $\square$

In the sequel, we use $A^{(t)}(s,a)$, $Q^{(t)}(s,a)$, and $V^{(t)}(s)$ as the shorthand of $A^{\pi_\omega^{(t)}}(s,a)$, $Q^{\pi_\omega^{(t)}}(s,a)$, and $V^{\pi_\omega^{(t)}}(s)$, respectively. For ease of exposition, we divide the action space into the following subsets based on the advantage function:

$$I_s^+ := \{a \in \mathcal{A} : \lim_{t\to\infty} A^{(t)}(s,a) > 0\} \tag{121}$$

$$I_s^- := \{a \in \mathcal{A} : \lim_{t\to\infty} A^{(t)}(s,a) < 0\} \tag{122}$$

$$I_s^0 := \{a \in \mathcal{A} : \lim_{t\to\infty} A^{(t)}(s,a) = 0\} \tag{123}$$

Note that the above action sets are well-defined as the limiting value functions exist by Lemma 15. Moreover, we would like to highlight that the theoretical results in Appendix C.1 and C.2 are directly applicable to the general MDP case as long as the limiting value functions exist.

For ease of notation, for each state $s$, we define

$$\Delta_s := \min_{a \in I_s^+ \cup I_s^-} |A^{(t)}(s,a)|. \tag{124}$$

Accordingly, we know that for each state $s \in \mathcal{S}$, there must exist some $\bar{T}_s$ such that the following hold :

- (i) For all $a \in I_s^+$, we have

$$A^{(t)}(s,a) \geq +\frac{\Delta_s}{4}, \quad \text{for all } t \geq \bar{T}_s, \tag{125}$$

- (ii) For all $a \in I_s^-$, we have

$$A^{(t)}(s,a) \leq -\frac{\Delta_s}{4}, \quad \text{for all } t \geq \bar{T}_s. \tag{126}$$

- (iii) For all $a \in I_s^0$, we have

$$|A^{(t)}(s,a)| \leq \frac{\Delta_s}{4}, \quad \text{for all } t \geq \bar{T}_s. \tag{127}$$

**Lemma 16.** *For any state $s \in \mathcal{S}$, we have $\sum_{a \in I_s^+ \cup I_s^-} \pi_\theta^{(t)}(a|s) \to 0$, as $t \to \infty$. As a result, we also have $\sum_{a \in I_s^0} \pi_\theta^{(t)}(a|s) \to 1$, as $t \to \infty$.*

*Proof of Lemma 16.* Given that the limiting value functions exist as well as the fact that $d_\mu^{\pi_\theta}(s) \geq \frac{\mu(s)}{1-\gamma} > 0$, we know that for any state-action pair $(s,a)$,

$$\pi_\theta^{(t)}(a|s) A^{\pi_\theta^{(t)}}(s,a) \to 0, \quad \text{as } t \to \infty. \tag{128}$$

- For any $a \in I_s^+$, by definition we have $\lim_{t\to\infty} A^{\pi_\theta^{(t)}}(s,a) > 0$. By (128), this implies that $\pi_\theta^{(t)}(a|s) \to 0$, as $t \to \infty$.

- Similarly, for any $a \in I_s^-$, by definition we have $\lim_{t\to\infty} A^{\pi_\theta^{(t)}}(s,a) < 0$. Again, by (128), this property implies that $\pi_\theta^{(t)}(a|s) \to 0$, as $t \to \infty$.

Hence, we have $\sum_{a \in I_s^+ \cup I_s^-} \pi_\theta^{(t)}(a|s) \to 0$, as $t \to \infty$. $\qquad\square$

**Lemma 17.** *Let $a$ be an action in $I_s^+$. Under APG, $\theta_{s,a}^{(t)}$ and $\omega_{s,a}^{(t)}$ must be bounded from below, for all $t$.*

*Proof of Lemma 17.* Recall that we define $\Delta_s := \min_{a \in I_s^+ \cup I_s^-} |A^{\pi_\omega^{(t)}}(s,a)|$. Then, there must exist $\bar{T}_s \in \mathbb{N}$ such that $A^{\pi_\omega^{(t)}}(s,a) \geq \frac{\Delta_s}{4}$, for all $t \geq \bar{T}_s$.

For ease of notation, we let $\delta_{\bar{T}_s} := \theta_{s,a}^{(\bar{T}_s)} - \theta_{s,a}^{(\bar{T}_s-1)}$. Regarding the case $\delta_{\bar{T}_s} \geq 0$, the result directly holds since $A^{\pi_\omega^{(t)}}(s,a) \geq 0$, for all $t \geq \bar{T}_s$. Considering $\delta_{\bar{T}_s} < 0$, by a similar argument, for any $M \in \mathbb{N}$, we have

$$\theta_{s,a}^{(\bar{T}_s+M)} = \omega_{s,a}^{(\bar{T}_s+M-1)} + \eta^{(\bar{T}_s+M)} \cdot \underbrace{\frac{\partial V^{\pi_\theta}(\mu)}{\partial \theta_{s,a}}\Big|_{\theta=\omega^{(\bar{T}_s+M-1)}}}_{\geq 0} \tag{129}$$

$$\geq \theta_{s,a}^{(\bar{T}_s+M-1)} \tag{130}$$

$$+ \mathbb{I}\left\{V^{\pi_\varphi^{(\bar{T}_s+M-1)}}(\mu) \geq V^{\pi_\theta^{(\bar{T}_s+M-1)}}(\mu)\right\} \cdot \frac{\bar{T}_s+M-2}{\bar{T}_s+M+1}(\theta_{s,a}^{(\bar{T}_s+M-1)} - \theta_{s,a}^{(\bar{T}_s+M-2)}) \tag{131}$$

$$\geq \theta_{s,a}^{(\bar{T}_s)} + \frac{\bar{T}_s-1}{\bar{T}_s+2}\delta_{\bar{T}_s} + \frac{\bar{T}_s(\bar{T}_s-1)}{(\bar{T}_s+3)(\bar{T}_s+2)}\delta_{\bar{T}_s} + \cdots + \frac{(\bar{T}_s+M-2)\cdots(\bar{T}_s-1)}{(\bar{T}_s+M+1)\cdots(\bar{T}_s+2)}\delta_{\bar{T}_s} \tag{132}$$

$$= \theta_{s,a}^{(\bar{T}_s)} + \left[\frac{\bar{T}_s-1}{\bar{T}_s+2} + \frac{\bar{T}_s(\bar{T}_s-1)}{(\bar{T}_s+3)(\bar{T}_s+2)} + \sum_{\tau=2}^{M}\frac{(\bar{T}_s+1)\bar{T}_s(\bar{T}_s-1)}{(\bar{T}_s+\tau+2)(\bar{T}_s+\tau+1)(\bar{T}_s+\tau)}\right]\delta_{\bar{T}_s}. \tag{133}$$

Note that for any $M \in \mathbb{N}$,

$$\sum_{\tau=2}^{M}\frac{(\bar{T}_s+1)\bar{T}_s(\bar{T}_s-1)}{(\bar{T}_s+\tau+2)(\bar{T}_s+\tau+1)(\bar{T}_s+\tau)} \tag{134}$$

$$= (\bar{T}_s+1)\bar{T}_s(\bar{T}_s-1)\sum_{\tau=2}^{M}\frac{1}{2}\left(\frac{1}{(\bar{T}_s+\tau)(\bar{T}_s+\tau+1)} - \frac{1}{(\bar{T}_s+\tau+1)(\bar{T}_s+\tau+2)}\right) \tag{135}$$

$$= (\bar{T}_s+1)\bar{T}_s(\bar{T}_s-1)\cdot\frac{1}{2}\left(\frac{1}{(\bar{T}_s+2)(\bar{T}_s+3)} - \frac{1}{(\bar{T}_s+M+1)(\bar{T}_s+M+2)}\right) \tag{136}$$

$$\leq \frac{\bar{T}_s}{2}. \tag{137}$$

Therefore, we know that for any $M \in \mathbb{N}$,

$$\theta_{s,a}^{(\bar{T}_s+M)} \geq \theta_{s,a}^{(\bar{T}_s)} - (2 + \frac{\bar{T}_s}{2})|\delta_{\bar{T}_s}|. \tag{138}$$

Hence, $\theta_{s,a}^{(t)} \geq \theta_{s,a}^{(\bar{T}_s)} - (2 + \frac{\bar{T}_s}{2})|\delta_{\bar{T}_s}|$, for all $t \geq \bar{T}_s$. As the gradient under softmax parameterization is always bounded, this also implies that $\omega_{s,a}^{(t)}$ is bounded from below, for all $t$. $\qquad\square$

**Lemma 18.** *Let $a$ be an action in $I_s^-$. Under APG, $\theta_{s,a}^{(t)}$ and $\omega_{s,a}^{(t)}$ must be bounded from above, for all $t$.*

*Proof of Lemma 18.* To prove this, we could follow the same procedure as that in Lemma 17. Again, for ease of notation, we define $\Delta_s := \min_{a \in I_s^+ \cup I_s^-}|A^{\pi_\omega^{(t)}}(s,a)|$ and define $\delta_{\bar{T}_s} := \theta_{s,a}^{(\bar{T}_s)} - \theta_{s,a}^{(\bar{T}_s-1)}$. Accordingly, there must exist $\bar{T}_s \in \mathbb{N}$ such that $A^{\pi_\omega^{(t)}}(s,a) \leq -\frac{\Delta_s}{4}$, for all $t \geq \bar{T}_s$. Moreover, by the update scheme of APG, we have

$$\theta_{s,a}^{(\bar{T}_s+1)} = \omega_{s,a}^{(\bar{T}_s)} + \eta^{(\bar{T}_s+1)} \cdot \underbrace{\frac{\partial V^{\pi_\theta}(\mu)}{\partial \theta_{s,a}}\Big|_{\theta=\omega^{(\bar{T}_s)}}}_{\leq 0} \leq \omega_{s,a}^{(\bar{T}_s)}. \tag{139}$$

Similarly, regarding the case $\delta_{\bar{T}_s} \leq 0$, the result directly holds since $A^{\pi_\omega^{(t)}}(s,a) \leq 0$, for all $t \geq \bar{T}_s$. Considering $\delta_{\bar{T}_s} > 0$, for any $M \in \mathbb{N}$, we have

$$\theta_{s,a}^{(\bar{T}_s+M)} = \omega_{s,a}^{(\bar{T}_s+M-1)} + \eta^{(\bar{T}_s+M)} \cdot \underbrace{\frac{\partial V^{\pi_\theta}(\mu)}{\partial \theta_{s,a}}\Big|_{\theta=\omega^{(\bar{T}_s+M-1)}}}_{\leq 0} \tag{140}$$

$$\leq \theta_{s,a}^{(\bar{T}_s+M-1)} \tag{141}$$

$$+ \mathbb{I}\left\{V^{\pi_\varphi^{(T_0+M-1)}}(\mu) \geq V^{\pi_\theta^{(T_0+M-1)}}(\mu)\right\} \cdot \frac{\bar{T}_s+M-2}{\bar{T}_s+M+1}(\theta_{s,a}^{(\bar{T}_s+M-1)} - \theta_{s,a}^{(\bar{T}_s+M-2)}) \tag{142}$$

$$\leq \theta_{s,a}^{(\bar{T}_s)} + \frac{\bar{T}_s-1}{\bar{T}_s+2}\delta_{\bar{T}_s} + \frac{\bar{T}_s(\bar{T}_s-1)}{(\bar{T}_s+3)(\bar{T}_s+2)}\delta_{\bar{T}_s} + \cdots + \frac{(\bar{T}_s+M-2)\cdots(\bar{T}_s-1)}{(\bar{T}_s+M+1)\cdots(\bar{T}_s+2)}\delta_{\bar{T}_s} \tag{143}$$

$$= \theta_{s,a}^{(\bar{T}_s)} + \left[\frac{\bar{T}_s-1}{\bar{T}_s+2} + \frac{\bar{T}_s(\bar{T}_s-1)}{(\bar{T}_s+3)(\bar{T}_s+2)} + \sum_{\tau=2}^{M}\frac{(\bar{T}_s+1)\bar{T}_s(\bar{T}_s-1)}{(\bar{T}_s+\tau+2)(\bar{T}_s+\tau+1)(\bar{T}_s+\tau)}\right]\delta_{\bar{T}_s}. \tag{144}$$

By (134)-(137), we know $\sum_{\tau=2}^{M}\frac{(\bar{T}_s+1)\bar{T}_s(\bar{T}_s-1)}{(\bar{T}_s+\tau+2)(\bar{T}_s+\tau+1)(\bar{T}_s+\tau)} \leq \frac{\bar{T}_s}{2}$. As a result, for any $M \in \mathbb{N}$,

$$\theta_{s,a}^{(\bar{T}_s+M)} \leq \theta_{s,a}^{(\bar{T}_s)} + (2+\frac{\bar{T}_s}{2})|\delta_{\bar{T}_s}|. \tag{145}$$

Hence, $\theta_{s,a}^{(t)} \leq \theta_{s,a}^{(\bar{T}_s)} + (2+\frac{\bar{T}_s}{2})|\delta_{\bar{T}_s}|$, for all $t \geq \bar{T}_s$. As the gradient under softmax parameterization is always bounded, this also implies that $\omega_{s,a}^{(t)}$ is bounded from above, for all $t$. $\qquad\square$

**Lemma 19.** *Under APG, if $I_s^+$ is non-empty, then we have $\max_{a \in I_s^0} \theta_{s,a}^{(t)} \to \infty$, as $t \to \infty$.*

*Proof.* By Lemma 16, we know $\sum_{a \in I_s^0} \pi_\theta^{(t)}(a|s) \to 1$, as $t \to \infty$. Moreover, by Lemma 17, we know $\theta_{s,a}^{(t)}$ is bounded from below, for all $a \in I_s^+$. Therefore, under the softmax policy parameterization, we must have $\max_{a \in I_s^0} \theta_{s,a}^{(t)} \to \infty$. $\qquad\square$

Recall from (126) that for all $a \in I_s^-$, we have $A^{(t)}(s,a) \leq -\frac{\Delta_s}{4}$ for all $t \geq \bar{T}_s$.

**Lemma 20.** *Under APG, if $I_s^+$ is non-empty, then for any $a \in I_s^-$, we have $\theta_{s,a}^{(t)} \to -\infty$, as $t \to \infty$.*

*Proof of Lemma 20.* We prove this contradiction. Motivated by the proof of Lemma C11 in (Agarwal et al., 2021), our proof here extends the argument to the case with the momentum by considering the cumulative effect of all the gradient terms on the policy parameter.

Specifically, given an action $a \in I_s^-$, suppose that there exists $\vartheta$ such that $\theta_{s,a}^{(t)} > \vartheta$, for all $t \geq \bar{T}_s$. Then, by Lemma 13 and Lemma 19, we know there must exist an action $a' \in \mathcal{A}$ such that $\liminf_{t\to\infty} \theta_{s,a'}^{(t)} = -\infty$. Let $\delta > 0$ be some positive scalar such that $\theta_{s,a'}^{(\bar{T}_s)} \geq \vartheta - \delta$. For each $t \geq \bar{T}_s$, define

$$\nu(t) := \max\{\tau : \theta_{s,a'}^{(\tau)} \geq \vartheta - \delta, \bar{T}_s \leq \tau \leq t\}, \tag{146}$$

which is essentially the latest iteration at which $\theta_{s,a'}^{(\tau)}$ crosses $\vartheta - \delta$ from the above. Moreover, we define an index set

$$\mathcal{J}^{(t)} := \left\{\tau : \frac{\partial V^{(\tau)}(\mu)}{\partial \theta_{s,a'}} < 0, \nu(t) < \tau < t\right\}. \tag{147}$$

Define the cumulative effect (up to iteration $t$) of the gradient terms from those iterations in $\mathcal{J}^{(t)}$ as

$$Z^{(t)} := \sum_{t' \in \mathcal{J}^{(t)}} \eta^{(t'+1)} \cdot \frac{\partial V^{(t')}(\mu)}{\partial \theta_{s,a'}} \cdot G(t',t), \tag{148}$$

where $G(t, t')$ is the function defined in (30). Note that if $\mathcal{J}^{(t)} = \varnothing$, we define $Z^{(t)} = 0$. Accordingly, we know that for any $t > \bar{T}_s$, we have

$$Z^{(t)} \leq \sum_{t' \in \mathcal{J}^{(t)}} \eta^{(t'+1)} \cdot \frac{\partial V^{(t')}(\mu)}{\partial \theta_{s,a'}} \cdot G(t', t) + \underbrace{\sum_{t':t' \notin \mathcal{J}^{(t)}, \nu(t) < t' < t} \eta^{(t'+1)} \cdot \frac{\partial V^{(t')}(\mu)}{\partial \theta_{s,a'}} \cdot G(t', t)}_{\geq 0, \text{by the definition of } \mathcal{J}^{(t)}}$$

(149)

$$+ \underbrace{\sum_{t' \leq \nu(t)} \eta^{(t'+1)} \cdot \Big( \frac{\partial V^{(t')}(\mu)}{\partial \theta_{s,a'}} + \frac{1}{(1-\gamma)^2} \Big) \cdot G(t', t)}_{\geq 0, \text{by the fact that } |\partial V^{(t')}(\mu)/\partial \theta_{s,a'}| \leq 1/(1-\gamma)^2}$$

(150)

$$\leq (\theta_{s,a'}^{(t)} - \theta_{s,a}^{(1)}) + \sum_{t' \leq \nu(t)} \eta^{(t'+1)} \frac{1}{(1-\gamma)^2} G(t', t),$$

(151)

where (151) holds by the update scheme of APG as in Algorithm 2. Note that as $\liminf_{t \to \infty} \theta_{s,a'}^{(t)} = -\infty$, then $\nu(t)$ must be finite, for all $t$. This also implies that $\sum_{t' \leq \nu(t)} \eta^{(t'+1)} \frac{1}{(1-\gamma)^2} G(t', t)$ is finite, for all $t$. Therefore, by taking the limit infimum on both sides of (151), we know

$$\liminf_{t \to \infty} Z^{(t)} = -\infty.$$

(152)

Now we are ready to quantify $\theta_{s,a}^{(t)}$ for the action $a \in I_s^-$. For all $t' \in \mathcal{J}^{(t)}$, we must have

$$\frac{|\partial V^{(t')}(\mu)/\partial \theta_{s,a}|}{|\partial V^{(t')}(\mu)/\partial \theta_{s,a'}|} = \left| \frac{\pi^{(t')}(a|s) A^{(t')(s,a)}}{\pi^{(t')}(a'|s) A^{(t')(s,a')}} \right| \geq \exp(\vartheta - \theta_{s,a'}^{(t')}) \cdot \frac{(1-\gamma)\Delta_s}{4} \geq \exp(\delta) \cdot \frac{(1-\gamma)\Delta_s}{4},$$

(153)

where the first inequality follows from that $|A^{(t')}(s,a)| \leq 1/(1-\gamma)$ and that $A^{(t')}(s,a) \leq -\Delta_s/4$, and the second equality holds by the definition of $\nu(t)$. For any $\mathcal{J}^{(t)} \neq \varnothing$, we have

$$\theta_{s,a}^{(t)} - \theta_{s,a}^{(1)} = \sum_{t':1 \leq t' < \bar{T}_s} \eta^{(t'+1)} \cdot \frac{\partial V^{(t')}(\mu)}{\partial \theta_{s,a}} \cdot G(t', t) + \sum_{t':t' \geq \bar{T}_s} \eta^{(t'+1)} \cdot \frac{\partial V^{(t')}(\mu)}{\partial \theta_{s,a}} \cdot G(t', t)$$

(154)

$$\leq \sum_{t':1 \leq t' < \bar{T}_s} \eta^{(t'+1)} \cdot \frac{\partial V^{(t')}(\mu)}{\partial \theta_{s,a}} \cdot G(t', t) + \sum_{t':t' \in \mathcal{J}^{(t)}} \eta^{(t'+1)} \cdot \frac{\partial V^{(t')}(\mu)}{\partial \theta_{s,a}} \cdot G(t', t)$$

(155)

$$\leq \underbrace{\sum_{t':1 \leq t' < \bar{T}_s} \eta^{(t'+1)} \cdot \frac{\partial V^{(t')}(\mu)}{\partial \theta_{s,a}} \cdot G(t', t)}_{< \infty \text{ and does not depend on } t} + \exp(\delta) \cdot \frac{(1-\gamma)\Delta_s}{4} \underbrace{\sum_{t':t' \in \mathcal{J}^{(t)}} \eta^{(t'+1)} \cdot \frac{\partial V^{(t')}(\mu)}{\partial \theta_{s,a'}} \cdot G(t', t)}_{\equiv Z^{(t)}},$$

(156)

where (155) holds by the fact that $A^{(t)}(s, a) < 0$ for all $t \geq \bar{T}_s$ and (156) is a direct result of (153). Therefore, by taking the limit infimum on both sides of (156), we have $\liminf_{t \to \infty} \theta_{s,a}^{(t)} = -\infty$, which leads to contradiction. $\qquad \square$

For ease of notation, we define $\Delta \theta_{s,a}^{(t)} := \theta_{s,a}^{(t)} - \theta_{s,a}^{(t-1)}$, for each state-action pair $s, a$ and $t \in \mathbb{N}$.

**Lemma 21.** *Consider any state $s$ with non-empty $I_s^+$. Let $a_+ \in I_s^+$ and $a \in I_s^0$. Suppose $\theta_{s,a_+}^{(\tau)} > \theta_{s,a}^{(\tau)}$ and $\Delta \theta_{s,a_+}^{(\tau)} > \Delta \theta_{s,a}^{(\tau)}$, then we also have $\theta_{s,a_+}^{(t)} > \theta_{s,a}^{(t)}$ and $\Delta \theta_{s,a_+}^{(t)} > \Delta \theta_{s,a}^{(t)}$, for all $t > \tau > \bar{T}_s$.*

*Proof of Lemma 21.* We prove this by induction. Suppose at some time $\tau > \bar{T}_s$, we have $\theta_{s,a_+}^{(\tau)} > \theta_{s,a}^{(\tau)}$ and $\Delta\theta_{s,a_+}^{(\tau)} > \Delta\theta_{s,a}^{(\tau)}$. Then, we have

$$\omega_{s,a_+}^{(\tau)} = \theta_{s,a_+}^{(\tau)} + \mathbb{I}\left\{V^{\pi_\varphi^{(\tau)}}(\mu) \geq V^{\pi_\theta^{(\tau)}}(\mu)\right\} \cdot \frac{\tau-1}{\tau+2}(\theta_{s,a_+}^{(\tau)} - \theta_{s,a_+}^{(\tau-1)}) \tag{157}$$

$$> \theta_{s,a}^{(\tau)} + \mathbb{I}\left\{V^{\pi_\varphi^{(\tau)}}(\mu) \geq V^{\pi_\theta^{(\tau)}}(\mu)\right\} \cdot \frac{\tau-1}{\tau+2}(\theta_{s,a}^{(\tau)} - \theta_{s,a}^{(\tau-1)}) = \omega_{s,a}^{(\tau)} \tag{158}$$

Recall that we use $A^{(t)}(s,a)$, $Q^{(t)}(s,a)$ and $V^{(t)}(s)$ as the shorthand of $A^{\pi_\omega^{(t)}}(s,a)$, $Q^{\pi_\omega^{(t)}}(s,a)$ and $V^{\pi_\omega^{(t)}}(s)$, respectively. Note that

$$\frac{\partial V^{(t)}(s)}{\partial \theta_{s,a_+}} = \frac{1}{1-\gamma} \cdot d_\mu^{\pi_\omega^{(t)}}(s) \cdot \pi_\omega^{(t)}(a_+|s) \cdot (Q^{(t)}(s,a_+) - V^{(t)}(s)) \tag{159}$$

$$> \frac{1}{1-\gamma} \cdot d_\mu^{\pi_\omega^{(t)}}(s) \cdot \pi_\omega^{(t)}(a|s) \cdot (Q^{(t)}(s,a) - V^{(t)}(s)) \tag{160}$$

$$= \frac{\partial V^{(t)}(s)}{\partial \theta_{s,a}}, \tag{161}$$

where (160) holds by (158) and the fact that $\tau > \bar{T}_s$ implies $A^{(t)}(s,a_+) \geq A^{(t)}(s,a)$. Therefore, by (158)-(161), we must have

$$\theta_{s,a_+}^{(\tau+1)} = \omega_{s,a_+}^{(\tau)} + \eta^{(t+1)}\frac{\partial V^{(t)}(s)}{\partial \theta_{s,a_+}} > \omega_{s,a}^{(\tau)} + \eta^{(t+1)}\frac{\partial V^{(t)}(s)}{\partial \theta_{s,a}} = \theta_{s,a}^{(\tau+1)}, \tag{162}$$

$$\Delta\theta_{s,a_+}^{(\tau+1)} = \mathbb{I}\left\{V^{\pi_\varphi^{(\tau)}}(\mu) \geq V^{\pi_\theta^{(\tau)}}(\mu)\right\} \cdot \frac{\tau-1}{\tau+2}(\theta_{s,a_+}^{(\tau)} - \theta_{s,a_+}^{(\tau-1)}) + \frac{\partial V^{(t)}(s)}{\partial \theta_{s,a_+}} \tag{163}$$

$$> \mathbb{I}\left\{V^{\pi_\varphi^{(\tau)}}(\mu) \geq V^{\pi_\theta^{(\tau)}}(\mu)\right\} \cdot \frac{\tau-1}{\tau+2}(\theta_{s,a}^{(\tau)} - \theta_{s,a}^{(\tau-1)}) + \frac{\partial V^{(t)}(s)}{\partial \theta_{s,a}} = \Delta\theta_{s,a}^{(\tau+1)}. \tag{164}$$

By repeating the above argument, we know $\theta_{s,a_+}^t > \theta_{s,a}^t$ and $\Delta\theta_{s,a_+}^t > \Delta\theta_{s,a}^t$, for all $t > \tau$. $\quad\square$

Next, we take a closer look at the actions in $I_s^0$. We further decompose $I_s^0$ into two subsets as follows: For any state with non-empty $I_s^+$, for any $a \in I_s^+$, we define

$$B_s^0(a_+) := \left\{a \in I_s^0 : \text{For any } t \geq \bar{T}_s, \text{either } \theta_{s,a}^{(t)} \geq \theta_{s,a_+}^{(t)} \text{ or } \Delta\theta_{s,a}^{(t)} \geq \Delta\theta_{s,a_+}^{(t)}\right\} \tag{165}$$

We use $\bar{B}_s^0(a_+)$ to denote the complement of $B_s^0(a_+)$. As a result, we could write $\bar{B}_s^0(a_+)$ as

$$\bar{B}_s^0(a_+) := \left\{a \in I_s^0 : \theta_{s,a}^{(t)} < \theta_{s,a_+}^{(t)} \text{ and } \Delta\theta_{s,a}^{(t)} < \Delta\theta_{s,a_+}^{(t)}, \text{for some } t \geq \bar{T}_s\right\}. \tag{166}$$

**Lemma 22.** *Under APG, if $I_s^+$ is not empty, then:*

*a) For any $a_+ \in I_s^+$, we have*

$$\sum_{a \in B_s^0(a_+)} \pi_\theta^{(t)}(a|s) \to 1, \quad as\ t \to \infty. \tag{167}$$

*b) For any $a_+ \in I_s^+$, we have*

$$\max_{a \in B_s^0(a_+)} \theta_{s,a}^{(t)} \to \infty, \quad as\ t \to \infty. \tag{168}$$

*c) For any $a_+ \in I_s^+$, we have*

$$\sum_{a \in B_s^0(a_+)} \theta_{s,a}^{(t)} \to \infty, \quad as\ t \to \infty. \tag{169}$$

*Proof of Lemma 22.* Regarding **(a)**, by the definition of $\bar{B}_s^0(a_+)$, for each $a \in \bar{B}_s^0(a_+)$, there must exist some $T' \geq \bar{T}_s$ such that $\theta_{s,a_+}^{(T')} > \theta_{s,a}^{(T')}$ and $\Delta\theta_{s,a_+}^{(T')} > \Delta\theta_{s,a}^{(T')}$. Then, by Lemma 21, we know

$$\theta_{s,a_+}^{(t)} > \theta_{s,a}^{(t)} \text{ and } \Delta\theta_{s,a_+}^{(t)} > \Delta\theta_{s,a}^{(t)}, \quad \text{for all } t \geq T'. \tag{170}$$

Moreover, by Lemma 16 and that $a_+ \in I_s^+$, we have $\pi^{(t)}(a_+|s) \to 0$ as $t \to \infty$. Based on (170), this shall further imply that $\pi^{(t)}(a_+|s) \to 0$ as $t \to \infty$, for any $a \in \bar{B}_s^0(a_+)$. Hence, we conclude that

$$\sum_{a \in B_s^0(a_+)} \pi^{(t)}(a|s) \to 1, \quad \text{as } t \to \infty. \tag{171}$$

Regarding **(b)**, based on the result in **(a)**, we could leverage exactly the same argument as that of Lemma 19 and obtain that $\max_{a \in B_s^0(a_+)} \theta_{s,a}^{(t)} \to \infty$, as $t \to \infty$.

Regarding **(c)**, let us consider any action $a \in \bar{B}_s^0(a_+)$. By the definition of $B_s^0(a_+)$, at each iteration $t \geq \bar{T}_s$, either $\theta_{s,a}^{(t)} \geq \theta_{s,a_+}^{(t)}$ or $\Delta\theta_{s,a}^{(t)} \geq \Delta\theta_{s,a_+}^{(t)}$ holds. As a result, by Lemma 17, we know that $\theta_{s,a}^{(t)}$ must also be bounded from below, for all $t$. Therefore, based on the result in **(b)**, we know $\sum_{a \in B_s^0(a_+)} \theta_{s,a}^{(t)} \to \infty$, as $t \to \infty$. $\qquad\square$

**Lemma 23.** *Under APG, for any $a_+ \in I_s^+$, the following two properties about $\bar{B}_s^0(a_+)$ shall hold:*

*(a) There must exist some $T_{a_+}$ such that for all $a \in \bar{B}_s^0(a_+)$,*

$$\pi^{(t)}(a_+|s) > \pi^{(t)}(a|s) \quad \text{for all } t > T_{a_+}. \tag{172}$$

*(b) There must exist some $T_{a_+}^\dagger$ such that for all $a \in \bar{B}_s^0(a_+)$,*

$$|A^{(t)}(s,a)| < \frac{\pi^{(t)}(a_+|s)}{\pi^{(t)}(a|s)} \cdot \frac{\Delta_s}{16|\mathcal{A}|}, \quad \text{for all } t > T_{a_+}^\dagger. \tag{173}$$

*Moreover, this also implies that*

$$\sum_{a \in \bar{B}_s^0(a_+)} \pi^{(t)}(a|s)|A^{(t)}(s,a)| < \pi^{(t)}(a_+|s) \cdot \frac{\Delta_s}{16}, \quad \text{for all } t > T_{a_+}^\dagger. \tag{174}$$

*Proof of Lemma 23.* Regarding **(a)**, for each $a \in \bar{B}_s^0(a_+)$, we define

$$u_a(a_+) := \inf\{\tau \geq \bar{T}_s : \theta_{s,a_+}^{(\tau)} > \theta_{s,a}^{(\tau)} \text{ and } \Delta\theta_{s,a_+}^{(\tau)} > \Delta\theta_{s,a}^{(\tau)}\}. \tag{175}$$

By the definition of $\bar{B}_s^0(a_+)$, we know the following two facts: (i) $u_a(a_+)$ is finite, for any $a \in \bar{B}_s^0(a_+)$. (ii) By Lemma 21, for all $t \geq u_a(a_+)$, we must have $\theta_{s,a_+}^{(t)} > \theta_{s,a}^{(t)}$ and $\Delta\theta_{s,a_+}^{(t)} > \Delta\theta_{s,a}^{(t)}$. Therefore, by choosing $T_{a_+} := \max_{a \in \bar{B}_s^0(a_+)} u_a(a_+)$, we must have $\pi^{(t)}(a_+|s) > \pi^{(t)}(a|s)$, for all $t > T_{a_+}$.

Regarding **(b)**, as $\frac{\pi^{(t)}(a_+|s)}{\pi^{(t)}(a|s)} > 1$ for all $t > T_{a_+}$ (this is a direct result of **(a)**), we know that for each $a \in \bar{B}_s^0(a_+) \subseteq I_s^0$, there must exist some finite $t_a' > T_{a_+}$ such that

$$|A^{(t)}(s,a)| < \frac{\pi^{(t)}(a_+|s)}{\pi^{(t)}(a|s)} \cdot \frac{\Delta_s}{16|\mathcal{A}|}, \quad \text{for all } t \geq t_a'. \tag{176}$$

As a result, by choosing $T_{a_+}^\dagger := \max_{a \in \bar{B}_s^0(a_+)} t_a'$, we conclude that (173)-(174) indeed hold. $\qquad\square$

**Lemma 24.** *If $I_s^+$ is non-empty, then for any $a_+ \in I_s^+$, there exists some finite $\tilde{T}_{a_+}$ such that*

$$\sum_{a \in I_s^-} \pi^{(t)}(a|s)A^{(t)}(s,a) > -\pi^{(t)}(a_+|s)\frac{\Delta_s}{16}, \quad \text{for all } t \geq \tilde{T}_{a_+}. \tag{177}$$

*Proof.* Let $a_+ \in I_s^+$ and $a_- \in I_s^-$. By Lemma 17 and Lemma 20, we know $\theta_{s,a_+}^{(t)}$ is always bounded from below and $\theta_{s,a_-}^{(t)} \to \infty$, as $t \to \infty$. This implies that $\pi^{(t)}(a_-|s)/\pi^{(t)}(a_+|s) \to 0$, as $t \to \infty$. Therefore, there must exist some finite $t'_{a_-}$ such that

$$\frac{\pi^{(t)}(a_-|s)}{\pi^{(t)}(a_+|s)} < \frac{\Delta_s(1-\gamma)}{16|\mathcal{A}|}, \quad \text{for all } t \geq t'_{a_-}. \tag{178}$$

By choosing $\tilde{T}_{a_+} := \max_{a_- \in I_s^-} t'_{a_-}$, we know (177) holds for all $t \geq \tilde{T}_{a_+}$. $\qquad \square$

C.2   PUTTING EVERYTHING TOGETHER: ASYMPTOTIC CONVERGENCE OF APG

Now we are ready to put everything together and prove Theorem 1. For ease of exposition, we restate Theorem 1 as follows.

**Theorem 1** (**Global convergence under softmax parameterization**). *Consider a tabular softmax parameterized policy $\pi_\theta$. Under APG with $\eta^{(t)} = \frac{t}{t+1} \cdot \frac{(1-\gamma)^3}{16}$, we have $V^{\pi_\theta^{(t)}}(s) \to V^*(s)$ as $t \to \infty$, for all $s \in \mathcal{S}$.*

*Proof of Theorem 1.* We prove this by contradiction. Suppose there exists at least one state $s \in \mathcal{S}$ with a non-empty $I_s^+$. Consider an action $a_+ \in I_s^+$. Recall the definitions of $\bar{T}_s$, $T_{a_+}, T_{a_+}^\dagger$, and $\tilde{T}_{a_+}$ from (125)-(127), Lemma 23, and Lemma 24, respectively. We define $T_{\max} := \max\{\bar{T}_s, T_{a_+}, T_{a_+}^\dagger, \tilde{T}_{a_+}\}$. Note that for all $t > T_{\max}$, we have

$$0 = \sum_{a \in B_s^0(a_+)} \pi^{(t)}(a|s)A^{(t)}(s,a) + \underbrace{\sum_{a \in \bar{B}_s^0(a_+)} \pi^{(t)}(a|s)A^{(t)}(s,a)}_{> -\pi^{(t)}(a_+|s)\frac{\Delta_s}{16} \text{ by Lemma 23}} \tag{179}$$

$$+ \underbrace{\sum_{a \in I_s^+} \pi^{(t)}(a|s)A^{(t)}(s,a)}_{\geq \pi^{(t)}(a_+|s)\frac{\Delta_s}{4}} + \underbrace{\sum_{a \in I_s^-} \pi^{(t)}(a|s)A^{(t)}(s,a)}_{> -\pi^{(t)}(a_+|s)\frac{\Delta_s}{16} \text{ by Lemma 24}} \tag{180}$$

$$> \sum_{a \in B_s^0(a_+)} \pi^{(t)}(a|s)A^{(t)}(s,a) + \frac{1}{8} \cdot \pi^{(t)}(a_+|s)\Delta_s \tag{181}$$

$$> \sum_{a \in B_s^0(a_+)} \pi^{(t)}(a|s)A^{(t)}(s,a). \tag{182}$$

Note that (182) implies $\sum_{a \in B_s^0(a_+)} \frac{\partial V^{(t)}(\mu)}{\partial \theta_{s,a}} < 0$, for all $t > T_{\max}$. Moreover, we have

$$\sum_{a \in B_s^0(a_+)} \theta_{s,a}^{(t)} - \theta_{s,a}^{(1)} \tag{183}$$

$$= \sum_{a \in B_s^0(a_+)} \sum_{t'=1}^{t} \eta^{(t'+1)} \cdot \frac{\partial V^{(t')}(\mu)}{\partial \theta_{s,a'}} \cdot G(t', t) \tag{184}$$

$$= \sum_{t'=1}^{t} \eta^{(t'+1)} G(t', t) \cdot \left( \sum_{a \in B_s^0(a_+)} \frac{\partial V^{(t')}(\mu)}{\partial \theta_{s,a'}} \right) \tag{185}$$

$$= \underbrace{\sum_{t'=1}^{T_{\max}} \eta^{(t'+1)} G(t', t) \cdot \left( \sum_{a \in B_s^0(a_+)} \frac{\partial V^{(t')}(\mu)}{\partial \theta_{s,a'}} \right)}_{< \infty \text{ and does not depend on } t} \tag{186}$$

$$+ \sum_{t'=T_{\max}+1}^{t} \eta^{(t'+1)} G(t', t) \cdot \underbrace{\left( \sum_{a \in B_s^0(a_+)} \frac{\partial V^{(t')}(\mu)}{\partial \theta_{s,a'}} \right)}_{< 0 \text{ by (182)}}. \tag{187}$$

By taking the limit of the both sides of (186)-(187), we know that the left-hand side of (186)-(187) shall go to positive infinity by Lemma 22, but the right-hand side of (186)-(187) is bounded from above. This leads to contradiction and hence completes the proof. □

# D    CONVERGENCE RATE OF APG: GENERAL MDPS

## D.1    SUPPORTING LEMMAS OF CONVERGENCE RATE OF APG

**Lemma 25.** *Under APG, there exists a finite time $T_V$ such that $V^{\pi_\theta^{(t)}}(s) > Q^*(s, a_2(s))$ for all $s \in \mathcal{S}$ and $t \geq T_V$.*

*Proof of Lemma 25.* By Theorem 1, $V^{\pi_\theta^{(t)}}(s) \to V^*(s)$ for all $s \in \mathcal{S}$. Hence, by the $\varepsilon - \delta$ argument, given $\varepsilon = V^*(s) - Q^*(s, a_2(s)) > 0$, we have a finite $T_V(s)$ such that $V^{\pi_\theta^{(t)}}(s) > Q^*(s, a_2(s))$ for any $t \geq T_V(s)$. Accordingly, let $T_V = \max_{s \in \mathcal{S}} T_V(s)$, we complete the proof.    $\square$

**Lemma 26.** $\left\| \frac{d_\mu^{\pi_{\theta'}}}{d_\mu^{\pi_\theta}} \right\|_\infty \leq C = 1 + \varepsilon$ *with* $\varepsilon > 0$*, if* $\theta_{s, a^*(s)} - \theta_{s, a} > M$*, where* $M = \ln \left[ \frac{(2\gamma |\mathcal{S}|(1+\varepsilon) - 1 + \gamma)(|\mathcal{A}| - 1)}{1 - \gamma} \right]$*, and* $\theta'_{s, a^*(s)} > \theta_{s, a^*(s)}$ *for all* $s \in \mathcal{S}$ *and* $a \neq a^*(s)$*.*

**Remark 14.** Please note that the condition $\theta'_{s, a^*(s)} > \theta_{s, a^*(s)}$ for all $s \in \mathcal{S}$, as demonstrated in Lemma 26, is indeed achievable. We can establish that the condition $\theta_{s, a^*(s)} - \theta_{s, a} > M$ holds true for all $s \in \mathcal{S}$ and $a \neq a^*(s)$ can be met through the updates of APG. Consequently, once this condition is met, all parameter values $\theta'$, obtained after updating $\theta$ along the directions in the feasible update domain $\mathcal{U}$, adhere to the conditions $\theta'_{s, a^*(s)} > \theta_{s, a^*(s)}$ for all $s \in \mathcal{S}$ and $a \neq a^*(s)$.

*Proof of Lemma 26.* We bound the infinity-norm ratio by characterizing $d_\mu^{\pi_\theta}(s)$ for a fixed $s \in \mathcal{S}$. Given $\theta$, we can derive a policy $\pi_\theta$. We assume that $\theta$ satisfies that $\theta_{s, a^*(s)} - \theta_{s, a} > M$ for all $s \in \mathcal{S}$ and $a \neq a^*(s)$. For the fixed $\pi_\theta$, we can reduce our MDP into an MRP with $(|\mathcal{S}| + 1)$ states. Among these, $|\mathcal{S}|$ states are inherited from the original MDP, and an additional terminal state is introduced and labeled as $T$. Moreover, for considering the state visitation distribution $d_\mu^{\pi_\theta}(s)$ for a fixed $s \in \mathcal{S}$, we can further reduce the $(|\mathcal{S}| + 1)$-state MRP into a 2-state MRP $\mathcal{M}^{\pi_\theta}$, by merging $|\mathcal{S}|$ original states into a single state denoted as $X$. This merged state has two transitions: one leading back to itself with a probability of $P_X^\theta$ and another transitioning to the terminal state $T$. For calculating $d_\mu^{\pi_\theta}(s)$, it is equivalent to calculate the discounted probability of visiting $X$ in this 2-state MRP. Specifically, to find the probability $P_X^\theta$, which represents the one-step transition in $\mathcal{M}^{\pi_\theta}$ from $X$ to itself, we need to identify and sum the probabilities of all disjoint paths that originate from $X$ and return to $X$. After careful derivations, we obtain that

$$P_X^\theta = \gamma P^{\pi_\theta}(s|s) + \sum_{s' \neq s} P^{\pi_\theta}(s'|s) \sum_{k=1}^\infty \gamma^{k+1} \mathrm{Pr}^{\pi_\theta}(s' \to s \text{ in exactly } k \text{ steps}), \qquad (188)$$

where $P^{\pi_\theta}(s|s) = \sum_{a \in \mathcal{A}} \pi_\theta(a|s) \cdot \mathcal{P}(s|s, a)$ and $P^{\pi_\theta}(s'|s) = \sum_{a \in \mathcal{A}} \pi_\theta(a|s) \cdot \mathcal{P}(s'|s, a)$. Given our focus on the state $s$, it's important to note that the paths from $s'$ to $s$ are not influenced by the policy $\pi_\theta$ at state $s$ since they do not pass through $s$. Consequently, only $P^{\pi_\theta}(s|s)$ and $P^{\pi_\theta}(s'|s)$ are affected by the policy $\pi_\theta$ in this context. In particular, to maximize the difference, we consider the limit as $\pi_{\theta'}(a^*(s)|s)$ approaches 1. In this scenario, the differences in $P^{\pi_\theta}(s|s)$ and $P^{\pi_\theta}(s'|s)$ can be at most $2(\pi_{\theta'}(a^*(s)|s) - \pi_\theta(a^*(s)|s)) \leq 2(1 - \pi_\theta(a^*(s)|s))$. Consequently, the total impact, denoted by $D^\theta$, on $P_X^\theta$ will be

$$D^\theta \leq 2\gamma |\mathcal{S}|(1 - \pi_\theta(a^*(s)|s)) \qquad (189)$$

Next, we calculate the discounted state visitation distribution in $\mathcal{M}^{\pi_\theta}$,

$$d_\mu^{\pi_\theta}(s) = (1 - \gamma) \left[ \mu(s) \frac{1}{1 - P_X^\theta} + \sum_{s' \neq s} \mu(s') \sum_{k=1}^\infty \gamma^k \mathrm{Pr}^{\pi_\theta}(s' \to s \text{ in exactly } k \text{ steps}) \frac{1}{1 - P_X^\theta} \right] \tag{190}$$

$$= \frac{1}{1 - P_X^\theta} \left[ (1 - \gamma) \left( \mu(s) + \sum_{s' \neq s} \mu(s') \sum_{k=1}^\infty \gamma^k \mathrm{Pr}^{\pi_\theta}(s' \to s \text{ in exactly } k \text{ steps}) \right) \right]. \tag{191}$$

Therefore, by (189), we can see that the value of $\frac{d_\mu^{\pi_{\theta'}}(s)}{d_\mu^{\pi_\theta}(s)}$ could be at most

$$\frac{1 - P_X^\theta}{1 - P_X^{\theta'}} \leq \frac{1 - P_X^\theta}{1 - P_X^\theta - D^\theta} = 1 + \frac{D^\theta}{1 - P_X^\theta - D^\theta}. \tag{192}$$

To establish an upper bound for $\frac{D^\theta}{1-P_X^\theta-D^\theta}$, we need to determine an upper bound for $P_X^\theta$. To do this, we consider the extreme case for every MRP. In (188), the latter term decays geometrically by $\gamma$ at each step for every state $s'$. Therefore, in the extreme case, we consider the scenario where there is a probability 1 transition for each disjoint path from state $s' \neq s$ to state $s$. Consequently, the upper bound can be expressed as

$$P_X^\theta \leq \gamma P^{\pi_\theta}(s|s) + \sum_{s' \neq s} P^{\pi_\theta}(s'|s)\gamma^2 \tag{193}$$

$$= \gamma P^{\pi_\theta}(s|s) + \gamma^2(1 - P^{\pi_\theta}(s|s)) \tag{194}$$

$$\leq \gamma \tag{195}$$

Hence, we have the upper bound for (192). Since $\theta_{s,a^*(s)} - \theta_{s,a} > M$ for all $s \in \mathcal{S}$ and $a \neq a^*(s)$ with $M = \ln\left[\frac{(2\gamma|\mathcal{S}|(1+\varepsilon)-1+\gamma)(|\mathcal{A}|-1)}{1-\gamma}\right]$, we have that

$$\frac{D^\theta}{1 - P_X^\theta - D^\theta} \leq \varepsilon. \tag{196}$$

Since the statement holds for all $s \in \mathcal{S}$, the maximum will be the value we found above. Hence, $\left\|\frac{d_\mu^{\pi_{\theta'}}}{d_\mu^{\pi_\theta}}\right\|_\infty \leq C = 1 + \varepsilon.$ $\qquad\square$

Recall that according to Definition 2, $\mathcal{U}$ is a set consisting of $\mathbb{R}^{|\mathcal{S}| \times |\mathcal{A}|}$ vectors. However, we can simplify the analysis by fixing a specific state, denoted as $s$, and separately considering the updates at each state. Therefore, without loss of generality, we will use $d_1$ to represent $d(s, a^*(s))$, $d_2$ for $d(s, a_2(s))$, and so on.

**Lemma 27.** *Let $\theta$ be the parameter satisfying that, for all $s \in \mathcal{S}$ and $a \neq a^*(s)$, $\theta_{s,a^*(s)} - \theta_{s,a} > M$. We consider any unit vector direction $\boldsymbol{d}$ in the feasible update domain $\mathcal{U}$. Let $M = \ln[3\binom{|\mathcal{A}|-1}{2}] - 2\ln[d_1 - d_i] > 0$ be the maximum value over all $i > 1$.*

- *The function $\theta \to \pi_\theta(a^*(s)|s)$ are concave for all $s \in \mathcal{S}$ along the direction $\boldsymbol{d}$.*

- *The function $\theta \to \pi_\theta(a|s)$ are convex for all $s \in \mathcal{S}$ and $a \neq a^*(s)$ along the direction $\boldsymbol{d}$.*

*Proof of Lemma 27.* For any fixed $s \in \mathcal{S}$, We establish the convexity of the function $\theta \to \pi_\theta(a|s)$ for all $a \neq a^*(s)$ by demonstrating that if $\theta_{s,a^*(s)} - \theta_{s,a} > M$ for all $a \neq a^*(s)$, then the function is convex. Following that, since $\pi_\theta(a^*(s)|s) = 1 - \sum_{a \neq a^*} \pi_\theta(a|s)$ can be viewed as a summation of concave functions, it follows that $\theta \to \pi_\theta(a^*(s)|s)$ is concave.

Given an action $a_i(s) \neq a^*(s)$, since the convexity is determined by the behavior of a function on arbitrary line on its domain, it is sufficient to show that the following function is concave (i.e. the second derivative is non-positive) when $k \to 0$,

$$f(k) = \frac{e^{\theta_{s,a_i(s)}+k\cdot d_i}}{e^{\theta_{s,a^*(s)}+k\cdot d_1} + e^{\theta_{s,a_2(s)}+k\cdot d_2} + \cdots + e^{\theta_{s,a_{|\mathcal{A}|}(s)}+k\cdot d_{|\mathcal{A}|}}} \tag{197}$$

$$= \frac{1}{e^{(\theta_{s,a^*(s)}-\theta_{s,a_i(s)})+k\cdot(d_1-d_i)} + e^{(\theta_{s,a_2(s)}-\theta_{s,a_i(s)})+k\cdot(d_2-d_i)} + \cdots + e^{(\theta_{s,a_{|\mathcal{A}|}(s)}-\theta_{s,a_i(s)})+k\cdot(d_{|\mathcal{A}|}-d_i)}} \tag{198}$$

$$:= \frac{1}{m(k)}, \tag{199}$$

where $\boldsymbol{d} = [d_1, d_2, \cdots, d_{|\mathcal{A}|}]$ is any unit vector on the feasible update domain $\mathcal{U}$.

By taking the second derivative of $f(k)$, we have

$$f''(k) = \frac{2(m'(k))^2}{m(k)^3} - \frac{m''(k)}{m(k)^2}. \tag{200}$$

And so, we have the second derivative of $f(k)$ when $k \to 0$ is

$$f''(0) = \frac{1}{m(0)^2} \left( \frac{2(m'(0))^2}{m(0)} - m''(0) \right). \tag{201}$$

Note that since $m(k) \geq 0$ for all $k$, we have that $f''(0) > 0$ (convex) if and only if:

$$2(m'(0))^2 - m''(0) \cdot m(0) > 0, \tag{202}$$

where $m'(0) = \sum_{j \neq i} (d_j - d_i) \exp(\theta_{s,a_j(s)} - \theta_{s,a_i(s)})$ and $m''(0) = \sum_{j \neq i} (d_j - d_i)^2 \exp(\theta_{s,a_j(s)} - \theta_{s,a_i(s)})$. By plugging $m(0), m'(0), m''(0)$ into (202) we have

$$2(m'(0))^2 - m''(0) \cdot m(0) \tag{203}$$

$$= 2 \sum_{j \neq i} (d_j - d_i)^2 \exp(2\theta_{s,a_j(s)} - 2\theta_{s,a_i(s)}) \tag{204}$$

$$+ 2 \sum_{u,v \neq i, u < v} 2(d_u - d_i)(d_v - d_i) \exp(\theta_{s,a_u(s)} + \theta_{s,a_v(s)} - 2\theta_{s,a_i(s)}) \tag{205}$$

$$- \sum_{j \neq i} (d_j - d_i)^2 \exp(2\theta_{s,a_j(s)} - 2\theta_{s,a_i(s)}) \tag{206}$$

$$- \sum_{u,v \neq i, u < v} \left( (d_u - d_i)^2 + (d_v - d_i)^2 \right) \exp(\theta_{s,a_u(s)} + \theta_{s,a_v(s)} - 2\theta_{s.a_i(s)}) \tag{207}$$

$$- \sum_{j \neq i} (d_j - d_i)^2 \exp(\theta_{s,a_j(s)} - \theta_{s,a_i(s)}) \tag{208}$$

$$= \sum_{j \neq i} (d_j - d_i)^2 \left( \exp(2\theta_{s,a_j(s)} - 2\theta_{s,a_i(s)}) - \exp(\theta_{s,a_j(s)} - \theta_{s,a_i(s)}) \right) \tag{209}$$

$$+ \sum_{u,v \neq i, u < v} 2(d_u - d_i)(d_v - d_i) \exp(\theta_{s,a_u(s)} + \theta_{s,a_v(s)} - 2\theta_{s,a_i(s)}) \tag{210}$$

$$- \sum_{u,v \neq i, u < v} (d_u^2 + d_v^2) \exp(\theta_{s,a_u(s)} + \theta_{s,a_v(s)} - 2\theta_{s,a_i(s)}) \tag{211}$$

$$= (d_1 - d_i)^2 \left( \exp(2\theta_{s,a^*(s)} - 2\theta_{s,a_i(s)}) - \exp(\theta_{s,a^*(s)} - \theta_{s,a_i(s)}) \right) \tag{212}$$

$$+ \sum_{u,v \neq i, u < v} 2(d_u - d_i)(d_v - d_i) \exp(\theta_{s,a_u(s)} + \theta_{s,a_v(s)} - 2\theta_{s,a_i(s)}) \tag{213}$$

$$- \sum_{u,v \neq i, u < v} (d_u^2 + d_v^2) \exp(\theta_{s,a_u(s)} + \theta_{s,a_v(s)} - 2\theta_{s,a_i(s)}) \tag{214}$$

$$\geq (d_1 - d_i)^2 \left( \exp(2\theta_{s,a^*(s)} - 2\theta_{s,a_i(s)}) - \exp(\theta_{s,a^*(s)} - \theta_{s,a_i(s)}) \right) \tag{215}$$

$$- \sum_{u,v \neq i, u < v} 2 \exp(\theta_{s,a_u(s)} + \theta_{s,a_v(s)} - 2\theta_{s,a_i(s)}) \tag{216}$$

$$- \sum_{u,v \neq i, u < v} \exp(\theta_{s,a_u(s)} + \theta_{s,a_v(s)} - 2\theta_{s,a_i(s)}) \tag{217}$$

$$\geq (d_1 - d_i)^2 \exp(\theta_{s,a^*(s)} - \theta_{s,a_i(s)}) \left( \exp(\theta_{s,a^*(s)} - \theta_{s,a_i(s)}) - 1 \right) \tag{218}$$

$$- 3 \binom{|\mathcal{A}| - 1}{2} \max_{u,v \neq i, u \neq v} \exp(\theta_{s,a_u(s)} + \theta_{s,a_v(s)} - 2\theta_{s,a_i(s)}) \tag{219}$$

$$\geq (d_1 - d_i)^2 \exp(\theta_{s,a^*(s)} - \theta_{s,a_i(s)}) \left( \exp(\theta_{s,a^*(s)} - \theta_{s,a_i(s)}) - 1 \right) \tag{220}$$

$$-3\binom{|\mathcal{A}|-1}{2}\exp(2\theta_{s,a^*(s)}-2\theta_{s,a_i(s)}-M), \tag{221}$$

$$=\exp(\theta_{s,a^*(s)}-\theta_{s,a_i(s)})\Big[(d_1-d_i)^2(\exp(\theta_{s,a^*(s)}-\theta_{s,a_i(s)})-1)$$

$$-3\binom{|\mathcal{A}|-1}{2}\exp(\theta_{s,a^*(s)}-\theta_{s,a_i(s)}-M)\Big] \tag{222}$$

where (216) holds by considering the minimum of $(d_u-d_i)(d_v-d_i)\geq -1$, (217) is because $-(d_u^2+d_v^2)\geq -1$, (218) holds by taking $\exp(\theta_{s,a^*(s)}-\theta_{s,a_i(s)})$ our of the parenthesis, (221) is by using the hypothesis $\theta_{s,a^*(s)}-\theta_{s,a_i(s)}>M$ for all $a\neq a^*(s)$.

To show that (222) is greater than 0, since $\exp(\theta_{s,a^*(s)}-\theta_{s,a_i(s)})>0$, we need

$$(d_1-d_i)^2(\exp(\theta_{s,a^*(s)}-\theta_{s,a_i(s)})-1)>3\binom{|\mathcal{A}|-1}{2}\exp(\theta_{s,a^*(s)}-\theta_{s,a_i(s)}-M). \tag{223}$$

By rearranging the terms, we obtain

$$\exp(M)>\frac{3\binom{|\mathcal{A}|-1}{2}\exp(\theta_{s,a^*(s)}-\theta_{s,a_i(s)})}{(d_1-d_i)^2(\exp(\theta_{s,a^*(s)}-\theta_{s,a_i(s)})-1)} \tag{224}$$

$$=\frac{3\binom{|\mathcal{A}|-1}{2}\exp(\theta_{s,a^*(s)}-\theta_{s,a_i(s)}-1)+3\binom{|\mathcal{A}|-1}{2}}{(d_1-d_i)^2(\exp(\theta_{s,a^*(s)}-\theta_{s,a_i(s)})-1)} \tag{225}$$

$$=\frac{3\binom{|\mathcal{A}|-1}{2}}{(d_1-d_i)^2}+\frac{3\binom{|\mathcal{A}|-1}{2}}{(d_1-d_i)^2(\exp(\theta_{s,a^*(s)}-\theta_{s,a_i(s)})-1)} \tag{226}$$

$$>\frac{3\binom{|\mathcal{A}|-1}{2}}{(d_1-d_i)^2}. \tag{227}$$

Hence, if for all $s\in\mathcal{S}$ and $a\neq a^*(s)$, $\theta_{s,a^*(s)}-\theta_{s,a^*(s)}>\ln[3\binom{|\mathcal{A}|-1}{2}]-2\ln[d_1-d_i]$ for all $i>1$, then we obtain the desired results. $\qquad\square$

### D.2 Convergence Rate of APG

**Theorem 2 (Convergence Rate of APG; Formal).** Consider a tabular softmax parameterized policy $\pi_\theta$. Under APG with $\eta^{(t)}=\frac{t}{t+1}\cdot\frac{(1-\gamma)^3}{16}$, there exists a finite time $T$ such that for all $t\geq T$, we have

$$V^*(\rho)-V^{\pi_\theta^{(t)}}(\rho)\leq\frac{1}{1-\gamma}\left\|\frac{d_\rho^{\pi^*}}{\mu}\right\|_\infty\cdot\left(\frac{32(2+T)\left(\|\theta^{(T)}\|+2\ln(t-T)\right)^2+K_T}{(1-\gamma)^3(t+T+1)(t+T)}\right. \tag{228}$$

$$\left.+\frac{4|\mathcal{S}|}{(1-\gamma)^2}\frac{|\mathcal{A}|-1}{(t-T)^2+|\mathcal{A}|-1}\right), \tag{229}$$

where $K_T=2(2+T)(1+T)\left|\left\langle\nabla_\theta V^{\pi_\theta}(\mu)\Big|_{\theta=\theta_\omega^{(T)}},m^{(T)}\right\rangle\right|$ and $m^{(T)}$ is the initial momentum while entering the local $\frac{3}{2}$-nearly concave regime.

*Proof of Theorem 2.* By Lemma 1, we know that the objective function $\theta\to V^{\pi_\theta}(\mu)$ is $\frac{3}{2}$-nearly concave in the directions along the feasible update domain $\mathcal{U}$ when, for all $s\in\mathcal{S}$ and $a\neq a^*(s)$, $V^{\pi_\theta}(s)>Q^*(s,a_2(s))$, $\theta_{s,a^*(s)}-\theta_{s,a}>M=\max\left\{\ln\left[3\binom{|\mathcal{A}|-1}{2}\right]-2\ln[d_1-d_i],\ln\left[\frac{(3\gamma|\mathcal{S}|-1+\gamma)(|\mathcal{A}|-1)}{1-\gamma}\right]\right\}$. We have to show the following two statements:

- There is a lower bound for the value $d_1-d_i$ of the update directions after a finite time $T$ such that $M$ is a well-defined finite constant.

- There is a finite time $T$ such that for every update after time $T$, its direction is along the vectors in the feasible update domain $\mathcal{U}$.

To show the first point, by Lemma 2, we know that there is a finite time $T$ such that $\frac{\partial V^{\pi_\theta}(\mu)}{\partial \theta_{s,a^*(s)}}\Big|_{\theta=\omega^{(t)}} \geq 0 \geq \frac{\partial V^{\pi_\theta}(\mu)}{\partial \theta_{s,a}}\Big|_{\theta=\omega^{(t)}}$ and $\omega_{s,a^*(s)}^{(t)} - \theta_{s,a^*(s)}^{(t)} \geq \omega_{s,a}^{(t)} - \theta_{s,a}^{(t)}$ for all $s \in \mathcal{S}$ and $a \neq a^*(s)$. Due to the direction of gradient, i.e., $a^*(s)$ is positive and others are negative, and the order between the momentum of each action, for any $t > T$, the minimum difference between the updates of $a^*(s)$ and other actions $a$ will always be greater than the difference of the updates at step $T$. Hence, let $d_1 = \omega_{s,a^*(s)}^{(T)} - \theta_{s,a^*(s)}^{(T)} + \eta^{(T+1)} \cdot \frac{\partial V^{\pi_\theta}(\mu)}{\partial \theta_{s,a^*(s)}}\Big|_{\theta=\omega^{(T)}}$ and $d_a = \omega_{s,a}^{(T)} - \theta_{s,a}^{(T)} + \eta^{(T+1)} \cdot \frac{\partial V^{\pi_\theta}(\mu)}{\partial \theta_{s,a}}\Big|_{\theta=\omega^{(T)}}$, we can plugin $\min_{a\neq a^*(s)} d_1 - d_a$ to the definition of $M$, which makes $M$ well-defined and becoming a finite constant.

Next, we prove the second statement. Similar to the above argument, after time $T$, both the gradient and momentum of $a^*(s)$ are the greatest for all $s \in \mathcal{S}$. Thus, for the updates after $T$, $d_1$ are always the greatest, that is, the update direction $d$ belongs to $\mathcal{U}$.

Furthermore, it is evident that all the conditions outlined in Lemma 1 can be met by applying Lemma 2. Consequently, the APG algorithm achieves the local $C$-nearly concave region for all $t \geq T$, where $T$ is a finite value.

As the objective enters a locally $\frac{3}{2}$-nearly concave region after time $T$, it is necessary to account for a shift in the initial learning rate due to the passage of time $T$. In other words, if we divide the update process into two phases based on time $T$, the latter phase will commence with a modified learning rate. By Corollary 1, Lemma 9 with $\eta^{(t)} = \frac{t}{t+1}\frac{1}{2L}$, $c = T$, and $\theta^{**} = \theta^{(t)**}$ we have that

$$V^{\pi_{\theta^{**}}^{(t)}}(\mu) - V^{\pi_\theta^{(t)}}(\mu) \leq \frac{4L(2+T)\left\|\theta^{(T)} - \theta^{(t)**}\right\|^2 + 2(2+T)(1+T)\left|\left\langle \nabla_\theta V^{\pi_\theta}(\mu)\Big|_{\theta=\theta_{md}^{(T)}}, m^{(T)}\right\rangle\right|}{(t+T+1)(t+T)}$$

$$(230)$$

$$= \frac{32(2+T)\left(\left\|\theta^{(T)}\right\| + 2\ln(t-T)\right)^2 + 2(2+T)(1+T)\left|\left\langle \nabla_\theta V^{\pi_\theta}(\mu)\Big|_{\theta=\theta_{md}^{(T)}}, m^{(T)}\right\rangle\right|}{(1-\gamma)^3(t+T+1)(t+T)},$$

$$(231)$$

where $\theta^{(t)**} := [2\ln(t-T), 0, 0, \cdots, 0]$ is a chosen surrogate optimal solution at time $t$. Additionally, we have the sub-optimality gap between the original optimal solution and the surrogate optimal solution can be bounded as

$$V^*(\mu) - V^{\pi_{\theta^{**}}^{(t)}}(\mu) = \frac{1}{1-\gamma}\sum_s d_\mu^{\pi^*}(s)\sum_a (\pi^*(a|s) - \pi_{\theta^{**}}^{(t)}(a|s)) \cdot A^{\pi_{\theta^{**}}^{(t)}}(s,a) \quad (232)$$

$$\leq \frac{4|\mathcal{S}|}{(1-\gamma)^2}\left(1 - \frac{\exp(2\ln(t-T))}{\exp(2\ln(t-T)) + \exp(0)\cdot(|\mathcal{A}|-1)}\right) \quad (233)$$

$$= \frac{4|\mathcal{S}|}{(1-\gamma)^2}\left(\frac{|\mathcal{A}|-1}{(t-T)^2 + |\mathcal{A}|-1}\right), \quad (234)$$

where (233) consider the upper bound of $A^{\pi_{\theta^{**}}^{(t)}}(s,a) \leq \frac{2}{1-\gamma}$ and the sum of differences between sub-optimal actions can be at most the difference between the optimal actions, which causes a pre-constant 2.

By Lemma 11, we can change the convergence rate from $V^\pi(\mu)$ to $V^\pi(\rho)$,

$$V^*(\rho) - V^{\pi_\theta^{(t)}}(\rho) \quad (235)$$

$$\leq \frac{1}{1-\gamma}\cdot\left\|\frac{d_\rho^{\pi^*}}{\mu}\right\|_\infty \cdot \left(V^*(\mu) - V^{\pi_\theta^{(t)}}(\mu)\right) \quad (236)$$

$$= \frac{1}{1-\gamma}\cdot\left\|\frac{d_\rho^{\pi^*}}{\mu}\right\|_\infty \cdot \left(\left(V^*(\mu) - V^{\pi_{\theta}^{(t)**}}(\mu)\right) + \left(V^{\pi_{\theta}^{(t)**}}(\mu) - V^{\pi_\theta^{(t)}}(\mu)\right)\right) \quad (237)$$

$$\leq \frac{1}{1-\gamma} \cdot \left\| \frac{d_\rho^{\pi^*}}{\mu} \right\|_\infty \tag{238}$$

$$\cdot \left( \frac{32(2+T)\left( \|\theta^{(T)}\| + 2\ln(t-T)\right)^2 + 2(2+T)(1+T)\left| \left\langle \nabla_\theta V^{\pi_\theta}(\mu)\Big|_{\theta=\theta_{md}^{(T)}}, m^{(T)} \right\rangle \right|}{(1-\gamma)^3(t+T+1)(t+T)} \right. \tag{239}$$

$$\left. + \frac{4|\mathcal{S}|}{(1-\gamma)^2} \frac{|\mathcal{A}|-1}{(t-T)^2 + |\mathcal{A}|-1} \right). \tag{240}$$

Finally, we complete our proof. $\qquad\square$

**Lemma 1 (Local $C$-Nearly Concavity; Formal).** The objective function $\theta \to V^{\pi_\theta}(\mu)$ is $C$-nearly concave in the directions along the feasible update domain $\mathcal{U}$ when, for all $s \in \mathcal{S}$ and $a \neq a^*(s)$, $V^{\pi_\theta}(s) > Q^*(s, a_2(s)), \theta_{s,a^*(s)} - \theta_{s,a} > M$, where

$$M = \max\left\{ \ln\left[3\binom{|\mathcal{A}|-1}{2}\right] - 2\ln[d_1 - d_i], \ln\left[\frac{(2\gamma|\mathcal{S}|C-1+\gamma)(|\mathcal{A}|-1)}{1-\gamma}\right] \right\}, \tag{241}$$

and $d_i$ are as defined in Lemma 27.

*Proof of Lemma 1.* According to Lemma 27, we can conclude that for any $s \in \mathcal{S}$, the function $\theta \to \pi_\theta(a^*(s)|s)$ is concave along the directions in $\mathcal{U}$, and for any $a \neq a^*(s)$, the functions $\theta \to \pi_\theta(a|s)$ are convex along the directions in $\mathcal{U}$, provided that $\theta_{s,a^*} - \theta_{s,a} > M$ for all $a \neq a^*$.

Now, suppose $\theta$ is the parameter such that $\theta_{s,a^*(s)} - \theta_{s,a} > M$ for all $s \in \mathcal{S}$ and $a \neq a^*(s)$. In addition, consider $\theta'$ as any parameter update obtained from $\theta$ along the directions from $\mathcal{U}$. To show that $\theta \to V^{\pi_\theta}(\mu)$ is $C$-nearly concave under the APG updates, we have to show that

$$V^{\pi_{\theta'}}(\mu) - V^{\pi_\theta}(\mu) \leq C \cdot \left\langle \nabla_\theta V^{\pi_\theta}(\mu)\Big|_{\theta=\theta}, \theta'-\theta \right\rangle. \tag{242}$$

By Lemma 5,

$$V^{\pi_{\theta'}}(\mu) - V^{\pi_\theta}(\mu) = \frac{1}{1-\gamma}\sum_{s,a} d_\mu^{\pi_{\theta'}}(s)\pi_{\theta'}(a|s)A^{\pi_\theta}(s,a) \tag{243}$$

$$= \frac{1}{1-\gamma}\sum_{s,a} d_\mu^{\pi_{\theta'}}(s)(\pi_{\theta'}(a|s) - \pi_\theta(a|s))A^{\pi_\theta}(s,a) \tag{244}$$

$$= \sum_{s,a} \frac{d_\mu^{\pi_{\theta'}}(s)}{d_\mu^{\pi_\theta}(s)}(\pi_{\theta'}(a|s) - \pi_\theta(a|s))\frac{1}{1-\gamma}d_\mu^{\pi_\theta}(s)A^{\pi_\theta}(s,a) \tag{245}$$

$$\leq \left\| \frac{d_\mu^{\pi_{\theta'}}}{d_\mu^{\pi_\theta}} \right\|_\infty \left\langle \nabla_{\pi_\theta} V^{\pi_\theta}(\mu)|_{\pi_\theta=\pi_\theta}, \pi_{\theta'} - \pi_\theta. \right\rangle. \tag{246}$$

Since $\theta_{s,a^*} - \theta_{s,a} > M$ and by Lemma 26, we have that $\|d_\mu^{\pi_{\theta'}}/d_\mu^{\pi_\theta}\|_\infty \leq C$. Moreover, by Lemma 7, we have

$$\left\langle \nabla_{\pi_\theta} V^{\pi_\theta}(\mu)|_{\pi_\theta=\pi_\theta}, \pi_{\theta'} - \pi_\theta \right\rangle = \left\langle \nabla_\theta V^{\pi_\theta}(\mu)|_{\theta=\theta}, \frac{\pi_{\theta'} - \pi_\theta}{\pi_\theta} \right\rangle. \tag{247}$$

Hence, it is remaining to show that

$$\left\langle \nabla_\theta V^{\pi_\theta}(\mu)|_{\theta=\theta}, \frac{\pi_{\theta'} - \pi_\theta}{\pi_\theta} \right\rangle \leq \left\langle \nabla_\theta V^{\pi_\theta}(\mu)|_{\theta=\theta}, \theta'-\theta \right\rangle \tag{248}$$

According to the concavity of $\theta \to \pi_\theta(a^*(s)|s)$, we have

$$\frac{\pi_{\theta'}(a^*(s)|s) - \pi_\theta(a^*(s)|s)}{\pi_\theta(a^*(s)|s)} \leq \frac{\left\langle \nabla_\theta \pi_\theta(a^*(s)|s)|_{\theta=\theta}, \theta'-\theta \right\rangle}{\pi_\theta(a^*(s)|s)} \tag{249}$$

$$= \frac{\sum_a \frac{\partial \pi_\theta(a^*(s)|s)}{\partial \theta_{s,a}} \cdot (\theta'_{s,a} - \theta_{s,a})}{\pi_\theta(a^*(s)|s)} \tag{250}$$

$$= (1 - \pi_\theta(a^*(s)|s)) \cdot (\theta'_{s,a^*(s)} - \theta_{s,a^*(s)}) \tag{251}$$

$$- \sum_{a \neq a^*(s)} \pi_\theta(a|s) \cdot (\theta'_{s,a} - \theta_{s,a}) \tag{252}$$

$$= (\theta'_{s,a^*(s)} - \theta_{s,a^*(s)}) \tag{253}$$

$$- \sum_a \pi_\theta(a|s) \cdot (\theta'_{s,a} - \theta_{s,a}). \tag{254}$$

On the other hand, since $\theta \to \pi_\theta(a_i|s)$ is convex for all $a_i \neq a^*(s)$, we have

$$\frac{\pi_{\theta'}(a_i|s) - \pi_\theta(a_i|s)}{\pi_\theta(a_i|s)} \geq \frac{\left\langle \nabla_\theta \pi_\theta(a_i|s) \Big|_{\theta=\theta}, \theta' - \theta \right\rangle}{\pi_\theta(a_i|s)} \tag{255}$$

$$= \frac{\sum_a \frac{\partial \pi_\theta(a_i|s)}{\partial \theta_{s,a}} \cdot (\theta'_{s,a} - \theta_{s,a})}{\pi_\theta(a_i|s)} \tag{256}$$

$$= (1 - \pi_\theta(a_i|s)) \cdot (\theta'_{s,a_i} - \theta_{s,a_i}) \tag{257}$$

$$- \sum_{a \neq a_i} \pi_\theta(a|s) \cdot (\theta'_{s,a} - \theta_{s,a}) \tag{258}$$

$$= (\theta'_{s,a_i} - \theta_{s,a_i}) \tag{259}$$

$$- \sum_a \pi_\theta(a|s) \cdot (\theta'_{s,a} - \theta_{s,a}). \tag{260}$$

Finally, we put everything together, by (254) and (260), we have

$$\left\langle \nabla_\theta V^{\pi_\theta}(\mu) \Big|_{\theta=\theta}, \frac{\pi_{\theta'} - \pi_\theta}{\pi_\theta} \right\rangle \leq \left\langle \nabla_\theta V^{\pi_\theta}(\mu)|_{\theta=\theta}, \theta' - \theta \right\rangle$$

$$- \sum_s \frac{\partial V^{\pi_\theta}(\mu)}{\partial \theta_{s,a^*(s)}} \sum_a \pi_\theta(a|s) \cdot (\theta'_{s,a} - \theta_{s,a}) \tag{261}$$

$$- \sum_s \sum_{a \neq a^*(s)} \frac{\partial V^{\pi_\theta}(\mu)}{\partial \theta_{s,a}} \sum_{a'} \pi_\theta(a'|s) \cdot (\theta'_{s,a'} - \theta_{s,a'}) \tag{262}$$

$$= \left\langle \nabla_\theta V^{\pi_\theta}(\mu) \Big|_{\theta=\theta}, \theta' - \theta \right\rangle$$

$$- \sum_{a'} \pi_\theta(a'|s) \cdot (\theta'_{s,a'} - \theta_{s,a'}) \sum_a \frac{\partial V^{\pi_\theta}(\mu)}{\partial \theta_{s,a}} \tag{263}$$

$$= \left\langle \nabla_\theta V^{\pi_\theta}(\mu) \Big|_{\theta=\theta}, \theta' - \theta \right\rangle, \tag{264}$$

where (261) and (262) hold by (254) and (260), and since $V^{\pi_\theta}(s) > Q^*(s, a_2(s))$, we have that only $A^{\pi_\theta}(s, a^*(s)) > 0$ and $A^{\pi_\theta}(s, a) < 0$ for all $a \neq a^*(s)$ by Lemma 12. (264) is because

$$\sum_a \frac{\partial V^{\pi_\theta}(\mu)}{\partial \theta_{s,a}} = \sum_a \frac{1}{1-\gamma} d_\mu^{\pi_\theta}(s) \pi_\theta(a|s) A^{\pi_\theta}(s, a) \tag{265}$$

$$= \frac{1}{1-\gamma} d_\mu^{\pi_\theta}(s) \sum_a \pi_\theta(a|s) A^{\pi_\theta} = 0. \tag{266}$$

Therefore, we obtain (248), which leads to the desired result. Hence, we complete the proof. $\qquad\square$

**Remark 15.** Please note that the $C$-near concavity in Lemma 1 is derived from $\|\frac{d_\mu^{\pi_{\theta'}}}{d_\mu^{\pi_\theta}}\|_\infty \leq C$ in (246). As demonstrated in Lemma 26, for any given $\varepsilon > 0$, it is possible to achieve an upper bound $C = 1 + \varepsilon$ by considering a policy that is even closer to being optimal. This implies that as we move closer to a near-optimal policy $\pi_\theta$, the objective $\theta \to V^{\pi_\theta}(\mu)$ becomes increasingly closer to being concave, with the deviation becoming arbitrarily small as $\varepsilon$ approaches zero.

**Lemma 2.** *Consider a tabular softmax parameterized policy $\pi_\theta$. Under APG $\eta^{(t)} = \frac{t}{t+1} \cdot \frac{(1-\gamma)^3}{16}$, given any $M > 0$, there exists a finite time $T$ such that for all $t \geq T$, $s \in \mathcal{S}$, and $a \neq a^*(s)$, we have, (i) $\theta_{s,a^*(s)} - \theta_{s,a} > M$, (ii) $V^{\pi_\theta^{(t)}}(s) > Q^*(s, a_2(s))$, (iii) $\left.\frac{\partial V^{\pi_\theta}(\mu)}{\partial \theta_{s,a^*(s)}}\right|_{\theta=\omega^{(t)}} \geq 0 \geq \left.\frac{\partial V^{\pi_\theta}(\mu)}{\partial \theta_{s,a}}\right|_{\theta=\omega^{(t)}}$, (iv) $\omega^{(t)}_{s,a^*(s)} - \theta^{(t)}_{s,a^*(s)} \geq \omega^{(t)}_{s,a} - \theta^{(t)}_{s,a}$.*

*Proof of Lemma 2.* Let's fix a state $s \in \mathcal{S}$. We first show the (i) $\theta_{s,a^*(s)} - \theta_{s,a} > M$ part. By Theorem 1 and Assumption 3, we have $\pi_\theta(a^*(s)|s) \to 1$ as $t \to \infty$. Given $M > 0$. If there is no such finite $T_M(s)$, then there are infinitely many $t$ such that for every $t$, there is an action $a_t$ satisfying that $\theta^{(t)}_{s,a^*(s)} - \theta^{(t)}_{s,a} \leq M$. Since the action space is finite, there must be an action $\tilde{a}$ such that $\theta^{(t)}_{s,a^*(s)} - \theta^{(t)}_{s,\tilde{a}} \leq M$ holds for infinitely many $t$. Accordingly, there are infinitely many $t$ such that

$$\pi_\theta^{(t)}(a^*(s)|s) = \frac{\exp(\theta^{(t)}_{s,a^*(s)})}{\sum_a \exp(\theta^{(t)}_{s,a})} \leq \frac{\exp(\theta^{(t)}_{s,\tilde{a}} + M)}{\exp(\theta^{(t)}_{s,\tilde{a}} + M) + \exp(\theta^{(t)}_{s,\tilde{a}})} = \frac{\exp(M)}{\exp(M) + 1} < 1, \quad (267)$$

which contradicts to the limit of $\pi_\theta(a^*(s)|s)$. Hence, given any $M > 0$, there exists a finite $T_M(s)$ such that for all $t \geq T_M(s)$, $\theta_{s,a^*(s)} - \theta_{s,a} > M$. We take $T_M = \max_{s \in \mathcal{S}} T_M(s)$.

(ii) For the $V^{\pi_\theta^{(t)}}(s) > Q^*(s, a_2(s))$ part, it can be directly proven by Lemma 25 and choosing the finite time $T_V(s)$. We take $T_V = \max_{s \in \mathcal{S}} T_V(s)$.

Finally, we prove (iii) $\left.\frac{\partial V^{\pi_\theta}(\mu)}{\partial \theta_{s,a^*(s)}}\right|_{\theta=\omega^{(t)}} \geq 0 \geq \left.\frac{\partial V^{\pi_\theta}(\mu)}{\partial \theta_{s,a}}\right|_{\theta=\omega^{(t)}}$ for all $a \neq a^*(s)$ and (iv) $\omega^{(t)}_{s,a^*(s)} - \theta^{(t)}_{s,a^*(s)} \geq \omega^{(t)}_{s,a} - \theta^{(t)}_{s,a}$ together, we make the following claim.

**Claim 1.** *The following hold:*

*a) There exists a finite $T_1(s)$ such that if $t \geq T_1(s)$, then $\left.\frac{\partial V^{\pi_\theta}(\mu)}{\partial \theta_{s,a^*(s)}}\right|_{\theta=\omega^{(t)}} \geq 0 \geq \left.\frac{\partial V^{\pi_\theta}(\mu)}{\partial \theta_{s,a}}\right|_{\theta=\omega^{(t)}}$ for all $a \neq a^*(s)$.*

*b) There exists a finite $T_\omega(s) \geq T_1(s)$ such that if $t \geq T_\omega(s)$, then $\omega^{(t)}_{s,a^*(s)} - \theta^{(t)}_{s,a^*(s)} \geq \omega^{(t)}_{s,a} - \theta^{(t)}_{s,a}$ for all $a \neq a^*(s)$.*

Then, we prove Claim 1 as follows:

**Claim a).** By Theorem 1, we know that our value converge to the optimal, i.e., $V^{\pi_\omega^{(t)}}(s) \to V^*(s)$ as $t \to \infty$. Thus, there is a finite $T_1(s)$ such that if $t \geq T_1(s)$, $V^{\pi_\omega^{(t)}}(s) > Q^*(s, a_2(s))$. By Lemma 12, we know that $Q^{\pi_\omega^{(t)}}(s, a) \leq Q^*(s, a_2(s))$ for $a \neq a^*(s)$. Hence, $A^{\pi_\omega^{(t)}}(s, a^*(s)) > 0$ and $A^{\pi_\omega^{(t)}}(s, a) < 0$ for all $a \neq a^*(s)$, which directly leads to the fact that $\left.\frac{\partial V^{\pi_\theta}(\mu)}{\partial \theta_{s,a^*(s)}}\right|_{\theta=\omega^{(t)}} \geq 0 \geq \left.\frac{\partial V^{\pi_\theta}(\mu)}{\partial \theta_{s,a}}\right|_{\theta=\omega^{(t)}}$ for all $a \neq a^*(s)$.

**Claim b).** We prove this claim by two parts. Part (i): We show that for any $t \geq T_1(s)$ and any $a \in \mathcal{A}$, if $\omega^{(t)}_{s,a^*(s)} - \theta^{(t)}_{s,a^*(s)} \geq \omega^{(t)}_{s,a} - \theta^{(t)}_{s,a}$, then $\omega^{(t+1)}_{s,a^*(s)} - \theta^{(t+1)}_{s,a^*(s)} \geq \omega^{(t+1)}_{s,a} - \theta^{(t+1)}_{s,a}$. By considering the updates (7), we have

$$\theta^{(t+1)}_{s,a} \leftarrow \theta^{(t)}_{s,a} + (\omega^{(t)}_{s,a} - \theta^{(t)}_{s,a}) + \eta^{(t+1)} \left.\frac{\partial V^{\pi_\theta}(\mu)}{\partial \theta_{s,a}}\right|_{\theta=\omega^{(t)}}. \quad (268)$$

By using the update (8) and (9) with respect to the optimal action $a^*(s)$ at state $s$ and (268), and for simplicity, let $\mathbb{I}^{(t+1)} := \mathbb{I}\left\{V^{\pi_\varphi^{(t+1)}}(\mu) \geq V^{\pi_\theta^{(t+1)}}(\mu)\right\}$, we have

$$\omega^{(t+1)}_{s,a^*(s)} - \theta^{(t+1)}_{s,a^*(s)} = \mathbb{I}^{(t+1)} \frac{t}{t+3}(\theta^{(t+1)}_{s,a^*(s)} - \theta^{(t)}_{s,a^*(s)}) \quad (269)$$

$$= \mathbb{I}^{(t+1)} \frac{t}{t+3} \left( \omega^{(t)}_{s,a^*(s)} - \theta^{(t)}_{s,a^*(s)} + \eta^{(t+1)} \left. \frac{\partial V^{\pi_\theta}(\mu)}{\partial \theta_{s,a^*(s)}} \right|_{\theta=\omega^{(t)}} \right) \tag{270}$$

$$\geq \mathbb{I}^{(t+1)} \frac{t}{t+3} \left( \omega^{(t)}_{s,a} - \theta^{(t)}_{s,a} + \eta^{(t+1)} \left. \frac{\partial V^{\pi_\theta}(\mu)}{\partial \theta_{s,a}} \right|_{\theta=\omega^{(t)}} \right) \tag{271}$$

$$= \mathbb{I}^{(t+1)} \frac{t}{t+3} (\theta^{(t+1)}_{s,a} - \theta^{(t)}_{s,a}) \tag{272}$$

$$= \omega^{(t+1)}_{s,a} - \theta^{(t+1)}_{s,a}, \tag{273}$$

where (271) is followed by the hypothesis $\omega^{(t)}_{s,a^*(s)} - \theta^{(t)}_{s,a^*(s)} \geq \omega^{(t)}_{s,a} - \theta^{(t)}_{s,a}$ and $\left. \frac{\partial V^{\pi_\theta}(\mu)}{\partial \theta_{s,a^*(s)}} \right|_{\theta=\omega^{(t)}} \geq \left. \frac{\partial V^{\pi_\theta}(\mu)}{\partial \theta_{s,a}} \right|_{\theta=\omega^{(t)}}$ because $t \geq T_1(s)$. We complete the proof of Part(i).

Part (ii): We show that there exists a finite $T_\omega(s) \geq T_1(s)$ such that if $t \geq T_\omega(s)$, then $\omega^{(t)}_{s,a^*(s)} - \theta^{(t)}_{s,a^*(s)} \geq \omega^{(t)}_{s,a} - \theta^{(t)}_{s,a}$. We prove it by contradiction. Suppose that there is no such $T_\omega(s)$. Then, there are infinitely many $t \geq T_1(s)$ such that there is an action $a$ violating the condition, $\omega^{(t)}_{s,a^*(s)} - \theta^{(t)}_{s,a^*(s)} < \omega^{(t)}_{s,a} - \theta^{(t)}_{s,a}$. Moreover, since our action space is finite, there must be an action $\tilde{a}$ such that $\omega^{(t)}_{s,a^*(s)} - \theta^{(t)}_{s,a^*(s)} < \omega^{(t)}_{s,\tilde{a}} - \theta^{(t)}_{s,\tilde{a}}$ hold for infinitely many $t \geq T_1(s)$.

We claim that for all $t \geq T_1(s)$, $\omega^{(t)}_{s,a^*(s)} - \theta^{(t)}_{s,a^*(s)} < \omega^{(t)}_{s,\tilde{a}} - \theta^{(t)}_{s,\tilde{a}}$. If not, there is a $t_0 \geq T_1(s)$ such that $\omega^{(t_0)}_{s,a^*(s)} - \theta^{(t_0)}_{s,a^*(s)} \geq \omega^{(t_0)}_{s,\tilde{a}} - \theta^{(t_0)}_{s,\tilde{a}}$. Then, by Part (i), $\omega^{(t)}_{s,a^*(s)} - \theta^{(t)}_{s,a^*(s)} \geq \omega^{(t)}_{s,\tilde{a}} - \theta^{(t)}_{s,\tilde{a}}$ hold for every $t \geq t_0$, which contradicts to $\omega^{(t)}_{s,a^*(s)} - \theta^{(t)}_{s,a^*(s)} < \omega^{(t)}_{s,\tilde{a}} - \theta^{(t)}_{s,\tilde{a}}$ hold for infinitely many $t \geq T_1(s)$. Therefore, $\omega^{(t)}_{s,a^*(s)} - \theta^{(t)}_{s,a^*(s)} < \omega^{(t)}_{s,\tilde{a}} - \theta^{(t)}_{s,\tilde{a}}$ hold for all $t \geq T_1(s)$. Note that $\omega^{(t)}_{s,a^*(s)} - \theta^{(t)}_{s,a^*(s)} \neq \omega^{(t)}_{s,\tilde{a}} - \theta^{(t)}_{s,\tilde{a}}$ hold for all $t \geq T_1(s)$, which implies that no $\mathbb{I}^{(t)}$ can be 0 for all $t \geq T_1(s)$. If $\mathbb{I}^{(t)} = 0$, both $\omega^{(t)}_{s,a^*(s)} - \theta^{(t)}_{s,a^*(s)}, \omega^{(t)}_{s,\tilde{a}} - \theta^{(t)}_{s,\tilde{a}}$ will be zero and they will be equal.

To meet the contradiction, for any $N > T_1(s)$, we consider

$$\theta^{(N)}_{s,a^*(s)} = \theta^{(T_1(s))}_{s,a^*(s)} + \sum_{t=T_1(s)}^{N-1} (\theta^{(t+1)}_{s,a^*(s)} - \theta^{(t)}_{s,a^*(s)}) \tag{274}$$

$$= \theta^{(T_1(s))}_{s,a^*(s)} + \sum_{t=T_1(s)}^{N-1} \frac{t+3}{t} (\omega^{(t+1)}_{s,a^*(s)} - \theta^{(t+1)}_{s,a^*(s)}) \tag{275}$$

$$< \theta^{(T_1(s))}_{s,a^*(s)} + \sum_{t=T_1(s)}^{N-1} \frac{t+3}{t} (\omega^{(t+1)}_{s,\tilde{a}} - \theta^{(t+1)}_{s,\tilde{a}}) \tag{276}$$

$$= \theta^{(T_1(s)))}_{s,a^*(s)} + \sum_{t=T_1(s)}^{N-1} \frac{t+3}{t} \left[ \frac{t}{t+3} (\theta^{(t+1)}_{s,\tilde{a}} - \theta^{(t)}_{s,\tilde{a}}) \right] \tag{277}$$

$$= \theta^{(T_1(s))}_{s,a^*(s)} + \sum_{t=T_1(s)}^{N-1} (\theta^{(t+1)}_{s,\tilde{a}} - \theta^{(t)}_{s,\tilde{a}}) \tag{278}$$

$$= \theta^{(T_1(s))}_{s,a^*(s)} - \theta^{(T_1(s))}_{s,\tilde{a}} + \theta^{(N)}_{s,\tilde{a}}, \tag{279}$$

where (274) uses a simple telescope argument, (275) and (277) use the update (9) of APG and the fact that $\mathbb{I}^{(t)} = 1$ for all $t \geq T_1(s)$, and (276) holds since $t \geq T_1(s)$.

Since (279) holds for arbitrary $N > T_1(s)$, we obtain $\theta^{(N)}_{s,a^*(s)} < \theta^{(T_1(s))}_{s,a^*(s)} - \theta^{(T_1(s))}_{s,\tilde{a}} + \theta^{(N)}_{s,\tilde{a}}$ for any $N > T_1(s)$, which contradicts $\pi^{(t)}_\theta(a^*) \to 1$ as $t \to \infty$, i.e., the asymptotic global convergence Theorem 1. Hence, there is a $T_\omega(s) \geq T_1(s)$ such that if $t \geq T_\omega(s)$, then $\omega^{(t)}_{s,a^*(s)} - \theta^{(t)}_{s,a^*(s)} \geq \omega^{(t)}_{s,a} - \theta^{(t)}_{s,a}$. Finally, let $T_\omega = \max_{s\in\mathcal{S}} T_\omega(s)$ and $T = \max\{T_M, T_V, T_\omega\}$, we complete the proof. $\qquad \square$

# E   DETAILED EXPLANATION OF THE MOTIVATING EXAMPLE AND THE EXPERIMENTAL CONFIGURATIONS

## 4.2 Motivating Examples of APG

Consider a simple two-action bandit with actions $a^*, a_2$ and reward function $r(a^*) = 1, r(a_2) = 0$. Accordingly, the objective we aim to optimize is $\mathbb{E}_{a \sim \pi_\theta}[r(a)] = \pi_\theta(a^*)$. By deriving the Hessian matrix with respect to our policy parameters $\theta_{a^*}$ and $\theta_{a_2}$, the Hessian matrix can be written as:

$$\mathbf{H} = \begin{bmatrix} \frac{\partial^2 \pi_\theta(a^*)}{\partial \theta_{a^*} \partial \theta_{a^*}} & \frac{\partial^2 \pi_\theta(a^*)}{\partial \theta_{a^*} \partial \theta_{a_2}} \\ \frac{\partial^2 \pi_\theta(a^*)}{\partial \theta_{a_2} \partial \theta_{a^*}} & \frac{\partial^2 \pi_\theta(a^*)}{\partial \theta_{a_2} \partial \theta_{a_2}} \end{bmatrix} \tag{280}$$

$$= \begin{bmatrix} \pi_\theta(a^*)(1 - \pi_\theta(a^*))(1 - 2\pi_\theta(a^*)) & \pi_\theta(a^*)(1 - \pi_\theta(a^*))(2\pi_\theta(a^*) - 1) \\ \pi_\theta(a^*)(1 - \pi_\theta(a^*))(2\pi_\theta(a^*) - 1) & \pi_\theta(a^*)(1 - \pi_\theta(a^*))(1 - 2\pi_\theta(a^*)) \end{bmatrix}, \tag{281}$$

where the eigenvalue $\lambda_1 = 2\pi_\theta(a^*)(\pi_\theta(a^*) - 1)(2\pi_\theta(a^*) - 1), \lambda_2 = 0$. So we have that if $\pi_\theta(a^*) \geq 0.5$, then $\lambda_1, \lambda_2 \leq 0$, leading to the fact that the Hessian is negative semi-definite. Additionally, we have that the Hessian is negative semi-definite if and only if the objective $\mathbb{E}_{a \sim \pi_\theta}[r(a)]$ is concave, which complete our proof.

## 6.1 Numerical Validation of the Convergence Rates of APG

**(Bandit)** We conduct a 3-action bandit experiment with actions $\mathcal{A} = [a^*, a_2, a_3]$, where the corresponding rewards are $r = [r(a^*), r(a_2), r(a_3)] = [1, 0.99, 0]$. We initialize the policy parameters with both a uniform initialization ($\theta^{(0)} = [0, 0, 0], \pi^{(0)} = [1/3, 1/3, 1/3]$) and a hard initialization ($\theta^{(0)} = [1, 3, 5], \pi^{(0)} = [0.01588, 0.11731, 0.86681]$ and hence the optimal action has the smallest initial probability).

**(MDP)** We conduct an experiment on an MDP with 5 states and 5 actions under the initial state distribution $\rho = [0.3, 0.2, 0.1, 0.15, 0.25]$. The reward, initial policy parameters, transition probability can be found in the following Table 1-8.

Table 1: Experimental settings: Reward function

| $r(s,a)$ | $a_1$ | $a_2$ | $a_3$ | $a_4$ | $a_5$ |
|---|---|---|---|---|---|
| $s_1$ | 1.0 | 0.8 | 0.6 | 0.7 | 0.4 |
| $s_2$ | 0.5 | 0.3 | 0.1 | 1.0 | 0.6 |
| $s_3$ | 0.6 | 0.9 | 0.8 | 0.7 | 1.0 |
| $s_4$ | 0.1 | 0.2 | 0.6 | 0.7 | 0.4 |
| $s_5$ | 0.8 | 0.4 | 0.6 | 0.2 | 0.9 |

Table 2: Experimental settings: Hard initialization

| $\theta_{s,a}^{(0)}$ | $a_1$ | $a_2$ | $a_3$ | $a_4$ | $a_5$ |
|---|---|---|---|---|---|
| $s_1$ | 1 | 2 | 3 | 4 | 5 |
| $s_2$ | 3 | 4 | 5 | 1 | 2 |
| $s_3$ | 5 | 2 | 3 | 4 | 1 |
| $s_4$ | 5 | 4 | 2 | 1 | 3 |
| $s_5$ | 2 | 4 | 3 | 5 | 1 |

Table 3: Experimental settings: Uniform initialization

| $\theta_{s,a}^{(0)}$ | $a_1$ | $a_2$ | $a_3$ | $a_4$ | $a_5$ |
|---|---|---|---|---|---|
| $s_1$ | 0 | 0 | 0 | 0 | 0 |
| $s_2$ | 0 | 0 | 0 | 0 | 0 |
| $s_3$ | 0 | 0 | 0 | 0 | 0 |
| $s_4$ | 0 | 0 | 0 | 0 | 0 |
| $s_5$ | 0 | 0 | 0 | 0 | 0 |

Table 4: Experimental settings: Transition proba-Table 5: Experimental settings: Transition probability $P(\cdot|s_0,\cdot)$ bility $P(\cdot|s_1,\cdot)$

| $P(s|s_0,a)$ | $a_1$ | $a_2$ | $a_3$ | $a_4$ | $a_5$ |
|---|---|---|---|---|---|
| $s_1$ | 0.1 | 0.6 | 0.5 | 0.4 | 0.2 |
| $s_2$ | 0.5 | 0.1 | 0.1 | 0.3 | 0.1 |
| $s_3$ | 0.1 | 0.1 | 0.1 | 0.1 | 0.1 |
| $s_4$ | 0.2 | 0.1 | 0.2 | 0.1 | 0.1 |
| $s_5$ | 0.1 | 0.1 | 0.1 | 0.1 | 0.5 |

| $P(s|s_1,a)$ | $a_1$ | $a_2$ | $a_3$ | $a_4$ | $a_5$ |
|---|---|---|---|---|---|
| $s_1$ | 0.1 | 0.4 | 0.1 | 0.4 | 0.2 |
| $s_2$ | 0.5 | 0.1 | 0.4 | 0.1 | 0.2 |
| $s_3$ | 0.2 | 0.2 | 0.3 | 0.1 | 0.2 |
| $s_4$ | 0.1 | 0.2 | 0.1 | 0.1 | 0.2 |
| $s_5$ | 0.1 | 0.1 | 0.1 | 0.3 | 0.2 |

Table 6: Experimental settings: Transition proba-Table 7: Experimental settings: Transition probability $P(\cdot|s_2,\cdot)$ bility $P(\cdot|s_3,\cdot)$

| $P(s|s_2,a)$ | $a_1$ | $a_2$ | $a_3$ | $a_4$ | $a_5$ |
|---|---|---|---|---|---|
| $s_1$ | 0.6 | 0.2 | 0.3 | 0.1 | 0.2 |
| $s_2$ | 0.1 | 0.4 | 0.3 | 0.4 | 0.1 |
| $s_3$ | 0.1 | 0.1 | 0.2 | 0.3 | 0.1 |
| $s_4$ | 0.1 | 0.2 | 0.1 | 0.1 | 0.1 |
| $s_5$ | 0.1 | 0.1 | 0.1 | 0.1 | 0.5 |

| $P(s|s_3,a)$ | $a_1$ | $a_2$ | $a_3$ | $a_4$ | $a_5$ |
|---|---|---|---|---|---|
| $s_1$ | 0.6 | 0.1 | 0.2 | 0.4 | 0.5 |
| $s_2$ | 0.1 | 0.5 | 0.1 | 0.3 | 0.1 |
| $s_3$ | 0.1 | 0.1 | 0.1 | 0.1 | 0.1 |
| $s_4$ | 0.1 | 0.2 | 0.1 | 0.1 | 0.2 |
| $s_5$ | 0.1 | 0.1 | 0.5 | 0.1 | 0.1 |

Table 8: Experimental settings: Transition probability $P(\cdot|s_4,\cdot)$

| $P(s|s_4,a)$ | $a_1$ | $a_2$ | $a_3$ | $a_4$ | $a_5$ |
|---|---|---|---|---|---|
| $s_1$ | 0.2 | 0.4 | 0.4 | 0.1 | 0.2 |
| $s_2$ | 0.2 | 0.1 | 0.1 | 0.4 | 0.5 |
| $s_3$ | 0.2 | 0.2 | 0.1 | 0.2 | 0.1 |
| $s_4$ | 0.2 | 0.2 | 0.3 | 0.1 | 0.1 |
| $s_5$ | 0.2 | 0.1 | 0.1 | 0.2 | 0.1 |

## F  Convergence Rate of Policy Gradient under Softmax Parameterization

The Gradient Descent (GD) method within the realm of optimization exhibits a convergence rate of $O(1/t)$ when applied to convex objectives. Remarkably, (Mei et al., 2020) have demonstrated that the Policy Gradient (PG) method, even when dealing with nonconvex objectives, can achieve the same convergence rate, leveraging the non-uniform Polyak–Łojasiewicz (PL) condition under softmax parameterization. In this paper, we employ a primary proof technique involving the characterization of local $C$-nearly concavity and subsequently establish that the Accelerated Policy Gradient (APG) method attains this convergence regime. Our investigation also reveals that the PG method with softmax parameterization shares this intriguing property.

In this section, we will begin by providing a proof demonstrating the $\tilde{O}(1/t)$ convergence rate for PG when optimizing the value objectives that are $C$-nearly concave, while employing the softmax parameterized policy. Additionally, we will offer a proof establishing the capability of PG to reach the local $C$-nearly concavity regime within a finite number of time steps, considering the general MDP setting.

### F.1  Convergence Rate under Nearly Concave Objectives for Policy Gradient

**Theorem 5.** *Let $\left\{\theta^{(t)}\right\}_{t\geq 1}$ be computed by PG and given a set $\mathcal{X}$ such that $V^{\pi_\theta}(\mu)$ is $C$-nearly concave in $\mathcal{X}$. Suppose $\left\{\theta^{(t)}\right\}_{t\geq 1}$ always remain in the set $\mathcal{X}$, for all $t$. Then for any $t \geq 1$ and any $\theta^{**}$, we have*

$$V^{\pi_{\theta^{**}}}(\mu) - V^{\pi_\theta^{(t)}}(\mu) \leq \frac{2LC^2\left(\|\theta^{**}\| + \kappa \ln t\right)^2}{t-1}, \tag{282}$$

*where $L$ is the Lipschitz constant of the objective and $\kappa \in \mathbb{R}$ is a finite constant.*

*Proof of Theorem 5.*   First, by Lemma 9 and (6), we have

$$-V^{\pi_\theta^{(t+1)}}(\mu) \leq -V^{\pi_\theta^{(t)}}(\mu) - \left\langle \nabla_\theta V^{\pi_\theta}(\mu)\Big|_{\theta=\theta^{(t)}}, \theta^{(t+1)} - \theta^{(t)} \right\rangle + \frac{L}{2}\left\|\theta^{(t+1)} - \theta^{(t)}\right\|^2 \tag{283}$$

$$= -V^{\pi_\theta^{(t)}}(\mu) - \eta\left\|\nabla_\theta V^{\pi_\theta}(\mu)\Big|_{\theta=\theta^{(t)}}\right\|^2 + \frac{L\eta^2}{2}\left\|\nabla_\theta V^{\pi_\theta}(\mu)\Big|_{\theta=\theta^{(t)}}\right\|^2 \tag{284}$$

$$= -V^{\pi_\theta^{(t)}}(\mu) - \frac{1}{2L}\left\|\nabla_\theta V^{\pi_\theta}(\mu)\Big|_{\theta=\theta^{(t)}}\right\|^2. \tag{285}$$

Also by the nearly concavity of objective in the set $\mathcal{X}$, we have

$$-V^{\pi_\theta^{(t)}}(\mu) \leq -V^{\pi_\theta^{**}}(\mu) + C \cdot \left\langle \nabla_\theta V^{\pi_\theta}(\mu)\Big|_{\theta=\theta^{(t)}}, \theta^{**} - \theta^{(t)} \right\rangle. \tag{286}$$

Define

$$\delta^{(t)} := V^{\pi_\theta^{**}}(\mu) - V^{\pi_\theta^{(t)}}(\mu). \tag{287}$$

Then (285) leads to

$$\delta^{(t+1)} - \delta^{(t)} \leq -\frac{1}{2L}\left\|\nabla_\theta V^{\pi_\theta}(\mu)\Big|_{\theta=\theta^{(t)}}\right\|^2. \tag{288}$$

Moreover, (286) leads to

$$\delta^{(t)} \leq C \cdot \left\langle \nabla_\theta V^{\pi_\theta}(\mu)\Big|_{\theta=\theta^{(t)}}, \theta^{**} - \theta^{(t)} \right\rangle \tag{289}$$

$$\leq C\left\|\nabla_\theta V^{\pi_\theta}(\mu)\Big|_{\theta=\theta^{(t)}}\right\|\left\|\theta^{**} - \theta^{(t)}\right\| \tag{290}$$

$$\leq C\left\|\nabla_\theta V^{\pi_\theta}(\mu)\Big|_{\theta=\theta^{(t)}}\right\|\left(\|\theta^{**}\| + \left\|\theta^{(t)}\right\|\right) \tag{291}$$

$$\leq C\left\|\nabla_\theta V^{\pi_\theta}(\mu)\Big|_{\theta=\theta^{(t)}}\right\|\left(\|\theta^{**}\| + \max_{t'\leq t}\left\|\theta^{(t')}\right\|\right). \tag{292}$$

By substituting and rearranging (288) and (292), we have

$$\delta^{(t+1)} \le \delta^{(t)} - \frac{1}{2LC^2 \left( \|\theta^{**}\| + \max_{t' \le t} \|\theta^{(t')}\| \right)^2} \cdot \delta^{(t)2} \tag{293}$$

$$\delta^{(t)} \ge \delta^{(t+1)} + \frac{1}{2LC^2 \left( \|\theta^{**}\| + \max_{t' \le t} \|\theta^{(t')}\| \right)^2} \delta^{(t)2} \tag{294}$$

$$\frac{1}{\delta^{(t+1)}} \ge \frac{1}{\delta^{(t)}} + \frac{\delta^{(t)}}{\left( 2LC^2 \left( \|\theta^{**}\| + \max_{t' \le t} \|\theta^{(t')}\| \right)^2 \right) \delta^{(t+1)}} \tag{295}$$

$$\frac{1}{\delta^{(t+1)}} - \frac{1}{\delta^{(t)}} \ge \frac{\delta^{(t)}}{\delta^{(t+1)}} \frac{1}{2LC^2 \left( \|\theta^{**}\| + \max_{t' \le t} \|\theta^{(t')}\| \right)^2} \ge \frac{1}{2LC^2 \left( \|\theta^{**}\| + \max_{t' \le t} \|\theta^{(t')}\| \right)^2}. \tag{296}$$

By implementing telescoping on (296), we have

$$\delta^{(t)} = V^{\pi_\theta^{**}}(\mu) - V^{\pi_\theta^{(t)}}(\mu) \le \frac{2LC^2 \left( \|\theta^{**}\| + \max_{t' \le t} \|\theta^{(t')}\| \right)^2}{t - 1}. \tag{297}$$

Finally, we claim that the $\theta$-norm exhibits a convergence rate of $O(\ln t)$. Without loss of generality, we assume that $\sum_{a \in \mathcal{A}} \theta_{s,a}^{(t)} = 0$ for all $s \in \mathcal{S}$. Now we prove it by contradiction. Assume that $\theta$-norm doesn't exhibit a convergence rate of $O(\ln t)$, then for any $M \in \mathbb{R}$, there must exist a finite time $T_1$ such that $\|\theta^{(t)}\| / \ln t > M$ for all $t > T_1$.

Moreover, by the asymptotic convergence results of (Agarwal et al., 2021), we have there exist an upper bound $\bar{M}$ and a finite time $T_2$ such that

$$\theta_{s,a}^{(t)} < \bar{M}, \text{ for all } a \ne a^*(s), s \in \mathcal{S}, t \in \mathbb{N}, \tag{298}$$

$$\theta_{s,a^*(s)}^{(t)} > \theta_{s,a}^{(t)}, \text{ for all } a \ne a^*(s), s \in \mathcal{S}, t > T_2. \tag{299}$$

Hence, by choosing $M = 2|\mathcal{A}|$, then we have

$$\frac{\|\theta^{(t)}\|}{\ln t} > 2|\mathcal{A}|, \text{ for all } t > \max\{T_1, T_2\}. \tag{300}$$

Note that by (298)-(299) and the fact that $\|\theta^{(t)}\| > 2|\mathcal{A}|\ln t$, we have that $\theta_{s,a^*(s)} > 2\ln t$. Hence, for all $t > \max\{T_1, T_2\}, s \in \mathcal{S}$, we have that

$$\pi_\theta^{(t)}(a^*(s)|s) \ge \frac{\exp(2\ln t)}{\exp(2\ln t) + (|\mathcal{A}| - 1)\exp(\bar{M})} = \frac{t^2}{t^2 + (|\mathcal{A}| - 1)\exp(\bar{M})}, \tag{301}$$

which directly leads to the bound of the gradient

$$\left| \frac{\partial V^{\pi_\theta^{(t)}}(\mu)}{\partial \theta_{s,a^*(s)}} \right| = \left| \frac{1}{1 - \gamma} \cdot d_\mu^{\pi_\theta^{(t)}}(s) \cdot \pi_\theta^{(t)}(a^*(s)|s) \cdot A^{\pi_\theta^{(t)}}(s, a^*(s)) \right| \tag{302}$$

$$\le \left| \frac{1}{1 - \gamma} \cdot 1 \cdot 1 \cdot (Q^{\pi_\theta^{(t)}}(s, a^*(s)) - V^{\pi_\theta^{(t)}}(s)) \right| \tag{303}$$

$$= \frac{1}{1 - \gamma} \cdot \left| Q^{\pi_\theta^{(t)}}(s, a^*(s)) - \pi_\theta^{(t)}(a^*(s)|s) Q^{\pi_\theta^{(t)}}(s, a^*(s)) \right. \tag{304}$$

$$\left. - \sum_{a \ne a^*(s)} \pi_\theta^{(t)}(a|s) Q^{\pi_\theta^{(t)}}(s, a)) \right| \tag{305}$$

$$\le \frac{1 - \pi_\theta^{(t)}(a^*(s)|s)}{1 - \gamma} \left( \left| Q^{\pi_\theta^{(t)}}(s, a^*(s)) \right| + \max_a \left| Q^{\pi_\theta^{(t)}}(s, a) \right| \right) \tag{306}$$

$$\le \frac{2(|\mathcal{A}| - 1)\exp(\bar{M})}{\left( t^2 + (|\mathcal{A}| - 1)\exp(\bar{M}) \right)(1 - \gamma)^2} = O(\frac{1}{t^2}), \tag{307}$$

$$\left|\frac{\partial V^{\pi_\theta^{(t)}}(\mu)}{\partial \theta_{s,a}}\right| = \left|\frac{1}{1-\gamma} \cdot d_\mu^{\pi_\theta^{(t)}}(s) \cdot \pi_\theta^{(t)}(a|s) \cdot A^{\pi_\theta^{(t)}}(s,a)\right| \tag{308}$$

$$\leq \left|\frac{1}{1-\gamma} \cdot 1 \cdot (1 - \pi_\theta^{(t)}(a^*(s)|s)) \cdot \frac{1}{1-\gamma}\right| \tag{309}$$

$$= \frac{(|\mathcal{A}|-1)\exp(\bar{M})}{\left(t^2 + (|\mathcal{A}|-1)\exp(\bar{M})\right)(1-\gamma)^2} = O(\frac{1}{t^2}), \text{ for all } a \neq a^*(s). \tag{310}$$

And so by the update rule of PG, for all $t > \max\{T_1, T_2\}, s \in \mathcal{S}$, we have

$$\left|\theta_{s,a^*(s)}^{(t+k)}\right| = \left|\theta_{s,a^*(s)}^{(t)} + \sum_{i=t}^{t+k-1} \eta \frac{\partial V^{\pi_\theta^{(i)}}(\mu)}{\partial \theta_{s,a^*(s)}}\right| \leq \left|\theta_{s,a^*(s)}^{(t)}\right| + \underbrace{\sum_{i=t}^{\infty} \eta \left|\frac{\partial V^{\pi_\theta^{(i)}}(\mu)}{\partial \theta_{s,a^*(s)}}\right|}_{<\infty, \text{ by } (307)}, \tag{311}$$

$$\left|\theta_{s,a}^{(t+k)}\right| = \left|\theta_{s,a}^{(t)} + \sum_{i=t}^{t+k-1} \eta \frac{\partial V^{\pi_\theta^{(i)}}(\mu)}{\partial \theta_{s,a}}\right| \leq \left|\theta_{s,a}^{(t)}\right| + \underbrace{\sum_{i=t}^{\infty} \eta \left|\frac{\partial V^{\pi_\theta^{(i)}}(\mu)}{\partial \theta_{s,a}}\right|}_{<\infty, \text{ by } (310)}. \tag{312}$$

which leads to the contradiction that $\left\|\theta^{(t)}\right\| \to \infty$ in (300).

Hence, follow from (297), we have

$$V^{\pi_\theta^{**}}(\mu) - V^{\pi_\theta^{(t)}}(\mu) \leq \frac{2LC^2\left(\|\theta^{**}\| + \max_{t' \leq t}\left\|\theta^{(t')}\right\|\right)^2}{t-1} \tag{313}$$

$$\leq \frac{2LC^2\left(\|\theta^{**}\| + \kappa \ln t\right)^2}{t-1}, \tag{314}$$

where $\kappa \in \mathbb{R}$ is a finite constant.

$\square$

**Remark 16.** It is crucial to emphasize that when using PG with softmax parameterization, the parameters cannot exhibit excessive growth. More specifically, we can demonstrate that the growth rate of the parameters remains bounded by $O(\ln t)$. This remarkable bounding property allows us to relax the objective structure to a $C$-nearly concave form while still achieving a convergence rate of $\tilde{O}(1/t)$. However, it is worth noting that due to the unbounded nature of softmax parameterization, it must also compensate for a logarithmic factor in its convergence rate, which may not be tightly aligned with results in the optimization regime.

### F.2 CONVERGENCE RATE OF POLICY GRADIENT

**Theorem 6 (Global convergence for softmax parameterization in (Agarwal et al., 2021)).** *Assume we follow the gradient ascent update rule as specified in (6) and that $\mu(s) > 0$ for all states $s$. Suppose $\eta^{(t)} \leq \frac{(1-\gamma)^3}{8}$, then we have that for all states $s$, $V^{(t)}(s) \to V^*(s)$ as $t \to \infty$.*

**Theorem 7 (Convergence Rate of PG).** *Consider a tabular softmax parameterized policy $\pi_\theta$. Under PG with $\eta^{(t)} = \frac{(1-\gamma)^3}{8}$, there exists a finite time $T$ such that for all $t \geq T$, we have*

$$V^*(\rho) - V^{\pi_\theta^{(t)}}(\rho) \leq \frac{1}{1-\gamma}\left\|\frac{d_\rho^{\pi^*}}{\mu}\right\|_\infty \left(\frac{16C^2(\kappa+2)^2[\ln(t-T)]^2}{(1-\gamma)^3(t-T-1)}\right. \tag{315}$$

$$\left. + \frac{4|\mathcal{S}|}{(1-\gamma)^2}\left(\frac{|\mathcal{A}|-1}{(t-T)^2+|\mathcal{A}|-1}\right)\right), \tag{316}$$

*where $\kappa \in \mathbb{R}$ is a finite constant.*

*Proof of Theorem 7.* By Lemma 1, we know that the objective function $\theta \to V^{\pi_\theta}(\mu)$ is $C$-nearly concave in the directions along the feasible update domain $\mathcal{U}$ when, for all

$s \in \mathcal{S}$ and $a \neq a^*(s)$, $V^{\pi_\theta}(s) > Q^*(s, a_2(s))$, $\theta_{s,a^*(s)} - \theta_{s,a} > M = \max\left\{\ln\left[3\binom{|\mathcal{A}|-1}{2}\right] - 2\ln[d_1 - d_i], \ln\left[\frac{(2\gamma|\mathcal{S}|C-1+\gamma)(|\mathcal{A}|-1)}{1-\gamma}\right]\right\}$. We have to show the following two statements:

- There is a lower bound for the value $d_1 - d_i$ of the update directions such that $M$ is a well-defined finite constant.

- There is a finite time $T$ such that for every update after time $T$, its direction is along the vector in the feasible update domain $\mathcal{U}$.

First of all, by Lemma 28, we have that there is a finite time $T$ such that $\left.\frac{\partial V^{\pi_\theta}(\mu)}{\partial \theta_{s,a^*(s)}}\right|_{\theta=\theta^{(t)}} \geq 0 \geq \left.\frac{\partial V^{\pi_\theta}(\mu)}{\partial \theta_{s,a}}\right|_{\theta=\theta^{(t)}}$, which means that only the gradient of the optimal action is positive. For the above first point, since $\mathbf{d}$ is unit vector, the updates of PG satisfy the summation over the updates of every action is zero, and only the first dimension, i.e., tha update of the optimal action, is positive, we can see that the minimum possible value of $d_1 - d_i$ can only be $\sqrt{\frac{1}{(|\mathcal{A}|-2)^2+|\mathcal{A}|-2}}$ if $|\mathcal{A}| > 2$ or $\frac{1}{\sqrt{2}}$ if $|\mathcal{A}| = 2$, both of which are bounded away from zero. Thus, plugin these values, we can obtain the well-defined finite constant $M$.

Next, the second point is easy to show because after time $T$, only the optimal action update is positive, it directly implies that $d_1$ is the maximum. Thus, PG attains the local $C$-nearly concave regime in a finite time $T$.

By Theorem 5,

$$V^{\pi_{\theta^{**}}^{(t)}}(\mu) - V^{\pi_{\theta}^{(t)}}(\mu) \leq \frac{2LC^2\left(\left\|\theta^{**(t)}\right\| + \kappa\ln(t-T)\right)^2}{(t-T)-1} \tag{317}$$

$$\leq \frac{16C^2\left(2\ln(t-T) + \kappa\ln(t-T)\right)^2}{(1-\gamma)^3(t-T-1)} \tag{318}$$

$$= \frac{16C^2(\kappa+2)^2[\ln(t-T)]^2}{(1-\gamma)^3(t-T-1)}, \tag{319}$$

where $\theta^{(t)**} := [2\ln(t-T), 0, 0, \cdots, 0]$ is a chosen surrogate optimal solution at time $t$. Additionally, we have the sub-optimality gap between the original optimal solution and the surrogate optimal solution can be bounded as

$$V^*(\mu) - V^{\pi_{\theta^{**}}^{(t)}}(\mu) = \frac{1}{1-\gamma}\sum_s d_\mu^{\pi^*}(s)\sum_a (\pi^*(a|s) - \pi_{\theta^{**}}^{(t)}(a|s)) \cdot A^{\pi_{\theta^{**}}^{(t)}}(s,a) \tag{320}$$

$$\leq \frac{4|\mathcal{S}|}{(1-\gamma)^2}\left(1 - \frac{\exp(2\ln(t-T))}{\exp(2\ln(t-T)) + \exp(0)\cdot(|\mathcal{A}|-1)}\right) \tag{321}$$

$$= \frac{4|\mathcal{S}|}{(1-\gamma)^2}\left(\frac{|\mathcal{A}|-1}{(t-T)^2+|\mathcal{A}|-1}\right), \tag{322}$$

where (321) consider the upper bound of $A^{\pi_{\theta^{**}}^{(t)}}(s,a) \leq \frac{2}{1-\gamma}$ and the sum of differences between sub-optimal actions can be at most the difference between the optimal actions, which causes a pre-constant 2. By Lemma 11, we can change the convergence rate from $V^\pi(\mu)$ to $V^\pi(\rho)$,

$$V^*(\rho) - V^{\pi_{\theta}^{(t)}}(\rho) \leq \frac{1}{1-\gamma}\left\|\frac{d_\rho^{\pi^*}}{\mu}\right\|_\infty \left(\frac{16C^2(\kappa+2)^2[\ln(t-T)]^2}{(1-\gamma)^3(t-T-1)}\right. \tag{323}$$

$$\left.+ \frac{4|\mathcal{S}|}{(1-\gamma)^2}\left(\frac{|\mathcal{A}|-1}{(t-T)^2+|\mathcal{A}|-1}\right)\right), \tag{324}$$

and we complete the proof. $\qquad\square$

**Lemma 28.** *Consider a tabular softmax parameterized policy $\pi_\theta$. Under PG with $\eta^{(t)} = \frac{(1-\gamma)^3}{8}$, given any $M > 0$, there exists a finite time $T$ such that for all $t \geq T$, $s \in \mathcal{S}$, and $a \neq a^*(s)$, we have*

- $\theta_{s,a^*(s)} - \theta_{s,a} > M,$

- $V^{\pi_\theta^{(t)}}(s) > Q^*(s, a_2(s)),$

- $\frac{\partial V^{\pi_\theta}(\mu)}{\partial \theta_{s,a^*(s)}}\bigg|_{\theta=\theta^{(t)}} \geq 0 \geq \frac{\partial V^{\pi_\theta}(\mu)}{\partial \theta_{s,a}}\bigg|_{\theta=\theta^{(t)}}.$

*Proof of Lemma 28.* By Assumption 3, there is a unique optimal action, so Theorem 6 implies that the policy weight probability of the optimal action approaches 1. For $\theta_{s,a^*(s)} - \theta_{s,a} > M$, since we also have the asymptotic convergence property, we can see that the same strategy works here as the proof shown in Lemma 2.

Moreover, by Lemma 12, we can show the second point to imply the third point. Regarding the second point, it is the direct result of Theorem 6 because $V^{(t)}$ can always enter a small enough neighborhood such that the values always are greater than $Q^*(s, a_2(s))$. Hence, we obtain the desired results. $\qquad\square$

## G CHALLENGES OF APG WITHOUT MOMENTUM RESTART MECHANISMS

In this section, we provide insights into the challenges encountered when applying APG without momentum restart mechanisms. Furthermore, we've established the convergence rate of $\tilde{O}(1/t^2)$ under APG without restart mechanisms in the bandit setting. More specifically, In Appendix G.1, we introduce the algorithm, NAPG, which is the original APG without restart mechanisms. Subsequently, in Appendix G.2, we provide a numerical validation in the bandit setting to illustrate the non-monotonic improvement of NAPG. Additionally, in Appendix G.3, we illustrate the challenges of establishing the limiting value functions in the MDPs setting.

### G.1 APG WITHOUT RESTART MECHANISMS (NAPG)

For ease of exposition, this subsection presents the algorithm for Nesterov Accelerated Policy Gradient (NAPG) without restart mechanisms in Algorithm 6. As depicted in Algorithm 6. It's worth noting that without these restart mechanisms, Algorithm 6 aligns precisely with the original Nesterov acceleration method outlined in (Nesterov, 1983).

---

**Algorithm 6** Nesterov Accelerated Policy Gradient (NAPG) *Without Restart Mechanisms*

---

**Input**: Learning rate $\eta^{(t)} > 0$.
**Initialize**: $\theta^{(0)} \in \mathbb{R}^{|\mathcal{S}||\mathcal{A}|}$, $\tau^{(0)} = 0$, $\omega^{(0)} = \theta^{(0)}$.
**for** $t = 1$ to $T$ **do**

$$\theta^{(t)} \leftarrow \omega^{(t-1)} + \eta^{(t)} \nabla_\theta V^{\pi_\theta}(\mu)\Big|_{\theta = \omega^{(t-1)}} \tag{325}$$

$$\omega^{(t)} \leftarrow \theta^{(t)} + \frac{t-1}{t+2}(\theta^{(t)} - \theta^{(t-1)}) \tag{326}$$

**end for**

---

### G.2 NUMERICAL VALIDATION OF THE NON-MONOTONIC IMPROVEMENT UNDER NAPG

In this subsection, we illustrate the difficulties involved in analyzing the convergence of NAPG, compared to the standard policy gradient methods through a numerical experiment. In contrast to the standard policy gradient (PG) method, which exhibits monotonic improvement, NAPG could experience non-monotonic progress as a result of the momentum term, which could lead to negative performance changes. To further demonstrate this phenomenon, we conduct a 3-action bandit experiment with a highly sub-optimal initialization, where the weight of the optimal action of the initial policy is extremely small. More specifically, we conduct a 3-action bandit experiment with actions $\mathcal{A} = [a^*, a_2, a_3]$, where the corresponding rewards are $r = [r(a^*), r(a_2), r(a_3)] = [1, 0.8, 0]$. We initialize the policy parameters as $\theta^{(0)} = [0, 3, 10]$, which

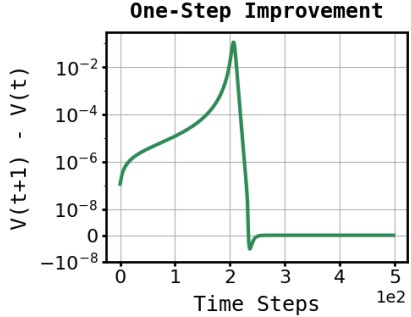

Figure 3: The one-step improvement of APG on a three-action bandit problem.

represents a highly sub-optimal initialization. Remarkably, the weight of the optimal action in the initial policy $\pi^{(0)} \approx [0.00005, 0.00091, 0.99904]$ is exceedingly small. As shown in Figure 3, the one-step improvement becomes negative around epoch 180 and provides nearly zero improvement after that point. Notably, the asymptotic global convergence of the standard PG is largely built on the monotonic improvement property, as shown in (Agarwal et al., 2021). With that said, the absence of monotonic improvement in NAPG poses a fundamental challenge in analyzing and achieving global convergence to an optimal policy.

### G.3 CHALLENGES OF ESTABLISHING THE LIMITING VALUE FUNCTIONS IN THE MDPs SETTING UNDER NAPG

In this subsection, we illustrate the challenges involved in establishing the limiting value functions under NAPG in the MDPs setting. Recall from Appendix G.2 that NAPG is not guaranteed to achieve monotonic improvement in each iteration due to the momentum. This is one salient difference from the standard PG, which inherently enjoys strict improvement and hence the existence of the limiting value functions (i.e., $\lim_{t\to\infty} V^{\pi_{\theta}(t)}(s)$) by Monotone Convergence Theorem (Agarwal et al., 2021). Without monotonicity, it remains unknown if the limiting value functions even exist.

Furthermore, while the gradient step ensures a guaranteed monotonic improvement, which can help counteract the non-monotonicity resulting from the momentum step, it is still a challenging task to quantify this non-monotonic behavior due to the cumulative effect of the momentum. To be more specific, we demonstrate that the influence exerted by the momentum term can lead to a substantial and challenging-to-control effect.

Let $\delta^{(T_0)} := \theta_{s,a}^{(T_0)} - \theta_{s,a}^{(T_0-1)}$, by the update rule of NAPG, we have

$$\theta_{s,a}^{(T_0+1)} = \theta_{s,a}^{(T_0)} + \frac{T_0-1}{T_0+2}\delta^{(T_0)} + \eta^{(T_0+1)} \cdot \frac{\partial V^{\pi_\theta}(\mu)}{\partial \theta_{s,a}}\Big|_{\theta=\omega^{(T_0)}} \tag{327}$$

$$\theta_{s,a}^{(T_0+2)} = \theta_{s,a}^{(T_0+1)} + \frac{T_0}{T_0+3}\delta^{(T_0+1)} + \eta^{(T_0+2)} \cdot \frac{\partial V^{\pi_\theta}(\mu)}{\partial \theta_{s,a}}\Big|_{\theta=\omega^{(T_0+1)}} \tag{328}$$

$$= \theta_{s,a}^{(T_0)} + \frac{T_0-1}{T_0+2}\delta^{(T_0)} + \eta^{(T_0+1)} \cdot \frac{\partial V^{\pi_\theta}(\mu)}{\partial \theta_{s,a}}\Big|_{\theta=\omega^{(T_0)}} \tag{329}$$

$$+ \frac{T_0}{T_0+3}\left(\frac{T_0-1}{T_0+2}\delta^{(T_0)} + \eta^{(T_0+1)} \cdot \frac{\partial V^{\pi_\theta}(\mu)}{\partial \theta_{s,a}}\Big|_{\theta=\omega^{(T_0)}}\right) \tag{330}$$

$$+ \eta^{(T_0+2)} \cdot \frac{\partial V^{\pi_\theta}(\mu)}{\partial \theta_{s,a}}\Big|_{\theta=\omega^{(T_0+1)}} \tag{331}$$

$$= \theta_{s,a}^{(T_0)} + \left(\frac{T_0-1}{T_0+2} + \frac{T_0}{T_0+3}\frac{T_0-1}{T_0+2}\right) \cdot \delta^{(T_0)} \tag{332}$$

$$+ \left(1 + \frac{T_0}{T_0+3}\right) \cdot \eta^{(T_0+1)} \cdot \frac{\partial V^{\pi_\theta}(\mu)}{\partial \theta_{s,a}}\Big|_{\theta=\omega^{(T_0)}} \tag{333}$$

$$+ \eta^{(T_0+2)} \cdot \frac{\partial V^{\pi_\theta}(\mu)}{\partial \theta_{s,a}}\Big|_{\theta=\omega^{(T_0+1)}} \tag{334}$$

$$\theta_{s,a}^{(T_0+M)} = \theta_{s,a}^{(T_0)} + \left(\frac{T_0-1}{T_0+2} + \frac{T_0}{T_0+3}\frac{T_0-1}{T_0+2} + \sum_{\tau=3}^{M}\frac{T_0+1}{T_0+\tau+1}\frac{T_0}{T_0+\tau}\frac{T_0-1}{T_0+\tau-1}\right) \cdot \delta^{(T_0)} \tag{335}$$

$$+ \cdots \tag{336}$$

where (335) holds by expanding (327)-(334) iteratively.

Note that for large enough $M \in \mathbb{N}$,

$$\sum_{\tau=3}^{M}\frac{(T_0+1)T_0(T_0-1)}{(T_0+\tau+2)(T_0+\tau+1)(T_0+\tau)} \tag{337}$$

$$= (T_0+1)T_0(T_0-1)\sum_{\tau=3}^{M}\frac{1}{2}\left(\frac{1}{(T_0+\tau)(T_0+\tau+1)} - \frac{1}{(T_0+\tau+1)(T_0+\tau+2)}\right) \tag{338}$$

$$= (T_0+1)T_0(T_0-1)\cdot\frac{1}{2}\left(\frac{1}{(T_0+3)(T_0+4)} - \frac{1}{(T_0+M+1)(T_0+M+2)}\right) \tag{339}$$

$$= \Theta(T_0). \tag{340}$$

Follow from (335) and (340), we could deduce that the momentum term $\delta^{(T_0)}$ in time $T_0$ results in an effect that is directly proportional to $T_0$. Therefore, it becomes crucial to establish the upper bound

of the gradient norm to confine the adverse effects stemming from the momentum term. However, this endeavor involves addressing the intertwined challenge of estimating both the gradient norm and the momentum norm simultaneously.

# H    ADDITIONAL EXPERIMENTS

## H.1    STOCHASTIC APG (SAPG)

In this subsection, we conduct an empirical evaluation of the performance of Stochastic Accelerated Policy Gradient (SAPG) on an MDP with 5 states and 5 actions, utilizing the true gradient. In the subsequent experimental results, we set the batch size to $B = 1$ and perform the experiments with 50 different seeds. It's important to highlight that the MDP used is identical to the one discussed in Section 6.1, and detailed configuration information is provided in Appendix E.

### H.1.1    IMPLEMENTATION DETAIL

For ease of exposition, we provide the pseudo code for Stochastic Accelerated Policy Gradient (SAPG).

---

**Algorithm 7** Stochastic Accelerated Policy Gradient (SAPG)

**Input**: Learning rate $\eta^{(t)} > 0$, batch size $B \in \mathbb{N}$.

**Initialize**: $\theta^{(0)} \in \mathbb{R}^{|\mathcal{S}||\mathcal{A}|}$, $\tau^{(0)} = 0$, $\omega^{(0)} = \theta^{(0)}$.

**for** $t = 1$ to $T$ **do**

Sample a batch of $\mathcal{S}^{(t)} \times \mathcal{A}^{(t)} \coloneqq \{(s_1^{(t)}, a_1^{(t)}), (s_2^{(t)}, a_2^{(t)}), \ldots, (s_B^{(t)}, a_B^{(t)})\}$ where state $s^{(t)}$ is sampled with probability $d_\mu^{\pi^{(t)}}(\cdot)$ and action $a^{(t)}$ is sampled with probability $\pi^{(t)}(\cdot|s^{(t)})$.

Calculate stochastic gradient $v^{(t)} = \sum_{s,a \in \mathcal{S}^{(t)} \times \mathcal{A}^{(t)}} \left. \frac{\partial V^{\pi_\theta}(\mu)}{\partial \theta_{s,a}} \right|_{\theta = \omega^{(t-1)}}$.

$$\theta^{(t)} \leftarrow \omega^{(t-1)} + \eta^{(t)} v^{(t)} \tag{341}$$

$$\varphi^{(t)} \leftarrow \theta^{(t)} + \frac{t-1}{t+2}(\theta^{(t)} - \theta^{(t-1)}) \tag{342}$$

$$\omega^{(t)} \leftarrow \begin{cases} \varphi^{(t)}, & \text{if } V^{\pi_\varphi^{(t)}}(\mu) \geq V^{\pi_\theta^{(t)}}(\mu), \\ \theta^{(t)}, & \text{otherwise.} \end{cases} \tag{343}$$

**end for**

---

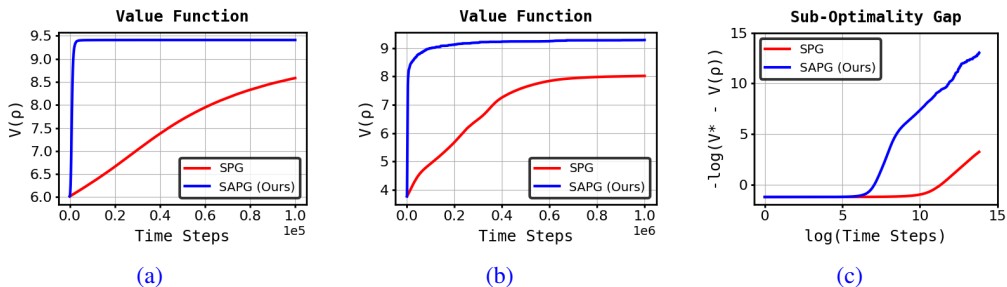

Figure 4: A comparison between the performance of SAPG and SPG under an MDP with 5 states, 5 actions, with the uniform and hard policy initialization: (a)-(b) show the value function under the uniform and the hard initialization, respectively. The optimal objective value $V^*(\rho) \approx 9.41$; (c) show the sub-optimality gaps under uniform initialization.

### H.1.2    SAPG UNDER AN MDP WITH 5 STATES AND 5 ACTIONS

In Figure 4, it is evident that SAPG outperforms Stochastic PG (SPG) with a remarkable speed of convergence. Furthermore, the results presented in Figure 4(c) suggest that SAPG may exhibit a convergence rate of $O(\frac{1}{t^\alpha})$, where $2 > \alpha > 1$. This observation opens up the possibility of

extending and exploring this potential in future research. Additionally, it's noteworthy that under hard policy initialization (where the optimal action has the smallest initial probability), SPG tends to get stuck at a local optimum for an extended period, whereas SAPG does not exhibit this issue.

## H.2    APG IN THE GYM ENVIRONMENTS

In this subsection, we empirically evaluate the performance of APG across various benchmark RL tasks. The evaluation is carried out in Gym Environments (Brockman et al., 2016), encompassing two distinct types of environments: *Lunar Lander* and *Bipedal Walker*. Notably, in Lunar Lander, the state space is *continuous*, and the action space is *discrete*, whereas in Bipedal Walker, both the state space and action space are *continuous*.

### H.2.1    IMPLEMENTATION DETAIL

Our implementation is based on the code base Spinningup for (Achiam, 2018) for APG, HBPG and PG. The hyperparameters are listed in Table 9 and Table 10.

**Remark 17.** *It's worth nothing that the learning rate of PG is greater than the one of APG and HBPG. The learning rate displayed in the table represents the optimal performance selected from the set* $\{0.3, 0.1, 0.03, 0.01, 0.003, 0.001, 0.0007, 0.0005, 0.0003, 0.0001\}$.

Table 9: Hyperparameters for APG, HBPG and PG in LunarLander-v2.

| HYPERPARAMETERS | APG | HBPG | PG |
|---|---|---|---|
| EPOCH | 1100 | 1100 | 1100 |
| GAMMA (DISCOUNT FACTOR) | 0.999 | 0.999 | 0.999 |
| ITERATION OF TRAINING VALUE FUNCTION | 80 | 80 | 80 |
| LAMBDA FOR GAE | 0.97 | 0.97 | 0.97 |
| LEARNING RATE FOR POLICY OPTIMIZER | **0.0007** | **0.0007** | **0.03** |
| LEARNING RATE FOR VALUE FUNCTION OPTIMIZER | 0.001 | 0.001 | 0.001 |
| NUMBER OF STEPS OF INTERACTION IN EACH EPOCH | 4000 | 4000 | 4000 |
| MLP ACTOR-CRITIC HIDDEN LAYER | [32, 32] | [32, 32] | [32, 32] |
| MAXIMUM LENGTH OF THE TRAJECTORY | 1000 | 1000 | 1000 |

Table 10: Hyperparameters for APG, HBPG and PG in BipedalWalker-v3.

| HYPERPARAMETERS | APG | HBPG | PG |
|---|---|---|---|
| EPOCH | 5000 | 5000 | 5000 |
| GAMMA (DISCOUNT FACTOR) | 0.999 | 0.999 | 0.999 |
| ITERATION OF TRAINING VALUE FUNCTION | 80 | 80 | 80 |
| LAMBDA FOR GAE | 0.97 | 0.97 | 0.97 |
| LEARNING RATE FOR POLICY OPTIMIZER | **0.0005** | **0.0005** | **0.03** |
| LEARNING RATE FOR VALUE FUNCTION OPTIMIZER | 0.001 | 0.001 | 0.001 |
| NUMBER OF STEPS OF INTERACTION IN EACH EPOCH | 4000 | 4000 | 4000 |
| MLP ACTOR-CRITIC HIDDEN LAYER | [32, 32] | [32, 32] | [32, 32] |
| MAXIMUM LENGTH OF THE TRAJECTORY | 1600 | 1600 | 1600 |

### H.2.2    LUNARLANDER-V2 AND BIPEDALWALKER-V3

Based on the findings in Figure 5-6, it is evident that APG exhibits superior performance. While HBPG demonstrates competitiveness with APG, it is crucial to highlight that HBPG may encounter instability in the latter half of the training process.

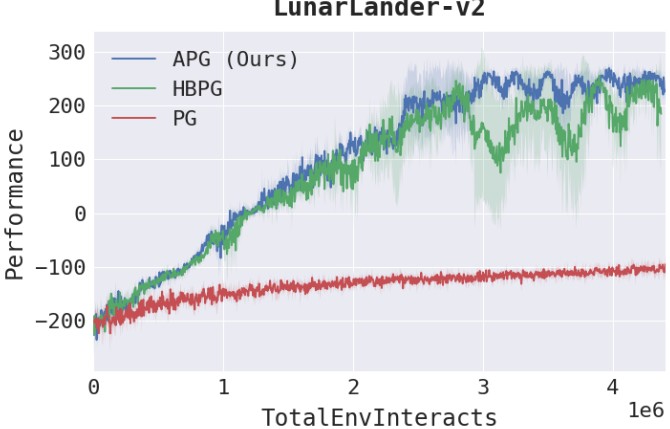

Figure 5: A comparison between the performance of APG and other benchmark methods algorithms in Lunar Lander. All the results are averaged over 3 random seeds (with the shaded area showing the range of mean ± std).

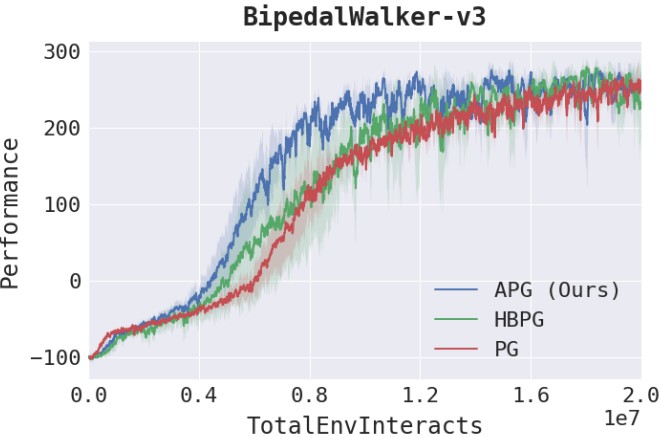

Figure 6: A comparison between the performance of APG and other benchmark methods algorithms in Bipedal Walker. All the results are averaged over 3 random seeds (with the shaded area showing the range of mean ± std).

## I   A DETAILED COMPARISON OF THE ACCELERATION METHODS FOR RL

In the optimization literature, there are several major categories of acceleration methods, including momentum, regularization, natural gradient, and normalization. Interestingly, these approaches all play an important role in the context of RL.

### I.1   MOMENTUM

**Empirical Aspect**: In practice, momentum is one of the most commonly used approaches for acceleration in policy optimization for RL, mainly due to its simplicity. Specifically, a variety of classic momentum methods (e.g., Nesterov momentum, heavy-ball method, and AdaGrad) have been already implemented in optimization solvers and deep learning frameworks (e.g., PyTorch) as a basic machinery and shown to improve the empirical convergence results than the standard policy gradient methods like A2C and PPO (e.g., see (Henderson et al., 2018)).

**Theoretical Aspect**: However, despite the popularity of these approaches in empirical RL, it has remained largely unknown whether momentum could indeed improve the theoretical global convergence rates of RL algorithms. To address this fundamental question, we establish the very first global convergence rate of Nesterov momentum in RL (termed APG in our paper) and show that APG could improve the rate of PG from $O(1/t)$ to $\tilde{O}(1/t^2)$ (in the constant step-size regime). Given the wide application of momentum, our analytical framework and the insights from our convergence result (e.g., local near-concavity) serve as an important first step towards better understanding the momentum approach in RL.

**Theoretical Advantage**: In addition, we would like to highlight that in the constant step-size regime (which is one of the most widely adopted setting in practice), our proposed APG indeed achieves a superior global convergence rate of $\tilde{O}(1/t^2)$, which is better than the $O(1/t)$ rate of vanilla PG and the $O(1/t)$ rate of NPG under constant step sizes.

## I.2 REGULARIZATION

**Theoretical Aspect**: In the convex optimization literature, it is known that adding a strongly convex regularizer achieves acceleration by changing the optimization landscape. In the context of RL, despite the non-concave objective, regularization has also been shown to achieve faster convergence, such as the log-barrier regularization with $O(1/\sqrt{t})$ rate (Agarwal et al., 2021) and the entropy regularization (i.e., adding a policy entropy bonus to the reward) with linear convergence rate $O(e^{-ct})$ (Mei et al., 2020), both in the exact gradient setting with constant step sizes.

Despite the above convergence results, one common attack on regularization approaches is that they essentially change the objective function to that of regularized MDPs and hence do not directly address the original objective in unregularized RL, as pointed out by (Mei et al., 2021b; Xiao, 2022).

## I.3 NATURAL GRADIENTS (A SPECIAL CASE OF POLICY MIRROR DESCENT)

Natural policy gradient (NPG), another RL acceleration technique, borrows the idea from the natural gradient, which uses the inverse of Fisher information matrix as the preconditioner of the gradient and can be viewed as achieving approximate steepest descent in the distribution space (Amari, 1998; Kakade, 2001). Moreover, under direct policy parameterization, NPG is also known as a special case of policy mirror descent (PMD) with KL divergence as the proximal term (Shani et al., 2020; Xiao, 2022). Regarding the global convergence rates, NPG has been shown to achieve: (i) $O(1/t)$ rate under constant step sizes (Agarwal et al., 2021; Xiao, 2022); (ii) linear convergence either under adaptively increasing step sizes (Khodadadian et al., 2021) or non-adaptive geometrically increasing step sizes (Xiao, 2022). Notably, under direct policy parameterization and geometrically increasing step sizes, similar linear convergence results can also be established for the more general PMD method (Xiao, 2022).

**Empirical Issues**: Despite the above convergence rates, as also pointed out by (Mei et al., 2021b), it is known that NPG in general requires solving a costly optimization problem in each iteration (cf. (Kakade, 2001; Agarwal et al., 2021)) even with the help of compatible approximation theorem (Kakade, 2001) (unless the policy parametrization adopts some special forms, e.g., log-linear policies (Yuan et al., 2022)). Notably, the TRPO algorithm, a variant of NPG, is known to suffer from this additional computational overhead. Similarly, the PMD in general also requires solving a constrained optimization problem in each policy update (Xiao, 2022). This computational complexity could be a critical factor for RL in practice.

## I.4 NORMALIZATION

**Theoretical Aspect**: In the exact gradient setting, the normalization approach, which exploits the non-uniform smoothness by normalizing the true policy gradient by its L2 gradient norm, has also been shown to achieve acceleration in RL. Specifically, the geometry-aware normalized (GNPG) has been shown to exhibit linear convergence rate under softmax policies (Mei et al., 2021b). This rate could be largely attributed to the effective increasing step size induced by the normalization technique.

**Empirical Issues**: Despite the linear convergence property, it has been already shown that GNPG could suffer from divergence in the standard on-policy stochastic setting, even in simple 1-state MDPs. This phenomenon is mainly due to the committal behavior induced by the increasing effective step size. This behavior could substantially hinder the use of GNPG in practice.

