# OpenReview forum: "Accelerated Policy Gradient: On the Nesterov Momentum for Reinforcement Learning"
_ICLR.cc/2024/Conference — Submitted to ICLR 2024_

### Official Review · Reviewer_y1oq · 2023-10-31

**Soundness:** 2 fair
**Presentation:** 3 good
**Contribution:** 2 fair
**Rating:** 5
**Confidence:** 3

**Summary:**

This paper presents a novel RL algorithm based on Nesterov Accelerated Gradient (NAG) to improve the convergence rate. Specifically, the authors adapt the NAG into policy gradient to develop APG and mathematically show the global convergence rate of $\tilde{\mathcal{O}}(1/t^2)$, which is faster than the existing $\mathcal{O}(1/t)$., with the true gradient and softmax policy parameterization. The authors also show that regardless of initialization, APG is able to reach a locally nearly-concave regime, within finite iterations. To validate the theory, the authors use two simple benchmarks to demonstrate that APG outperforms the standard PG.

**Strengths:**

The investigated topic of convergence rate improvement for RL is important and interesting. This work seems theoretically strong in analysis by combining two well-known methods, NAG and PG to improve the rate. Such a combination is simple and straightforward. The paper is well written and easy to follow.

**Weaknesses:**

1. The novelty is incremental. APG is not novel in terms of the algorithmic framework.

2. Some assumptions to characterize the analysis are strong and not justified well. Why is Assumption 1 required to guarantee the convergence? Please justify in the paper. Otherwise, it might be a strong condition in the work. How likely is the assumption satisfied in the various real-world scenarios? What is the point to have Assumption 2? Not a more generic range [-R, R] in many existing works? Why is Assumption 4 is required for the convergence? I understand it could be attained in practice.

3. In Theorem 1, the authors mentioned that the softmax parameterized policy is tabular. What does it mean by tabular here? Would it mean the policy acts like a lookup table? Do the conclusions still apply if removing tabular? It is confusing in the paper.

4. The experimental results are not promising. The benchmark models are quite simple. The authors should present more complex benchmark models. Continuous environment should be utilized to showcase APG's superiority.

5. Definitions 1 and 2 are a bit ah-hoc for the convergence proof in this work. If they are existing, the authors should cite references. Otherwise, the author should justify why they are needed.

6. Section 6.2 did not really present anything new on the lower bound. How to relate Theorem 3 to APG? The authors only said due to Theorem 3, there was no tighter lower bound for APG. This seems to me quite simple. They should show the contradiction if there existed a tighter lower bound for APG.

I think the paper still requires a substantial amount of work to make it technically solid and sound.

************************Post-rebuttal*****************************
Thanks the authors for addressing my comments and revising the paper with additional results. I really appreciate that. After carefully reviewing the rebuttal and comments from other reviewers, I have raised my score. While the assumptions to me are a still a bit strong in this work.

**Questions:**

Please see the questions in the weaknesses.

---

> ### Author Response · Authors · 2023-11-20
> **Response to Reviewer y1oq (Part 1)**
>
> **We thank the reviewer for the detailed comments and helpful suggestions.**
>
> ### **Q. Highlight the novelty of this work.**
> We would like to highlight the novelty and the contributions of this paper as follows:
>
> - One major contribution is the first global convergence rate of Nesterov acceleration in the context of RL. Despite the fact that APG is built on the classic Nesterov momentum and is not a completely new algorithm, it is rather remarkable that this method still achieves a convergence rate of $\tilde{O}(1/t^2)$ under the complex, non-concave RL objective (The non-concavity has been demonstrated in [Mei et al., 2020]). Notably, addressing non-concavity poses challenges in establishing global convergence and convergence rates under Nesterov acceleration (compared to [Ghadimi et al., 2017; Carmon el at., 2018] that have only achieved convergence to a stationary point). Despite this, this paper successfully establishes such a rate.
>
> - Notably, in the constant step-size regime (which is one of the most widely adopted setting in practice), we establish that APG indeed achieves a superior global convergence rate of $\tilde{O}(1/t^2)$, which is better than the $O(1/t)$ rate of vanilla PG and the $O(1/t)$ rate of NPG under constant step sizes.
>
> - Additionally, we would like to underscore the significance of the local near-concavity property (cf. Lemma 1), a novel and important finding that is not present in the existing literature. Over the decades, despite the adaptation of various acceleration methods to RL, the non-concavity of the RL objective has deterred comprehensive analyses of the underlying convergence behavior. This lemma unveils the local near-concavity of the complex RL objective, and this important structural property could naturally spark future analysis in more RL settings (e.g., stochastic settings or under function approximation).
>
> &nbsp;
>
> ### **Q. Clarification on the assumptions**
> We would like to clarify the general applicability of Assumptions 1, 2, and 4 mentioned by the reviewer.
>
> - To begin with, Assumption 1, which requires the strict positivity of the surrogate initial state distribution $\mu$, is a standard assumption adopted in the existing RL literature on the global convergence of policy gradient methods [Agarwal et al., 2020; Ding et al., 2020; Mei et al., 2020; Mei et al., 2021 Xiao, 2022]. Moreover, we would like to emphasize that the surrogate initial distribution $\mu$, which is different from the true initial distribution $\rho$, is a parameter that could be freely chosen by the RL algorithm. With the help of the surrogate distribution $\mu$, the policy gradient methods (e.g., PG and APG) could work without the knowledge about the true distribution $\rho$, as also indicated and adopted by the existing works (e.g., [Agarwal et al., 2020; Mei et al., 2020; Xiao, 2022])
>
>
> - Regarding Assumption 2, specifying the range of the reward serves to simplify the pre-constant in the convergence rate. For instance, assuming the reward range to be $[-R, R]$ introduces a factor of $2|R|$ in the pre-constant. It is noteworthy that altering the reward range can be interpreted as applying a linear transformation from the original $[0, 1]$ range and hence preserve the fundamental properties of the MDP environment (e.g., an optimal policy remains optimal after such transformation) and the associated lemma/theorem. Therefore, Assumption 2 could be adopted without any loss of generality. This assumption, common in the RL literature, aligns with the standard assumptions used in the existing works on the convergence of policy gradient methods (e.g., [Agarwal et al., 2020; Ding et al., 2020; Mei et al., 2020; Ding et al., 2022; Xiao, 2022]).
>
> - Regarding Assumption 4, it can be easily met in practice (as acknowledged by the reviewer) and is employed only to ensure the existence of the limiting value vector $V^{\pi^{(t)}}$ under APG. Based on the existence of the limiting $V^{\pi^{(t)}}$, we proceed to establish the asymptotic global convergence of APG, which is a highly non-trivial result due to the momentum (cf. Appendix C) and an important building block of the convergence rate analysis, by considering the actions with positive, negative, and zero advantage value in the limit. The detailed usage of Assumption 4 is discussed in Lemma 15 (page 24).

---

> ### Author Response · Authors · 2023-11-20
> **Response to Reviewer y1oq (Part 2)**
>
> ### **Q. The softmax tabular parameterization.**
> The softmax tabular parameterization, extensively used and discussed in the RL literature (e.g., [Szepesvári et al., 2022]), serves as a method to express a policy in a well-defined tabular format (and hence we follow the convention by using the term "tabular"). This representation involves storing and manipulating the policy explicitly in a table, where each row is associated with a state, and each column is associated with each possible action.
>
> Moreover, it is important to note that tabular parameterization is a commonly used scheme in the convergence rate analysis of policy gradient methods (e.g., [Agarwal et al., 2020; Mei et al., 2020; Xiao 2022].
>
> &nbsp;
>
> ### **Q. Empirical performance of APG in more complex benchmark environments**
> To begin with, we would like to first highlight that the main focus of this paper is the theoretical convergence rate of APG, and the experimental results in small-scale MDPs in Figures 1-2 are included mainly to validate the $\tilde{O}(1/t^2)$ convergence rate.
>
> On the other hand, as suggested by the reviewer, we further evaluate the performance of APG beyond the true gradient setting in more diverse RL environments. Specifically, we extend APG to the stochastic setting (please see Algorithm 7 in Appendix H for the pseudo code) and test the stochastic APG in LunarLander-v2 and BipedalWalker-v3 using the estimated gradients. The detailed experimental results and the learning curves can be found in Appendix H in the revised manuscript. At a glance, the following table highlights that APG consistently outperforms HBPG and PG in terms of convergence behavior in both LunarLander-v2 and BipedalWalker-v3. These results showcase the empirical superiority of the APG method.
>
> | Total Environment Interactions / Algorithms | LunarLander-v2 (1.5e6 steps) | LunarLander-v2 (2.5e6 steps) | LunarLander-v2 (4e6 steps) | BipedalWalker-v3 (6e6 steps) | BipedalWalker-v3 (1.2e7 steps) | BipedalWalker-v3 (2e7 steps) |
> |---------------------------------------------|:----------------------------:|:----------------------------:|:--------------------------:|:----------------------------:|:------------------------------:|------------------------------|
> |           **APG (mean / std dev)**          |    **59.41709 / 28.38159**   |   **177.36543 / 82.97050**   |   **235.86339 / 3.75312**  |   **115.29717 / 51.89256**   |    **254.83298 / 19.24260**    |   **249.83565 / 28.13334**   |
> |            HBPG (mean / std dev)            |      27.76393 / 20.25072     |     180.82551 / 56.37636     |    151.22692 / 83.93418    |      55.92438 / 113.5947     |      216.47174 / 70.40871      |     232.24193 / 59.57783     |
> |             PG (mean / std dev)             |     -144.25341 / 25.66134    |     -128.69148 / 13.92372    |    -107.66071 / 3.56015    |      1.47716 / 23.27089      |      191.22486 / 20.54212      |     249.27057 / 12.74333     |
>
> **(The results reported above are the averaged over 3 random seeds.)**

---

> ### Author Response · Authors · 2023-11-20
> **Response to Reviewer y1oq (Part 3)**
>
> ### **Q. Elucidating the need for introducing Definitions 1 and 2**
> We thank the reviewer for the suggestion. Here, we highlight the motivation for introducing Definitions 1 and 2, underscoring the innovative nature of Definition 1. In comparing our $C$-near concavity with traditional concavity, we accentuate that $C$-near concavity entails less stringent conditions than classic concavity. The differentiation emerges due to the inclusion of the tolerance constant $C$ in our definition of concavity. This signifies that there exists a $C$ times larger tolerance than the original concavity, rendering $C$-near concavity a more flexible and adaptable concept. For a more detailed discussion, please refer to our paper, particularly on page 7, where we have provided additional explanations on these definitions.
>
> Regarding Definition 2, this definition is to characterize a set of directions that are effective in enhancing policy performance under APG. Specifically, this definition is meant to characterize the update directions that maximize the increase in the probability of the optimal action. Under this definition, we show that the RL objective is $C$-nearly concave in these directions.
>
> In summary, both Definitions 1 and 2 constitute novel contributions in our work. Beyond their effectiveness in our specific settings and proofs, Definition 1 introduces a less stringent property for the  RL objective function, paving the way for further research. Moreover, Definition 2 serves as a characterization of update directions for policy improvement under APG.
>
> &nbsp;
>
> ### **Q. Elucidating the motivation for introducing Section 6.2**
> Section 6.2 aims to underscore the absence of contradiction with the lower bound established in [Mei et al., 2020]. This is attributed to APG's ability to traverse a larger region of the domain than what PG can reach, elucidating why no contradiction exists.
>
> &nbsp;
>
> ### **References**
> - [Mei et al., 2020] Jincheng Mei, Chenjun Xiao, Csaba Szepesvari, and Dale Schuurmans, “On the Global Convergence Rates of Softmax Policy Gradient Methods,” ICML 2020.
> - [Agarwal et al., 2020] Alekh Agarwal, Sham M Kakade, Jason D Lee, and Gaurav Mahajan, “On the Theory of Policy Gradient Methods: Optimality, Approximation, and Distribution Shift,” COLT 2020.
> - [Ghadimi et al., 2017] Saeed Ghadimi and Guanghui Lan. “Accelerated gradient methods for nonconvex nonlinear and stochastic programming,” Mathematical Programming 2016.
> - [Carmon el at., 2018] Yair Carmon, John C Duchi, Oliver Hinder, and Aaron Sidford. “Accelerated methods for nonconvex optimization,” SIAM Journal on Optimization 2018.
> - [Szepesvári et al., 2022] Csaba Szepesvári, “Algorithms for reinforcement learning,” Springer Nature 2022.
> - [Xiao, 2022] Lin Xiao, “On the Convergence Rates of Policy Gradient Methods,” JMLR 2022.
> - [Ding et al., 2020] Ding, Dongsheng, et al. “Natural policy gradient primal-dual method for constrained markov decision processes,” NeurIPS 2020.
> - [Ding et al., 2022] Ding, Yuhao, Junzi Zhang, and Javad Lavaei. “On the global optimum convergence of momentum-based policy gradient,” International Conference on Artificial Intelligence and Statistics. PMLR 2022.
> - [Mei et al., 2021] Jincheng Mei, et al. "Leveraging non-uniformity in first-order non-convex optimization," ICML 2021.

---

### Official Review · Reviewer_txPi · 2023-11-07

**Soundness:** 3 good
**Presentation:** 2 fair
**Contribution:** 3 good
**Rating:** 6
**Confidence:** 3

**Summary:**

This paper studies the Nesterov accelerated gradient method in the context of reinforcement learning. The authors theoretically show that with the true gradient, the proposed accelerated policy gradient with softmax policy parametrization converges to an optimal policy at a $O(1/t^2)$ rate. The empirical evaluation further demonstrates that APG exhibits $O(1/t^2)$ rate and APG could significantly improve the convergence behavior over the standard policy gradient.

**Strengths:**

In general, this paper is well-structured, and the main idea of this work is easy to follow. This paper proposes a novel accelerated policy gradient method with a fast $O(1/t^2)$ convergence rate, which is the first Nesterov accelerated method with a provable guarantee in reinforcement learning. The authors of this paper also develop a new technical analysis for the proposed method to prove its convergence rate. The authors also conduct experiments to verify the efficiency of the proposed method empirically.

**Weaknesses:**

(1) The major concern about this work is that the authors did not present a detailed discussion of the work [1]. The work [1] has shown that a linear convergence rate, which is faster than $O(1/t^2)$, can be achieved by the policy gradient with an exponentially increasing step size. Moreover, the result in [1] is also based on the non-regularized MDP, which is the same setting as in this submission. The authors need to provide a detailed comparison of the theoretical results in [1] and this submission and also discuss the significance of the result in this submission, given the linear convergence rate in [1].


(2) Additionally, since this work discusses the lower bound of policy gradient, it is interesting to show why the linear convergence rate in [1] does not conflict with the lower bound provided in this submission.


(3) The upper bound in this paper has a dependence on the factor $||\frac{1}{\mu}||\_\infty$. The recent work on policy gradient, e.g., [1] [2], has a convergence rate dependent on a tighter factor $||\frac{d\_{\rho}^{\pi\_*}}{\mu}||\_\infty$. Is it possible to sharpen such a factor in the result of this submission?


[1] Lin Xiao. On the convergence rates of policy gradient methods. Journal of Machine Learning Research, 23(282):1–36, 2022.

[2] Alekh Agarwal, Sham M Kakade, Jason D Lee, and Gaurav Mahajan. On the theory of policy gradient methods: Optimality, approximation, and distribution shift. Journal of Machine Learning Research, 22(1):4431–4506, 2021.


=================After rebuttal===================

Thanks for addressing my concerns. I raise my score accordingly.

**Questions:**

Please see the above section.

---

> ### Author Response · Authors · 2023-11-20
> **Response to Reviewer txPi (Part 1)**
>
> **We thank the reviewer for the detailed comments and helpful suggestions.**
>
> ### **Q. A detailed comparison between APG and the results in [Xiao, 2022]**
> We thank the reviewer for the helpful suggestion on offering a comparison between our work and [Xiao, 2022]. Notably, one major difference lies in the algorithm used for acceleration: [Xiao, 2022] focuses on the policy mirror descent methods, which include natural policy gradient (NPG) as a special case (under direct tabular parameterization), while we focus on Nesterov accelerated gradient for RL (under softmax parameterization), which is a momentum-based approach. Accordingly, we provide a detailed comparison of these two family of approaches:
>
> **(1) Momentum:**
>
> **Empirical Aspect:** In practice, momentum is one of the most commonly used approaches for acceleration in policy optimization for RL, mainly due to its simplicity. Specifically, a variety of classic momentum methods (e.g., Nesterov momentum, heavy-ball method, and AdaGrad) have been already implemented in optimization solvers and deep learning frameworks (e.g., PyTorch) as a basic machinery and shown to improve the empirical convergence results than the standard policy gradient methods like A2C and PPO (e.g., see [Henderson et al., 2018]).
>
> **Theoretical Aspect:** However, despite the popularity of these approaches in empirical RL, it has remained largely unknown whether momentum could indeed improve the theoretical global convergence rates of RL algorithms. To address this fundamental question, we establish the very first global convergence rate of Nesterov momentum in RL (termed APG in our paper) and show that APG could improve the rate of PG from $O(1/t)$ to $\tilde{O}(1/t^2)$ (in the constant step-size regime). Given the wide application of momentum, our analytical framework and the insights from our convergence result (e.g., local near-concavity) serve as an important first step towards better understanding the momentum approach in RL.
>
> **Theoretical Advantage:** In addition, we would like to highlight that in the constant step-size regime (which is one of the most widely adopted setting in practice), our proposed APG indeed achieves a superior global convergence rate of $\tilde{O}(1/t^2)$, which is better than the $O(1/t)$ rate of vanilla PG and the $O(1/t)$ rate of NPG under constant step sizes.
>
> **(2) Natural Gradients (or more generally, Policy Mirror Descent):**
> Theoretical Aspect: Natural policy gradient (NPG), another RL acceleration technique, borrows the idea from the natural gradient, which uses the inverse of Fisher information matrix as the preconditioner of the gradient and can be viewed as achieving approximate steepest descent in the distribution space [Amari, 1998; Kakade, 2001]. Moreover, under direct policy parameterization, NPG is also known as a special case of policy mirror descent (PMD) with KL divergence as the proximal term [Shani et al., 2020; Xiao et al., 2022]. Regarding the global convergence rates, NPG has been shown to achieve:
> (i) $O(1/t)$ rate under constant step sizes [Agarwal et al., 2020; Xiao, 2022]
> (ii) linear convergence either under adaptively increasing step sizes [Khodadadian et al., 2021] or non-adaptive geometrically increasing step sizes [Xiao, 2022].
> Notably, under direct policy parameterization and geometrically increasing step sizes, similar linear convergence results can also be established for the more general PMD method [Xiao, 2022].
>
> **Empirical Issues:**
> Despite the above convergence rates, as also pointed out by [Mei et al., 2021a], it is known that NPG in general requires solving a costly optimization problem in each iteration (cf. [Kakade, 2001; Agarwal et al., 2020]) even with the help of compatible approximation theorem [Kakade, 2001] (unless the policy parametrization adopts some special forms, e.g., log-linear policies [Yuan et al., 2023]). Notably, the TRPO algorithm, a variant of NPG, is known to suffer from this additional computational overhead. Similarly, the PMD in general also requires solving a constrained optimization problem in each policy update [Xiao, 2022]. This computational complexity could be a critical factor for RL in practice.
>
> **(We further add a new section in Appendix I for a detailed comparison of the acceleration methods in RL.)**

---

> ### Author Response · Authors · 2023-11-20
> **Response to Reviewer txPi (Part 2)**
>
> ### **Q. Additionally, since this work discusses the lower bound of policy gradient, it is interesting to show why the linear convergence rate in [Xiao 2022] does not conflict with the lower bound provided in this submission**
> The absence of conflict between the linear convergence rate described in [Mei et al., 2020] and the lower bound established for policy gradient in [Xiao, 2022] stems from the different parameterization methods and algorithms used. The $O(1/t)$ lower bound for policy gradient, as demonstrated in [Mei et al., 2020], is rooted in the softmax parameterization and the vanilla policy gradient approach, whereas the linear convergence rate in [Xiao, 2022] is based on direct parameterization and policy mirror descent.
>
> &nbsp;
>
> ### **Q. Sharpening the factor in the convergence rate of APG**
>
> We agree with the reviewer that it is indeed feasible to sharpen the bound by replacing $\lVert\frac{1}{\mu}\rVert_\infty$ with $\lVert\frac{d_{\rho}^{\pi_*}}{\mu}\rVert_\infty$. Specifically, in Equation (60) of the original manuscript, we opt for simplicity by directly taking 1 as an upper bound of $d_{\rho}^{\pi_*}$, resulting in an ostensibly looser bound $\lVert\frac{1}{\mu}\rVert_\infty$, albeit with a more lucid expression. Accordingly, we have updated the paper to incorporate the tighter bound with the term $\lVert\frac{d_{\rho}^{\pi_*}}{\mu}\rVert_\infty$ in Lemma 11 on page 18, Theorem 2 on pages 7 & 35, and Theorem 7 on pages 45 & 46.
>
> &nbsp;
>
> ### **References**
> - [Mei et al., 2020] Jincheng Mei, Chenjun Xiao, Csaba Szepesvari, and Dale Schuurmans, “On the Global Convergence Rates of Softmax Policy Gradient Methods,” ICML 2020.
> - [Xiao, 2022] Lin Xiao, “On the Convergence Rates of Policy Gradient Methods,” JMLR 2022.

---

### Official Review · Reviewer_6oea · 2023-11-09

**Soundness:** 3 good
**Presentation:** 3 good
**Contribution:** 3 good
**Rating:** 6
**Confidence:** 4

**Summary:**

This paper studies the convergence of Accelerated Policy Gradient (APG) with restart as shown in Algorithm 2. The main results show that this algorithm achieves a $\tilde{O}(1/t^2)$ convergence rate toward globally optimal policy in terms of value sub-optimality. The technical innovation includes showing that the value function is nearly $C$-concave when optimal action's probability is large enough (locally around optimal policy), as well as using AGD's $\tilde{O}(1/t^2)$ convergence results, and asymptotic global convergence in Agarwal et al.

**Strengths:**

1. Answering whether Nesterov's acceleration can be used in policy gradient is an interesting question.
2. The technical challenges are well explained and real, including using momentum in non-convexity, and unbounded parameters.
3. The simulations verify the proved rates.

**Weaknesses:**

1. There already exist acceleration methods for policy gradient, including natural policy gradient, and normalization which both lead to an exponential convergence rate, which might make this slower acceleration not that attractive to the community.

**Questions:**

1. How do you compare the acceleration provided by momentum with faster acceleration methods for policy gradient, such as natural policy gradient and normalization, as well as regularization?

2. Any idea of using generalizing the methods to stochastic settings, where the policy gradient has to be estimated from samples?

---

> ### Author Response · Authors · 2023-11-20
> **Response to Reviewer 6oea (Part 1)**
>
> **We thank the reviewer for the thoughtful comments and insightful suggestions.**
>
> ### **Q. Comparison of the acceleration methods for policy gradient in RL**
>
> We thank the reviewer for the helpful suggestion on providing a comparison of acceleration methods. In the optimization literature, there are several major categories of acceleration methods, including momentum, regularization, natural gradient, and normalization. Interestingly, these approaches all play an important role in the context of RL:
>
> **(1) Momentum:**
>
> **Empirical Aspect:** In practice, momentum is one of the most commonly used approaches for acceleration in policy optimization for RL, mainly due to its simplicity. Specifically, a variety of classic momentum methods (e.g., Nesterov momentum, heavy-ball method, and AdaGrad) have been already implemented in optimization solvers and deep learning frameworks (e.g., PyTorch) as a basic machinery and shown to improve the empirical convergence results than the standard policy gradient methods like A2C and PPO (e.g., see [Henderson et al., 2018]).
>
> **Theoretical Aspect:** However, despite the popularity of these approaches in empirical RL, it has remained largely unknown whether momentum could indeed improve the theoretical global convergence rates of RL algorithms. To address this fundamental question, we establish the very first global convergence rate of Nesterov momentum in RL (termed APG in our paper) and show that APG could improve the rate of PG from $O(1/t)$ to $\tilde{O}(1/t^2)$ (in the constant step-size regime). Given the wide application of momentum, our analytical framework and the insights from our convergence result (e.g., local near-concavity) serve as an important first step towards better understanding the momentum approach in RL.
>
> **Theoretical Advantage:** In addition, we would like to highlight that in the unregularized constant step-size regime of  (which is one of the most widely adopted setting in practice), our proposed APG indeed achieves a superior global convergence rate of $\tilde{O}(1/t^2)$, which is better than the $O(1/t)$ rate of vanilla PG and the $O(1/t)$ rate of NPG under constant step sizes.
>
> **(2) Regularization:**
>
> **Theoretical Aspect:** In the convex optimization literature, it is known that adding a strongly convex regularizer achieves acceleration by changing the optimization landscape. In the context of RL, despite the non-concave objective, regularization has also been shown to achieve faster convergence, such as the log-barrier regularization with $O(1/\sqrt{t})$ rate [Agarwal et al., 2020] and the entropy regularization (i.e., adding a policy entropy bonus to the reward) with linear convergence rate $O(e^{-ct})$ [Mei et al., 2020], both in the exact gradient setting with constant step sizes.
>
> Despite the above convergence results, one common attack on regularization approaches is that they essentially change the objective function to that of regularized MDPs and hence do not directly address the original objective in unregularized RL, as pointed out by [Xiao, 2022; Mei et al., 2021a].

---

> ### Author Response · Authors · 2023-11-20
> **Response to Reviewer 6oea (Part 2)**
>
> ### **Q. Comparison of the acceleration methods for policy gradient in RL (continued)**
>
> **(3) Natural Gradients (or more generally, Policy Mirror Descent):**
>
> **Theoretical Aspect:** Natural policy gradient (NPG), another RL acceleration technique, borrows the idea from the natural gradient, which uses the inverse of Fisher information matrix as the preconditioner of the gradient and can be viewed as achieving approximate steepest descent in the distribution space [Amari, 1998; Kakade, 2001]. Moreover, under direct policy parameterization, NPG is also known as a special case of policy mirror descent (PMD) with KL divergence as the proximal term [Shani et al., 2020; Xiao et al., 2022]. Regarding the global convergence rates, NPG has been shown to achieve:
>
> (i) $O(1/t)$ rate under constant step sizes [Agarwal et al., 2020; Xiao, 2022];
>
> (ii) linear convergence either under adaptively increasing step sizes [Khodadadian et al., 2021] or non-adaptive geometrically increasing step sizes [Xiao, 2022].
> Notably, under direct policy parameterization and geometrically increasing step sizes, similar linear convergence results can also be established for the more general PMD method [Xiao, 2022].
>
> **Empirical Issues:**
> Despite the above convergence rates, as also pointed out by [Mei et al., 2021a], it is known that NPG in general requires solving a costly optimization problem in each iteration (cf. [Kakade, 2001; Agarwal et al., 2020]) even with the help of compatible approximation theorem [Kakade, 2001] (unless the policy parametrization adopts some special forms, e.g., log-linear policies [Yuan et al., 2023]). Notably, the TRPO algorithm, a variant of NPG, is known to suffer from this additional computational overhead. Similarly, the PMD in general also requires solving a constrained optimization problem in each policy update [Xiao, 2022]. This computational complexity could be a critical factor for RL in practice.
>
> **(4) Normalization:**
>
> **Theoretical Aspect:** In the exact gradient setting, the normalization approach, which exploits the non-uniform smoothness by normalizing the true policy gradient by its L2 gradient norm, has also been shown to achieve acceleration in RL. Specifically, the geometry-aware normalized (GNPG) has been shown to exhibit linear convergence rate under softmax policies [Mei et al., 2021a]. This rate could be largely attributed to the effective increasing step size induced by the normalization technique.
>
> **Empirical Issues:** Despite the linear convergence property, it has been already shown that GNPG could suffer from divergence in the standard on-policy stochastic setting, even in simple 1-state MDPs. This phenomenon is mainly due to the committal behavior induced by the increasing effective step size. This behavior could substantially hinder the use of GNPG in practice.
>
> **We have also included the above detailed comparison in the updated manuscript (cf. Appendix I).**

---

> ### Author Response · Authors · 2023-11-20
> **Response to Reviewer 6oea (Part 3)**
>
> ### **Q. Generalizing the APG method to stochastic settings, where the policy gradient has to be estimated from samples?**
>
> **Algorithmic Aspect:** In the stochastic setting, we can extend APG (called Stochastic APG, or SAPG) by following a procedure similar to the classic stochastic policy gradient method by sampling a batch of data at each iteration (cf. Algorithm 7 in Appendix H in the updated manuscript). Subsequently, we estimate the stochastic gradients based on this batch and proceed to update our policy parameters. As for momentum, we directly update by incorporating the direction calculated as the difference between the current parameters and the previous parameters, scaled by the momentum step size. The above design resembles the typical Nesterov acceleration in the stochastic optimization literature [Ghadimi and Lan, 2016].
>
> **Empirical Aspect:** We conduct additional experiments to evaluate the Stochastic APG on two types of environments:
>
> (i) A 5-state, 5-action Markov Decision Process (MDP): In this part, we would like to have a glance at whether the improved convergence rate of APG over PG could be preserved in the stochastic setting. Please refer to Figure 4 in Appendix H.1 in our updated paper for the learning curves. The results show that in this example, stochastic PG has a rate of about $O(1/t)$ and SAPG exhibits a rate faster than $O(1/t)$. This shows the good potential of APG in the stochastic setting.
>
> (ii) More complex benchmark environments, including LunarLander-v2 and Bipedal Walker:
> In this part, we compare SAPG with the stochastic PG and the stochastic Heavy Ball Policy Gradient (HBPG). As shown in the table below, we can see that SAPG still exhibits faster convergence over PG in both LunarLander and BipedalWalker. Furthermore, SAPG also enjoys better stability than HBPG in LunarLander. Please refer to Appendix H in our updated paper for the full learning curves.
>
> | Total Environment Interactions / Algorithms | LunarLander-v2 (1.5e6 steps) | LunarLander-v2 (2.5e6 steps) | LunarLander-v2 (4e6 steps) | BipedalWalker-v3 (6e6 steps) | BipedalWalker-v3 (1.2e7 steps) | BipedalWalker-v3 (2e7 steps) |
> |---------------------------------------------|:----------------------------:|:----------------------------:|:--------------------------:|:----------------------------:|:------------------------------:|------------------------------|
> |           **APG (mean / std dev)**          |    **59.41709 / 28.38159**   |   **177.36543 / 82.97050**   |   **235.86339 / 3.75312**  |   **115.29717 / 51.89256**   |    **254.83298 / 19.24260**    |   **249.83565 / 28.13334**   |
> |            HBPG (mean / std dev)            |      27.76393 / 20.25072     |     180.82551 / 56.37636     |    151.22692 / 83.93418    |      55.92438 / 113.5947     |      216.47174 / 70.40871      |     232.24193 / 59.57783     |
> |             PG (mean / std dev)             |     -144.25341 / 25.66134    |     -128.69148 / 13.92372    |    -107.66071 / 3.56015    |      1.47716 / 23.27089      |      191.22486 / 20.54212      |     249.27057 / 12.74333     |
>
> **(The results reported above are the averaged over 3 random seeds.)**
>
> **Theoretical Aspect:**
> To extend the convergence analysis of APG to the stochastic setting, we expect to build on the novel local near-concavity property and borrow ideas from the stochastic Nesterov's Accelerated Gradient  literature (e.g., [Ghadimi and Lan, 2016]). Specifically, we need to establish the following two results in the stochastic setting: (i) We need to demonstrate that SAPG will reach the nearly locally concave region within a finite number of time steps with high probability, regardless of the initialization. (ii) We also need to relax the convergence result of stochastic NAG typically for concave functions to that with only near concavity.

---

> ### Author Response · Authors · 2023-11-20
> **Response to Reviewer 6oea (Part 4)**
>
> ### **References:**
>
> - [Henderson et al., 2018] Peter Henderson, Joshua Romoff, and Joelle Pineau, “Where Did My Optimum Go?: An Empirical Analysis of Gradient Descent Optimization in Policy Gradient Methods," European Workshop on Reinforcement Learning, 2018.
> - [Agarwal et al., 2020] Alekh Agarwal, Sham M. Kakade, Jason D. Lee, and Gaurav Mahajan, “On the Theory of Policy Gradient Methods: Optimality, Approximation, and Distribution Shift,” COLT 2020.
> - [Mei et al., 2020] Jincheng Mei, Chenjun Xiao, Csaba Szepesvari, and Dale Schuurmans, “On the Global Convergence Rates of Softmax Policy Gradient Methods,” ICML 2020.
> - [Khodadadian et al., 2021] Sajad Khodadadian, Prakirt Raj Jhunjhunwala, Sushil Mahavir Varma, and Siva Theja Maguluri, “On the Linear Convergence of Natural Policy Gradient Algorithm,” CDC 2021.
> - [Xiao, 2022] Lin Xiao, “On the Convergence Rates of Policy Gradient Methods,” JMLR 2022.
> - [Mei et al., 2021a] Jincheng Mei, Yue Gao, Bo Dai, Csaba Szepesvari, and Dale Schuurmans, “Leveraging Non-uniformity in First-order Non-convex Optimization,” ICML 2021.
> - [Amari, 1998] Shun-ichi Amari, “Natural Gradient Works Efficiently in Learning,” Neural Computation, 1998
> - [Kakade, 2001] Sham M. Kakade, "A Natural Policy Gradient," Advances in Neural Information Processing Systems, 2001.
> - [Shani et al., 2020] Lior Shani, Yonathan Efroni, and Shie Mannor, “Adaptive Trust Region Policy Optimization: Global Convergence and Faster Rates for Regularized MDPs,” AAAI 2020.
> - [Yuan et al., 2023] Rui Yuan, Simon S. Du, Robert M. Gower, Alessandro Lazaric, and Lin Xiao, “Linear Convergence of Natural Policy Gradient Methods with Log-Linear Policies,” ICLR 2023.
> - [Mei et al., 2021b]  Jincheng Mei, Bo Dai, Chenjun Xiao, Csaba Szepesvári, and Dale Schuurmans, “Understanding the Effect of Stochasticity in Policy Optimization,” NeurIPS 2021.
> - [Ghadimi and Lan, 2016] Saeed Ghadimi and Guanghui Lan, “Accelerated Gradient Methods for Nonconvex Nonlinear and Stochastic Programming,” Mathematical Programming, 2016.

---

### Official Review · Reviewer_aHMD · 2023-11-09

**Soundness:** 4 excellent
**Presentation:** 4 excellent
**Contribution:** 2 fair
**Rating:** 5
**Confidence:** 3

**Summary:**

This paper investigates the use of Nesterov’s accelerated gradient onto policy gradient. The work terms this methods accelerated
policy gradient. This improves the convergence rate from $O(1/t)$ to $O(1/t^2)$, provided that one has access to the true gradient. The work points out the intuition how the acceleration is benefited from the momentum.

The results are based on several assumptions on the RL problem structure. First, the surrogate initial state distribution has to be strictly positive for every state. Second, the optimal action has to be unique at every state.

Some numerical test are provided in the manuscript.

**Strengths:**

1. The work uses Nestrov acceleration on policy gradient and improves the convergence rate.
2. Techniques used in the proofs could be of independent interest.

**Weaknesses:**

One major concern is on the assumptions, especially Assumption 3 and Assumption 4. I believe these assumptions are way too strong, and drastically reduce the complexity of reinforcement learning problems and make the optimization landscape much easier to tackle with. The setting of true gradient and initial state distribution are also strong, though acceptable. I would prefer if the work, that claims they are the first to achieve Nestrov acceleration on PG, to be under a much more general setting.

**Questions:**

N/A

---

> ### Author Response · Authors · 2023-11-20
> **Response to Reviewer aHMD (Part 1)**
>
> **We thank the reviewer for the constructive feedback and helpful suggestions.**
>
> ### **Q. Clarification on the assumptions**
> We would like to emphasize the general applicability of Assumptions 1 and 4.
>
> - To begin with, Assumption 1, which requires the strict positivity of the surrogate initial state distribution, is a standard assumption adopted in the existing RL literature on the global convergence of policy gradient methods [Agarwal et al., 2020; Ding et al., 2020; Mei et al., 2020; Mei et al., 2021 Xiao, 2022]. Moreover, we would like to highlight that the surrogate initial distribution $\mu$, which is different from the true initial distribution $\rho$, is a parameter that could be freely chosen by the RL algorithm. With the help of the surrogate distribution $\mu$, the policy gradient methods (e.g., PG and APG) could work without the knowledge about the true distribution $\rho$, as also indicated and adopted by the existing works (e.g., [Agarwal et al., 2020; Mei et al., 2020; Xiao, 2022])
>
> - Regarding Assumption 4, we would like to emphasize two important facts:
>
>    (i) The surrogate initial distribution $\mu$, which is different from the true initial distribution $\rho$, is a parameter that could be chosen by the RL algorithm. Notably, with the help of the surrogate distribution $\mu$, the policy gradient methods (e.g., PG and APG) could work without the knowledge about the true distribution $\rho$, as also indicated and adopted by the existing works (e.g., [Agarwal et al., 2020; Mei et al., 2020; Xiao, 2022]).
>
>    (ii) As mentioned in Remark 2 of our paper, Assumption 4 is a fairly mild requirement and can be met with probability one by choosing a surrogate initial distribution $\mu$ uniformly at random from the unit simplex in practice, as also acknowledged by Reviewer y1oq.
>
> &nbsp;

---

> ### Author Response · Authors · 2023-11-20
> **Response to Reviewer aHMD (Part 2)**
>
> ### **Q. Convergence analysis in the true gradient setting**
> To begin with, we would like to highlight that the true gradient setting is widely employed as the first step of convergence rate analysis in RL, as supported by [Agarwal et al., 2020; Mei et al., 2020; Mei et al., 2021; Xiao, 2022]. Moreover, the structural properties obtained from the true gradient setting could subsequently inspire the convergence analysis for the more realistic stochastic settings. For example, the gradient dominance property (i.e., Polyak-Lojasiewicz condition) of PG was first discovered in the true gradient setting (e.g., [Agarwal et al., 2020; Mei et al., 2020]) and subsequently facilitates the convergence analysis of PG in the stochastic setting (e.g. [Yuan et al., 2022]).
> In our work, our convergence analysis of APG unveils several important structural properties of Nesterov momentum in RL, such as the local near-concavity property, the absorbing behavior of this locally nearly-concave region, and the validity of using a surrogate optimal parameter with bounded norm. These useful properties, which are obtained through analysis under true gradients, could naturally inspire future analysis in the stochastic settings.
>
> Additionally, to the best of our knowledge, even within the true gradient setting, our work represents the first attempt to characterize the global convergence rate of the Nesterov accelerated gradient in the context of RL.
>
> On the other hand, as suggested by Reviewer y1oq, we further evaluate the performance of APG beyond the true gradient setting in more diverse RL environments. Specifically, we extend APG to the stochastic setting (please see Algorithm 7 in Appendix H for the pseudo code) and test the stochastic APG in LunarLander-v2 and BipedalWalker-v3 using the estimated gradients. The detailed experimental results and the learning curves can be found in Appendix H in the updated manuscript. At a glance, the following table highlights that APG consistently outperforms HBPG and PG in terms of convergence behavior in both LunarLander-v2 and BipedalWalker-v3.
>
> | Total Environment Interactions / Algorithms | LunarLander-v2 (1.5e6 steps) | LunarLander-v2 (2.5e6 steps) | LunarLander-v2 (4e6 steps) | BipedalWalker-v3 (6e6 steps) | BipedalWalker-v3 (1.2e7 steps) | BipedalWalker-v3 (2e7 steps) |
> |---------------------------------------------|:----------------------------:|:----------------------------:|:--------------------------:|:----------------------------:|:------------------------------:|------------------------------|
> |           **APG (mean / std dev)**          |    **59.41709 / 28.38159**   |   **177.36543 / 82.97050**   |   **235.86339 / 3.75312**  |   **115.29717 / 51.89256**   |    **254.83298 / 19.24260**    |   **249.83565 / 28.13334**   |
> |            HBPG (mean / std dev)            |      27.76393 / 20.25072     |     180.82551 / 56.37636     |    151.22692 / 83.93418    |      55.92438 / 113.5947     |      216.47174 / 70.40871      |     232.24193 / 59.57783     |
> |             PG (mean / std dev)             |     -144.25341 / 25.66134    |     -128.69148 / 13.92372    |    -107.66071 / 3.56015    |      1.47716 / 23.27089      |      191.22486 / 20.54212      |     249.27057 / 12.74333     |
>
> **(The results reported above are the averaged over 3 random seeds.)**
>
> &nbsp;
>
> ### **References**
> - [Mei et al., 2020] Jincheng Mei, Chenjun Xiao, Csaba Szepesvari, and Dale Schuurmans, “On the Global Convergence Rates of Softmax Policy Gradient Methods,” ICML 2020.
> - [Agarwal et al., 2020] Alekh Agarwal, Sham M Kakade, Jason D Lee, and Gaurav Mahajan, “On the Theory of Policy Gradient Methods: Optimality, Approximation, and Distribution Shift,” COLT 2020.
> - [Xiao, 2022] Lin Xiao, “On the Convergence Rates of Policy Gradient Methods,” JMLR 2022.
> - [Ding et al., 2020] Ding, Dongsheng, et al. "Natural policy gradient primal-dual method for constrained markov decision processes." NeurIPS 2020.
> - [Mei et al., 2021] Jincheng Mei, et al. "Leveraging non-uniformity in first-order non-convex optimization," ICML 2021.
> - [Yuan et al., 2022] Rui Yuan, Robert M. Gower, and Alessandro Lazaric, “A General Sample Complexity Analysis of Vanilla Policy Gradient,” AISTATS 2022.

---

> ### Comment · Reviewer_aHMD · 2023-11-22
> **Reply**
>
> I thank the authors for providing additional arguments. Assumptions and true gradient being used in previous works does not (directly) justify that they are not strong. It is natural to expect the assumptions to be lifted and the more vanilla setting of RL to be analyzed. Meanwhile I did appreciate the new techniques involved such as local near-concavity. It seems to be on a good track to incorporating Nestrove momentum into policy gradient.

---

### Meta-Review · Area_Chair_3enb · 2023-12-04

**Metareview:**

This paper studies the convergence of Nesterov's Accelerated gradient method in policy optimization, where an $O(1/t^2)$ convergence rate to the globally optimal policy is derived. Under certain conditions, the authors show an interesting property that the value function is nearly-concave when the optimal action's probability is close to 1, which happens when the algorithm converges to a near optimal solution. However, the paper also has some issues that the assumption 4 (AS4) is strong and unnatural.

In particular, the justification of Remark 3 for AS4 is not correct. This is because AS4 is made for all pairs of $\pi$ and $\pi'$, while Remark 3 is only for a specific pair of $\pi$ and $\pi'$. Consider the following naive case with S = {s1,s2}, A = {a1,a2}, R(s1,a1) = R(s2,a1) = 1, R(s1,a2) = R(s2,a2) = 0. While $P(s_1|s_1,\cdot) = 1, P(s_2|s_2,\cdot) = 1$. That is, 2 independent 2-armed bandit with deterministic 0-1 reward. Then for any specific pair of $\pi$ and $\pi'$, the authors Remark 3 holds. However, as the AS4 is made for all pairs of policies, no matter which $\mu$ they choose, there exist infinitely many pairs of $\pi$ and $\pi'$ that violate AS4.

The AC believes that this issue can be fixed with some more careful discussion. However, the AC cannot guarantee such correction can be done in the camera-ready preparation where no one can check this. Therefore, unfortunately, the AC should put a rejection to this paper. This is a borderline rejection. However, the authors are encouraged to fix this issue (as well as the reviewers' concerns) and resubmit to another conference.

**Justification For Why Not Higher Score:**

There is a theoretical issue of the paper that has yet been justified at this moment.

**Justification For Why Not Lower Score:**

N/A

---

### Decision · Program_Chairs · 2024-01-16

Reject